# Lin28B-high breast cancer cells promote immune suppression in the lung pre-metastatic niche via exosomes and support cancer progression

Meiyan Qi[1,4], Yun Xia[2,4], Yanjun Wu[1], Zhuo Zhang[3], Xinyu Wang[1], Liying Lu[1], Cheng Dai[1], Yanan Song[1], Keying Xu[1], Weiwei Ji[1] & Lixing Zhan[1✉]

The formation of pre-metastatic niche is a key step in the metastatic burden. The pluripotent factor Lin28B is frequently expressed in breast tumors and is particularly upregulated in the triple negative breast cancer subtype. Here, we demonstrate that Lin28B promotes lung metastasis of breast cancer by building an immune-suppressive pre-metastatic niche. Lin28B enables neutrophil recruitment and N2 conversion. The N2 neutrophils are then essential for immune suppression in pre-metastatic lung by PD-L2 up-regulation and a dysregulated cytokine milieu. We also identify that breast cancer-released exosomes with low let-7s are a prerequisite for Lin28B-induced immune suppression. Moreover, Lin28B-induced breast cancer stem cells are the main sources of low-let-7s exosomes. Clinical data further verify that high Lin28B and low let-7s in tumors are both indicators for poor prognosis and lung metastasis in breast cancer patients. Together, these data reveal a mechanism by which Lin28B directs the formation of an immune-suppressive pre-metastatic niche.

[1] CAS Key Laboratory of Nutrition, Metabolism and Food Safety, Shanghai Institute of Nutrition and Health, Shanghai Institutes for Biological Sciences, University of Chinese Academy of Sciences, Chinese Academy of Sciences, 320 Yueyang Road, Shanghai 200031, China. [2] Department of Thyroid and Breast Surgery, Tongji Hospital, Tongji Medical College, Huazhong University of Science and Technology, 1095 Jiefang Avenue, Wuhan 430030, China. [3] Optogenetics & Synthetic Biology Interdisciplinary Research Center, State Key Laboratory of Bioreactor Engineering, East China University of Science and Technology, 130 Mei Long Road, Shanghai 200237, China. [4] These authors contributed equally: Meiyan Qi, Yun Xia. ✉email: lxzhan@sibs.ac.cn

Lin28, the evolutionarily conserved RNA-binding protein, is highly expressed in embryonic stem cells and the master regulator of self-renewal of embryonic stem cells[1,2]. MicroRNA let-7 family members are the regulatory target of Lin28[3] and the inhibitory Lin28-let-7s axis is the primary mechanism for Lin28 modulating stem cell self-renewal[4]. In mammals, there are two closely related family members, Lin28A and Lin28B. In contrast to their rare expression in most adult tissues, they are aberrantly overexpressed in a wide spectrum of human cancer types, and their increased expression has been correlated with clinical tumor stages and poor prognosis[5–7]. In breast cancer, Lin28A is mainly expressed in the HER2+ subtype, whereas Lin28B is mainly expressed in the triple-negative breast cancer (TNBC) subtype[3]. Lin28B promotes breast cancer progression by increasing tumor stemness, migration, and invasion[8,9]. However, little is known about its role in metastasis initiation, especially its immune suppression role in the early pre-metastatic niche.

The pre-metastatic niche formed before tumor cells' arrival plays a vital role in sustaining metastatic tumor cell growth. Tumor-derived elements are considered crucial regulators in the initiation of the pre-metastatic niche, which involves cross-talks between tumor-derived factors, distal stromal components, and tumor mobilized bone marrow-derived cells (BMDCs)[10,11]. BMDCs consist of neutrophils, macrophages, monocytes, and dendritic cells (DCs), among which neutrophils are the most abundant population[12]. Although reports indicate that neutrophils inhibit metastatic seeding of tumor cells[13], other studies have suggested a facilitating role in lung colonization of tumor cells[14,15]. What's remarkable is that recent increasing evidence pointed out neutrophils as an immune modulator in primary tumors[14,16,17]. Whatever, it is still unknown whether any signal influencing immune-suppressive mechanisms in the pre-metastatic niche exists, and how important it is to metastasis.

In this work, we seek to explore the mechanism by which Lin28B promotes breast cancer metastasis. We determine that Lin28B can establish an immunosuppressive pre-metastatic niche by inducing neutrophil infiltration and N2 conversion, which depends on the tumor-released exosomes with low let-7s. Moreover, we show that Lin28B induces primary tumors to produce more ALDH+ breast cancer stem cells (BCSCs), the principal source for tumor exosomes with low let-7s. Our study exposes a mechanism by which Lin28B dictates breast cancer metastasis through shaping immunosuppression in the metastatic lung niche.

## Results

**Lin28B is an independent prognostic factor and firmly correlates with metastasis in patients with breast cancer.** Lin28B expression was firstly evaluated in a tumor tissue microarray (TMA) with pairs of tumor and para-tumor tissue from patients with breast cancer (Cohort I, Supplementary Data 1). Consistent with the previous study[7], Lin28B was significantly elevated in tumor tissues relative to their paired non-tumor tissues (Fig. 1a, b). With the fresh tumor specimens, we obtained similar results suggesting that Lin28B was upregulated in 11 out of 24 tumor tissues (Fig. 1c). To analyze the association of Lin28B with breast cancer patients' clinicopathological characteristics, we probed Lin28B expression in a TMA Cohort II with breast tumors from a retrospective cohort of patients (140 cases, Supplementary Data 1). We found that Lin28B was significantly higher in TNBC relative to the luminal or HER2+ subtypes (Fig. 1d). No significant correlation was observed between Lin28B expression and patient age, tumor node metastasis (TNM) stage, vascular invasion, lymph node metastasis, or recurrence (Supplementary Table 1). However, Lin28B expression was associated with clinical

tumor grade and distal metastasis (Fig. 1e, f and Supplementary Table 1).

Moreover, there was an inverse correlation between Lin28B expression and overall post-surgery survival (Fig. 1g). We also assessed whether Lin28B expression serves as an independent predictor of patients' overall survival and found that tumor TNM stage, lymph node metastasis, recurrence, and distal metastasis, besides Lin28B expression, were associated with survival (Supplementary Table 2). Next, a multivariate Cox proportional hazards analysis showed that Lin28B expression was an independent prognosticator of breast cancer's overall survival (Lin28B high, HR = 5.802, $p < 0.001$, see Supplementary Table 2). To dissect the association of Lin28B with the survival of breast cancer patients harboring individual breast cancer subtypes, we probed Lin28B expression in an additional TMA cohort, Cohort III (145 cases, Supplementary Data 1). Combining these data with the results from Cohort II, we determined that high Lin28B expression was correlated with poor prognosis and was an independent prognosticator of overall survival among breast cancer patients for all three breast cancer subtypes (Fig. 1h and Supplementary Tables 3–5), hinting at Lin28B's important role in tumor progression.

To further clarify the clinical relevance of Lin28B expression, we analyzed the Gene Expression Omnibus (GEO) breast cancer cohort for the transcriptomic datasets of the clinical samples. We analyzed the expression profiles of a cohort of 204 patients with detailed metastatic information (GSE12276, Supplementary Data 1) and observed that high Lin28B expression was associated with lung metastasis, but not with brain or bone metastasis ($P = 0.045$ vs 0.300 and 0.369, respectively; see Supplementary Table 6). Together, these findings suggested a role of Lin28B in promoting breast cancer lung metastasis.

**Lin28B overexpression at primary sites is critical for the outgrowth step of macroscopic lung metastasis.** 4TO7 cells can disseminate to the distal organs but cannot produce visible metastatic lesions[18], hence being the desired model to study the metastasis-related mechanism. By the orthotopic breast cancer model, we determined that Lin28B was properly expressed in the primary tumors, which expression did not affect the primary tumor growth (Fig. 1i, j and Supplementary Fig. 1a). However, Lin28B expression markedly initiated the metastatic lung burden (Fig. 1k, l) and reduced the overall survival of tumor-bearing mice (Supplementary Fig. 1b). Distal metastasis prompted by Lin28B was occasionally found in bone, heart, kidney, and skin (Supplementary Fig. 1c). Moreover, we performed similar experiments with the Lin28B transgenic mice. Using the *MMTV-PyMT* mouse model (luminal subtype) (Supplementary Fig. 1d, e), we showed that Lin28B has no influence on primary tumor growth, but promoted the occurrence of lung metastasis (Supplementary Fig. 1f, g). The *MMTV-Neu* mouse model (HER2+ subtype) also demonstrated that Lin28B expression did not accelerate tumor growth at the initial sites, but increased the incidence of lung metastasis (Supplementary Fig. 1h–j). Therefore, we concluded that Lin28B promotes breast tumor distal metastasis with a major focus on the lung tissue.

As a pluripotent factor, Lin28B can reinforce the cell fraction positive for ALDH (Supplementary Fig. 1k). The human breast cancer cell line MDA-MB-231, which is invasive and highly metastatic, expresses high levels of Lin28B. Importantly, Lin28B knockdown in MDA-MB-231 cells reduced the ALDH+ cell fraction (Supplementary Fig. 1l, m). In addition, 4TO7-Lin28B cells formed about twice as many tumorspheres as seen with 4TO7 cells (Supplementary Fig. 1n). Lin28B expression in *PyMT* tumors also resulted in the expression of more stem-related genes

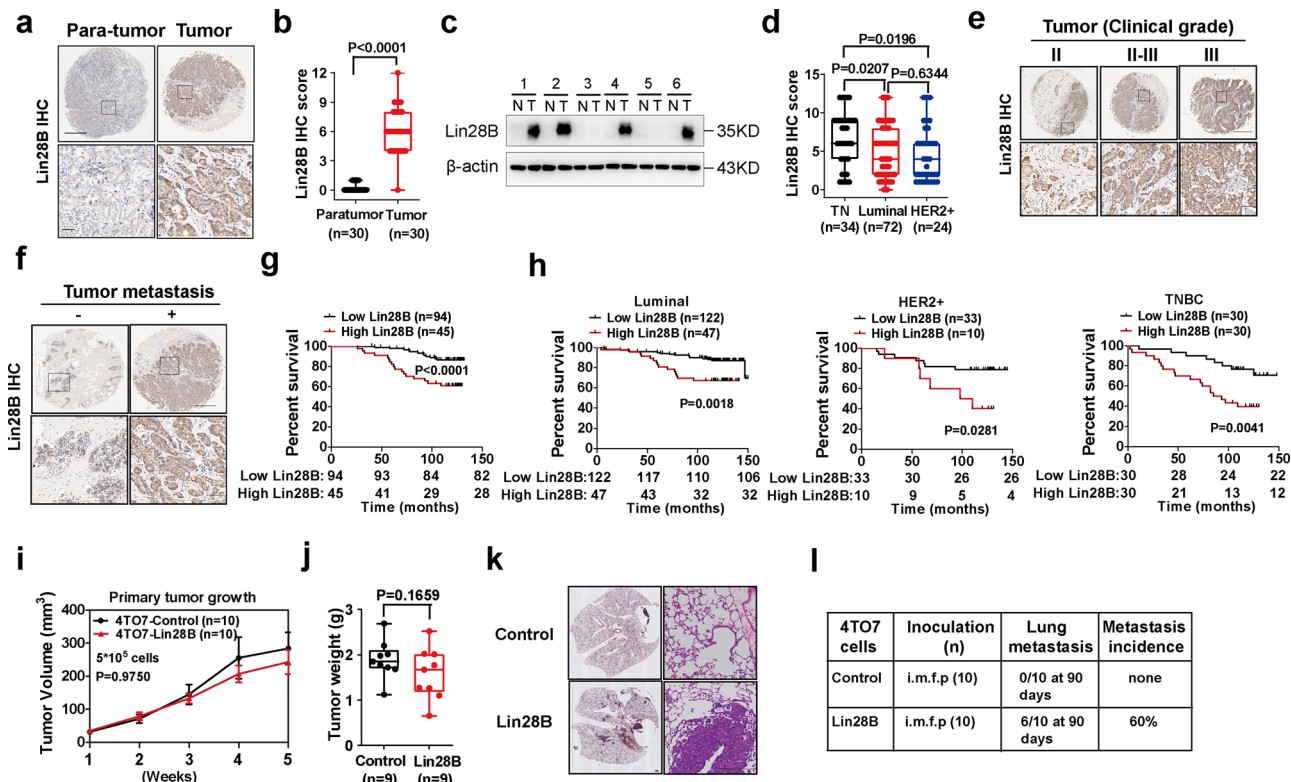

**Fig. 1 Lin28B is upregulated in human breast cancer tissues and its overexpression promotes mouse breast cancer lung metastasis. a** Representative IHC staining of Lin28B in Cohort I TMA. Scale bars: 200 μm. IHC immunohistochemistry. **b** IHC score of Lin28B as in panel **a**. **c** Representative western blotting for Lin28B expression in pairs of tumor and para-tumor tissues from patients with breast cancer. β-actin served as a loading control. T tumor; N para-tumor. **d** IHC score of Lin28B stratified by breast cancer subtypes in Cohort II TMA. TN triple-negative subtype. **e** IHC staining of Lin28B in different clinical stages of breast tumors. Scale bars: 200 μm. **f** IHC staining of Lin28B in breast tumors with different metastatic abilities. Scale bars: 200 μm. **g** Kaplan–Meier survival plot of breast cancer patients based on Lin28B expression in TMA Cohort II. **h** Kaplan–Meier survival plot of breast cancer patients of different subtypes based on Lin28B expression in TMA Cohort II and III. **i** The graphs showed the measured tumor volumes in mice inoculated with indicated cells (n = 10). **j** A similar experiment to panel **i**. Tumor weights are shown (n = 9). **k, l** Representative H&E stained sections of lung metastasis and metastasis incidence summary in mice from experiments in **g**. Scale bars: 200 μm. In **i**, data were presented as mean ± SEM. A two-sided log-rank test was used for statistical analysis of (**g**) and (**h**), one-way ANOVA for (**b**) and (**d**), and two-way ANOVA for (**i**), followed by multiple comparisons, two-tailed Student's t-test for (**j**). For panels (**b**), (**d**), and (**j**), boxes represent data within the 25th to 75th percentiles. Whiskers depict the range of all data points. Horizontal lines within boxes represent mean values. For panels (**e**) and (**f**), experiments were repeated three times independently with similar results; data from one representative experiment are shown. Source data are provided as a Source Data file.

(Supplementary Fig. 1o). Limiting dilution assays confirmed the increased in vivo tumorigenic capacity of Lin28B-expressing 4TO7 tumors (Supplementary Fig. 1p), suggesting that Lin28B facilitates the stem cell properties of breast tumor cells.

**Lin28B overexpression in primary tumors promotes pre-metastatic niche formation in lung tissue by advancing neutrophil accumulation.** To test the role of Lin28B in the pre-metastatic niche, we first traced the time window in which the pre-metastatic niche was to be performed. 4TO7 derivative cells were infected with the retroviral vector TGL encoding herpes simplex virus thymidine-kinase 1, green fluorescent protein (GFP), and firefly luciferase (FL). In turn, we used qRT-PCR to monitor the tumor cell presence in the lung tissue[19]. In the control mice, the *TGL* gene was first detected in the lungs at 4 weeks after tumor inoculation (Supplementary Fig. 2a). Lin28B expression did not exacerbate the TGL gene's presence in the lungs (Supplementary Fig. 2a). Thus, the pre-metastatic niche should be formed within 4 weeks after tumor inoculation. Next, we analyzed lung expression of niche characteristic genes[15]. The results revealed that niche characteristic genes are expressed in the lungs of tumor-bearing mice at 2 weeks after tumor inoculation, and this trend became more evident following tumor

progression (Supplementary Fig. 2b). In addition, Lin28B expression increased their production in the lung tissue (Fig. 2a), indicating the promoting role of Lin28B in pre-metastatic niche formation.

Following tumor inoculation, the percentage of CD45+ cells and the total lung cell number were increased (Supplementary Fig. 2c, d). Lin28B did not change these cell fractions (Supplementary Fig. 2c, d). Among BMDCs, we noticed CD11b+Ly6G+Ly6C+ neutrophils were expanded upon Lin28B expression (Fig. 2b, c); F4/80+ macrophages accumulated similarly, but to absolute numbers that were much lower than those of neutrophils (Supplementary Fig. 2e, f). In contrast, monocytes and DCs showed no differences (Supplementary Fig. 2g). Immunofluorescence (IF) assays confirmed these findings (Fig. 2d and Supplementary Fig. 2h). With immunohistochemistry (IHC) assay, we further revealed that Lin28B significantly increased the expression of the niche characteristic gene Fibronectin (FN) in the pre-metastatic lung. It was frequently observed at the peritracheal and peribronchial (red arrow, Supplementary Fig. 2i). Moreover, neutrophils showed similar accumulation (blue arrow, Supplementary Fig. 2i), suggesting the increased neutrophil distribution in the pre-metastatic niche induced by Lin28B. Moreover, Ly6G was well co-localized with CD45, CD11b, and Ly6C (Fig. 2e), consistent with the FACS results

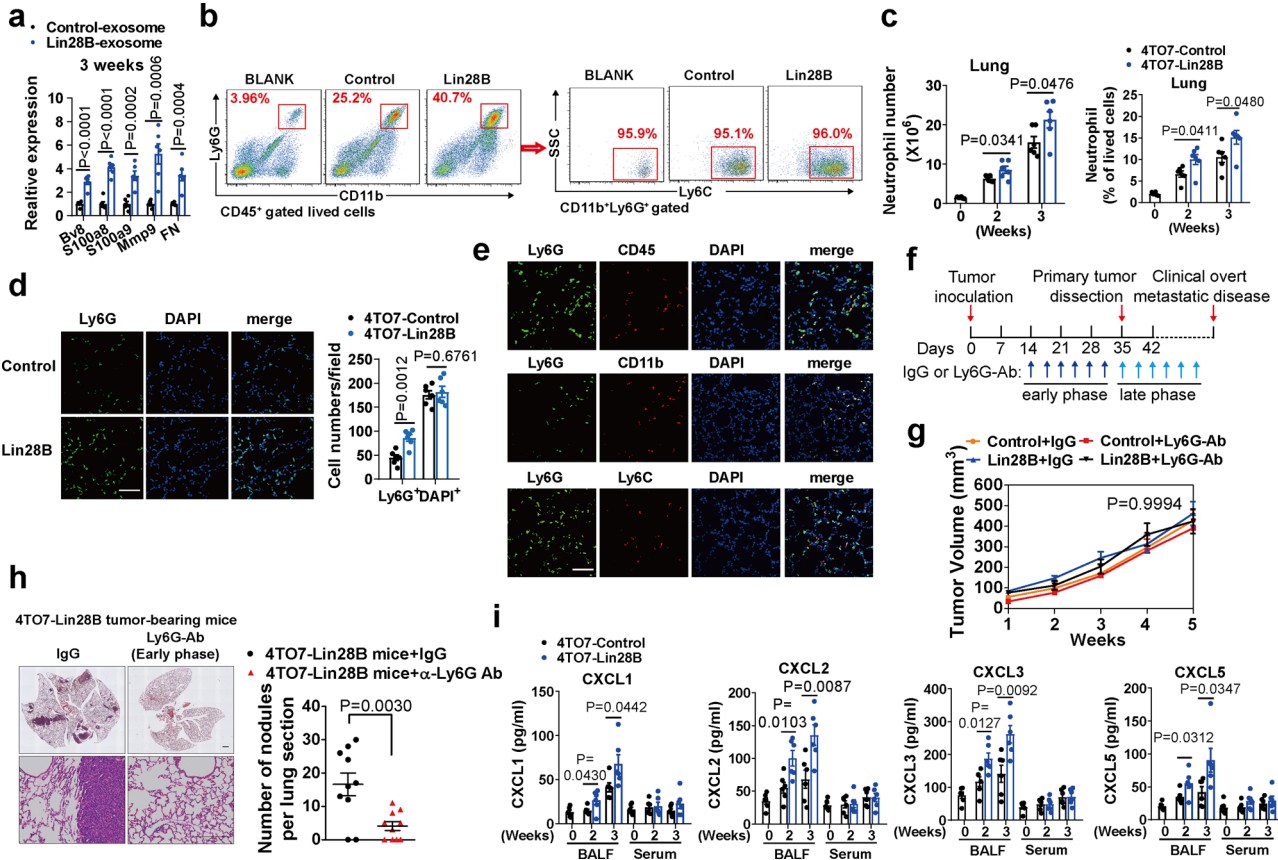

**Fig. 2 Lin28B overexpression in primary tumors promotes pre-metastatic niche formation in lung tissue by advancing neutrophil accumulation. a** Niche-characteristic genes were detected by qRT-PCR in the pre-metastatic lung of tumor-bearing mice at 3 weeks after tumor inoculation. GAPDH served as an internal control ($n = 6$ mice in each group). FN Fibronectin. **b** Representative flow cytometry plots of CD11b+Ly6G+Ly6C+ neutrophils gated on CD45+ lived cells in the pre-metastatic lung at 3 weeks after tumor inoculation. **c** The proportions and numbers of neutrophils as detected in panel b ($n = 6$ mice in each group). **d** IF images and quantitation of Ly6G staining in the pre-metastatic lung of tumor-bearing mice at 3 weeks after tumor inoculation ($n = 6$ random microscopic fields (RMFs) from 6 mice in each group). Scale bar, 100 μm. **e** Representative IF images for Ly6G co-localization with CD45, CD11b, and Ly6C in the pre-metastatic lung at 3 weeks after tumor inoculation. Scar bar: 100 um. **f** Schematic illustration of neutrophil depletion in tumor-bearing mice. The blue arrow indicates neutrophil depletion in the early phase, which starts on day 14 after tumor inoculation until the primary tumors were resected for consecutive six times, while the sky blue arrow indicates neutrophil depletion in the late phase, which begins on day 35 just after primary tumors were removed for consecutive six times. **g** The growth kinetics of primary tumors in mice treated with the indicated antibody ($n = 8$ mice in each group). **h** H&E and quantification of lung metastasis in 4TO7-Lin28B mice in which the neutrophils were depleted in the early phase of tumor growth as illustrated in Fig. 2f ($n = 10$ RMFs from six mice each group). Scale bars: 200 μm. **i** Quantification of CXCLs in BALF and serum of tumor-bearing mice at 3 weeks after tumor inoculation ($n = 6$ mice in each group). Data were presented as mean ± SEM. P values were calculated using two-tailed Student's t-test for (**a**), (**c**), (**d**), (**h**), and (**i**), two-way ANOVA followed by multiple comparisons for (**g**). *$p < 0.05$, **$p < 0.01$, ***$p < 0.001$, ns no significance. Source data are provided as a Source Data file.

indicating that Ly6G+ neutrophils were CD45+CD11b+Ly6C+ (Fig. 2b). Using the *PyMT* and *MMTV-Neu* models, we confirmed that Lin28B-promoted neutrophil recruitment in the pre-metastatic lung (Supplementary Fig. 2j). We concluded that neutrophils were preferentially enriched in the pre-metastatic lung upon Lin28B expression.

To uncover the role of neutrophils upon lung metastasis of breast tumors, we systemically depleted neutrophils in tumor-bearing mice by monoclonal anti-Ly6G antibody (Clone 1A8) (Fig. 2f). An obvious reduction in lung and systemic neutrophils was noted (Supplementary Fig. 2k). Consistent with a previous study[14], neutrophil depletion did not affect primary tumor growth (Fig. 2g). However, it strikingly reduced lung metastasis of 4TO7-Lin28B mice in the early stage of tumor growth (Fig. 2h, denoted by the blue arrow in Fig. 2f). When we depleted neutrophils in the late phase (Fig. 2f, denoted by sky blue arrow), a similar metastasis-inhibiting effect was not found (Supplementary Fig. 2l), excluding the metastasis-promoting role of lung

neutrophils in the late stage. Concordantly, unlike in the early stage, Lin28B did not promote neutrophil accumulation in the late stage (Supplementary Fig. 2m, n). In fact, tumor resection, which is accompanied by neutrophil decline[14], can initiate emergent metastasis, hinting at a possible harmful function of neutrophils in the late phase. However, lung metastasis was not observed in 4TO7-Control mice following neutrophil depletion (Supplementary Fig. 2o), implying that 4TO7-Control cells are deficient in metastatic ability even after harmful neutrophil removal in the late stage. Together, our results showed that neutrophils have a metastasis-promoting role in the early stage of tumorigenesis under Lin28B expression conditions.

There was no distinct difference in neutrophil proportions in bone marrow, spleen, and peripheral blood between both groups (Supplementary Fig. 2p). In addition, the levels of chemokines were specifically elevated in bronchoalveolar lavage fluids (BALFs) in 4TO7-Lin28B mice (Fig. 2i), suggesting specific upregulation of CXCLs in the lung tissues induced by Lin28B.

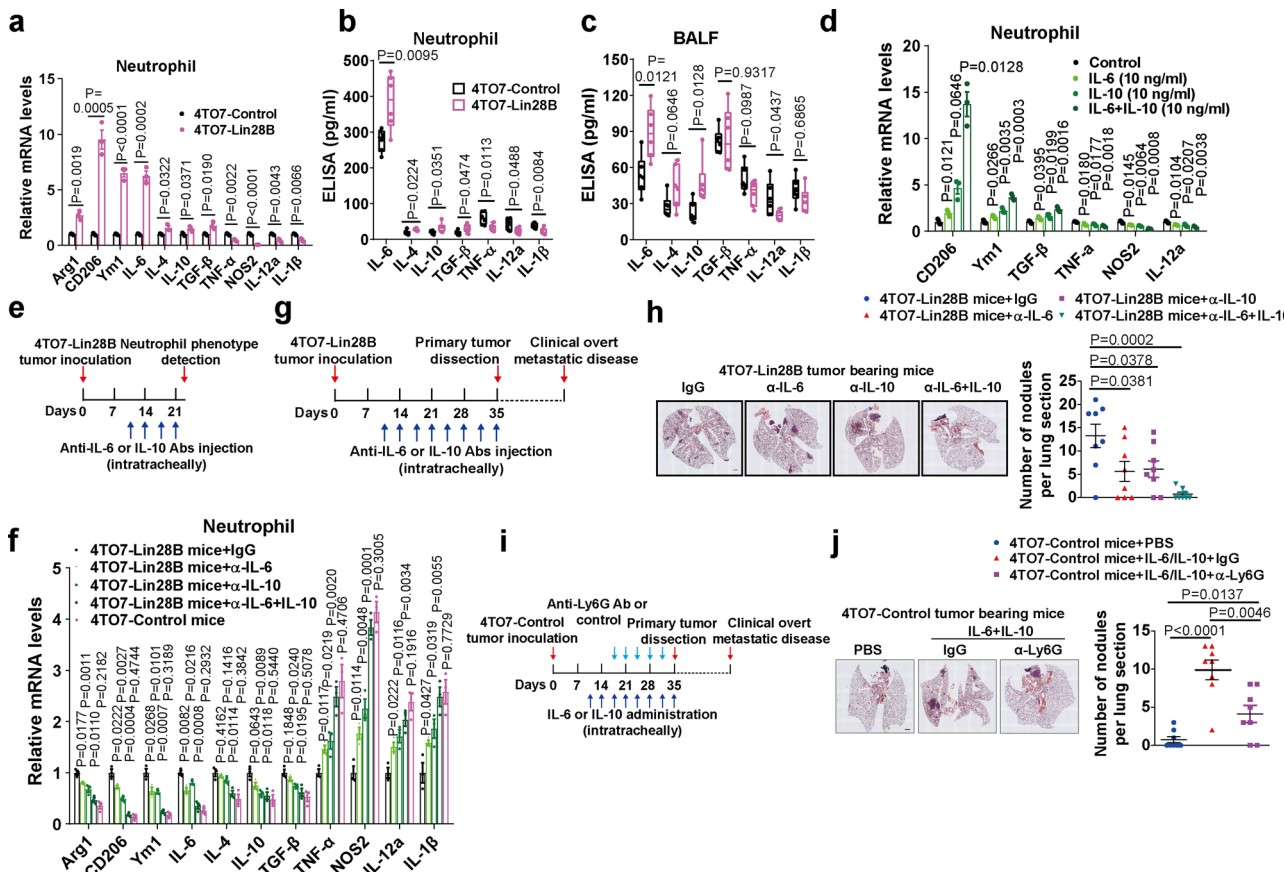

**Fig. 3 Lin28B induces neutrophil N2 conversion and IL-6 and IL-10 are identified as promoters of neutrophil N2 conversion. a** Quantification of N1/N2 marker genes in neutrophils collected from the pre-metastatic lung at 3 weeks after tumor inoculation. GAPDH served as an internal control ($n = 3$ mice in each group). **b** Neutrophils were purified from the pre-metastatic lung at 3 weeks after tumor inoculation and the cells were subjected to in vitro culture for supernatant collection. ELISA assay was conducted to quantify the cytokine expression ($n = 6$ mice in each group). **c** Quantification of indicated cytokines in BALFs of tumor-bearing mice at 3 weeks after tumor inoculation ($n = 6$ mice in each group). **d** The neutrophils purified from the bone marrow of naïve mice were cultured in vitro in the presence of cytokines as indicated. Twenty-four hours later, the mRNA levels of N1/N2 markers of neutrophils were detected by qRT-PCR. GAPDH served as an internal control ($n = 3$ culturing experiments). **e** Schematic illustration for inhibiting N2 conversation of lung neutrophils by intratracheally injecting with anti-IL-6 Ab, anti-IL-10 Ab, or both combined into 4TO7-Lin28B tumor-bearing mice. **f** Neutrophil phenotypes were detected in mice treated as indicated in panel **e** ($n = 3$ mice in each group). **g** Schematic diagram of N2-conversion inhibition upon the influence of lung metastasis. **h** Image and quantification of lung metastasis of mice treated as indicated in **g** ($n = 8$ RMFs from three mice each group). Scale bar: 200 μm. **i** Schematic diagram for inducing N2 conversion upon lung metastasis. **j** Image and quantification of lung metastasis of mice treated as indicated in **i** ($n = 8$ RMFs from three mice each group). Scale bar: 200 μm. Data were presented as the mean ± SEM, and $p$ values were calculated using a two-tailed Student's $t$-test. *$p < 0.05$, **$p < 0.01$, ***$p < 0.001$, ns no significance. For panels (**b**) and (**c**), boxes represent data within the 25th to 75th percentiles. Whiskers depict the range of all data points. Horizontal lines within boxes represent mean values. Source data are provided as a Source Data file.

**Lin28B induces neutrophil N2 conversion and IL-6 and IL-10 are identified as the promoters of neutrophil N2 conversion.** Phenotypic conversion is considered to be the pivotal mechanism by which neutrophils modulate primary tumor growth[20]. Under certain circumstances, neutrophils can be polarized to the anti-tumor N1 phenotype or the pro-tumor N2 phenotype[21,22]. It is unknown whether a similar mechanism is recapitulated under our experimental conditions. We speculated that Lin28B not only triggers neutrophil recruitment but also enables their phenotypic conversion, thus endowing neutrophils with a metastasis-promoting function. To address this issue, neutrophils in the pre-metastatic lung were purified by FACS, and the expression of marker genes characteristic for classic N1 and N2 subsets then was analyzed by qRT-PCR. *Arg1*, *CD206*, *Ym1*, and *IL-6* were upregulated upon Lin28B expression (Fig. 3a). The expression of *IL-4*, *IL-10*, and *TGF-β* also was slightly increased (Fig. 3a). In contrast, the N1 markers *TNF-α*, *NOS2*, *IL-12a*, and *IL-1β* were downregulated (Fig. 3a). An enzyme-linked immunosorbent assay (ELISA) assay confirmed elevated N2 and reduced N1

cytokine production upon Lin28B expression (Fig. 3b). In addition, we determined that N2 phenotype of neutrophils also appeared in the pre-metastatic lungs in *PyMT* and *MMTV-Neu* mice (Supplementary Fig. 3a, b). Furthermore, N2-converted neutrophils were not detected in the spleen, peripheral blood, or bone marrow (Supplementary Fig. 3c), implying that the conversion occurs in the pre-metastatic lung.

To identify factors responsible for neutrophil N2 conversion, we screened the cytokines that were differentially expressed in BALFs. IL-6 has been reported as an N2 inducer of neutrophils by cooperating with G-CSF[23]. Our results showed that Lin28B increased IL-6 production in BALFs (Fig. 3c). Additionally, IL-10, a known M2 inducer of macrophages[24], was also elevated upon Lin28B expression (Fig. 3c). Moreover, IL-4, TGF-β, TNF-α, and IL-1β were less affected, while IL-12a was downregulated (Fig. 3c). IL-6, IL-10, and IL-12a were specially altered in BALFs, but not in the serum (Supplementary Fig. 3d), indicating that IL-6, IL-10, and IL-12a were locally dis-regulated by Lin28B. By stimulating neutrophils with the cytokines in vitro, we found that IL-6 and

IL-10 can shift neutrophils toward N2 conversion (Supplementary Fig. 3e, f) and the combined stimulation produced a much stronger effect (Fig. 3d).

To address whether IL-6 and IL-10 are required for neutrophil N2 conversion in vivo, we administered 4TO7-Lin28B mice intratracheally with the neutralizing antibodies (Fig. 3e). In anti-IL-6 antibody-treated mice, the N2 conversion of lung neutrophils were markedly inhibited (Fig. 3f). Anti-IL-10 blocking antibody produced a similar effect as well (Fig. 3f). Furthermore, combined depletion impaired neutrophil N2 conversion (Fig. 3f). By administering IL-6 and IL-10 intratracheally into 4TO7-Control mice (Supplementary Fig. 3g), we found that either IL-6 or IL-10 supply drove neutrophil N2 conversion, and the combined treatment of IL-6 and IL-10 produced a stronger effect close to that of Lin28B (Supplementary Fig. 3h), indicating that IL-6 and IL-10 are the main factors to drive neutrophil N2 conversion in vivo.

It was interesting to find that either IL-6 or IL-10 blocking markedly reduced lung metastasis of 4TO7-Lin28B mice, and the combined blockade diminished it (Fig. 3g, h). Conversely, intratracheal administration of the recombinant IL-6 and IL-10 initiated lung metastasis of 4TO7-Control mice, and this effect was reversed by neutrophil depletion (Fig. 3i, j). Thus, our results indicate that IL-6- and IL-10-mediated neutrophil N2 conversion is a key mechanism to promote lung metastasis of breast cancer in the context of Lin28B expression.

**N2-converted neutrophils inhibit T cell proliferation, activation, and Th1 differentiation**. To determine whether immune suppression occurs in the pre-metastatic niche, we first identified the infiltration of T cells in the lungs. Our results revealed a decrease in CD4$^+$ and CD8$^+$ T cells, and thereby a lower CD3$^+$ T cell fraction in 4TO7-Lin28B mice (Fig. 4a). The percentage of T cells was not reduced by Lin28B in peripheral blood or spleen (Supplementary Fig. 4a, b). Moreover, T cell chemokines CXCL9 and 10 in the lungs were not altered by Lin28B (Supplementary Fig. 4c), suggesting that Lin28B does not affect T cell recruitment.

To investigate whether N2 neutrophils were competent in adjusting T cell expansion, we measured the N2 neutrophils' effect on T cell proliferation. As expected, pre-metastatic neutrophils isolated from 4TO7-Lin28B mice (N2 neu) suppressed proliferation of both CD4$^+$ and CD8$^+$ T cells, whereas a similar effect was not found with control neutrophil exposure isolated from 4TO7-Control mice (control neu) (Fig. 4b). We repeated the experiments and obtained similar results using purified neutrophils collected by FACS (Supplementary Fig. 4d). When using a transwell system, the N2 neutrophils lost their ability to restrict T cell proliferation (Supplementary Fig. 4e). Through an antigen-specific immune response system by using OT-1 splenocytes, we confirmed that N2 neutrophils suppressed OT-1 CD8$^+$ T cell proliferation (Fig. 4c).

In addition, the lung-infiltrating T cells showed inactivated properties in 4TO7-Lin28B mice. CD8$^+$ T cells in the lung of 4TO7-Lin28B tumor-bearing host included a smaller CD44$^+$CD62L$^-$ T cell population and expressed lower CTLA-4 and granzyme B (Fig. 4d and Supplementary Fig. 4f). Similarly, CD4$^+$ T cells expressed lower ICOS and OX40 with a reduced CD44$^+$CD62L$^-$ T cell fraction (Fig. 4d and Supplementary Fig. 4f). Furthermore, both CD4$^+$ and CD8$^+$ T cells expressed decreased IFN-γ and TNF-α in the lungs of 4TO7-Lin28B mice (Fig. 4e), indicative of reduced Th1 differentiation. Accordingly, the CD4$^+$ and CD8$^+$ T cells downregulated T-bet expression in the context of Lin28B expression (Supplementary Fig. 4g), whereas GATA-3, FoxP3, and RoRγT expression in CD4$^+$ T cells were not affected (Supplementary Fig. 4h). Using a *PyMT* model, we confirmed that Lin28B expression inhibited the infiltration of CD4$^+$ and CD8$^+$ T cells in the pre-metastatic lung (Supplementary Fig. 5a). The decreased T cell populations expressed lower CD44 compared to the control (Supplementary Fig. 5b). Granzyme B expression in CD8$^+$ T cells was also significantly decreased (Supplementary Fig. 5c). Moreover, both CD4$^+$ and CD8$^+$ T cells downregulated IFN-γ, TNF-α, and T-bet expression upon Lin28B expression (Supplementary Fig. 5d–f). GATA-3, FoxP3, and RoRγT expression in CD4$^+$ T cells were not affected (Supplementary Fig. 5g). Furthermore, in the *MMTV-Neu* model, we verified that Lin28B expression was associated with decreased infiltration and activation of CD4$^+$ and CD8$^+$ T cell infiltration, and reduced Th1 differentiation of CD4$^+$ T cells in the pre-metastatic lung (Supplementary Fig. 5h–j). In addition, we noticed that the CD44$^-$CD62L$^+$ naïve subsets of CD4$^+$ and CD8$^+$ T cells were decreased in tumor-bearing mice compared with those in naïve mice (Fig. 4d), indicating their activation under tumor growth conditions. These data collectively suggested immune suppression of adaptive T lymphocytes in the pre-metastatic lung in the context of Lin28B expression.

To test whether N2 neutrophils are directly involved in CD4$^+$ and CD8$^+$ T cell activation, we activated spleen naïve CD4$^+$ and CD8$^+$ T cells in vitro using anti-CD3/CD28 antibodies alone or in the context of co-culture with pre-metastatic lung-derived neutrophils, and the results showed that N2 neutrophils markedly reduced CD44$^+$CD62L$^-$ populations in both CD4$^+$ and CD8$^+$ T cells (Fig. 4f). In contrast, the conditional media were not inhibitory (Fig. 4f).

Naïve CD4$^+$ T cells were incubated under Th1 conditions in the presence of N2 neutrophils in vitro, and then the cytokine phenotype was examined. As shown in Fig. 4g, N2 neutrophils inhibited the generation of the IFN-γ$^+$ subset of naïve CD4$^+$ T cells, whereas control neutrophils did not. Moreover, the supernatant could inhibit the Th1 differentiation of CD4$^+$ T cells (Fig. 4h), indicating it is in a contact-independent manner.

**Lung-infiltrating N2 neutrophils express high T cell exhaustion ligands**. The immune checkpoint-inhibitory ligands PD-L1 and PD-L2 of the B7 family are expressed on myeloid cell surfaces and are shown to be directly associated with immune suppression[25,26]. We set out to evaluate whether they are aberrantly expressed in lung N2 neutrophils. We discovered that lung neutrophils expressed higher PD-L2 levels in 4TO7-Lin28B mice (Fig. 5a). In contrast, PD-L1 was unchanged with Lin28B expression (Fig. 5a). Moreover, lung N2 neutrophils expressed a lower level of activating-ligand CD80 and unchanged levels of CD86, OX40L, and ICOSL (Fig. 5b). We noticed that exhaustion-ligand PD-L2 was slightly raised by Lin28B on lung macrophages, whereas DCs were hardly affected (Fig. 5c). The expression of PD-L1 in neither macrophages nor DCs was affected by Lin28B (Fig. 5c).

Administrating 4TO7-Lin28B mice with anti-IL-6 and anti-IL-10 neutralizing antibodies revealed that N2 neutrophil PD-L2 expression were significantly decreased and a similar decrease was not found in the control mice (Fig. 5d). An in vitro assay showed that IL-10 was a powerful inducer of PD-L2 mRNA in neutrophils, whereas IL-6 alone was ineffective (Fig. 5e). Interestingly, the combined treatment of IL-6 and IL-10 produced a stronger effect (Fig. 5e). N2 conversion of neutrophils results in increased immunosuppressive factor IL-4 expression, which has been shown to raise PD-L2 in macrophages[27,28]. We confirmed that IL-4 influenced PD-L2 production in neutrophils (Fig. 5e). PD-L1 was not similarly produced as PD-L2 upon the stimulations (Fig. 5e).

N2 neutrophils restricted CD4$^+$ and CD8$^+$ T cell proliferation (Fig. 5f) and the acquisition of a lower activated CD44$^+$/CD62L$^-$ phenotype (Fig. 5g). However, the N2 neutrophil-mediated suppression was reversed by the PD-L2 blockade (Fig. 5f, g). Cell

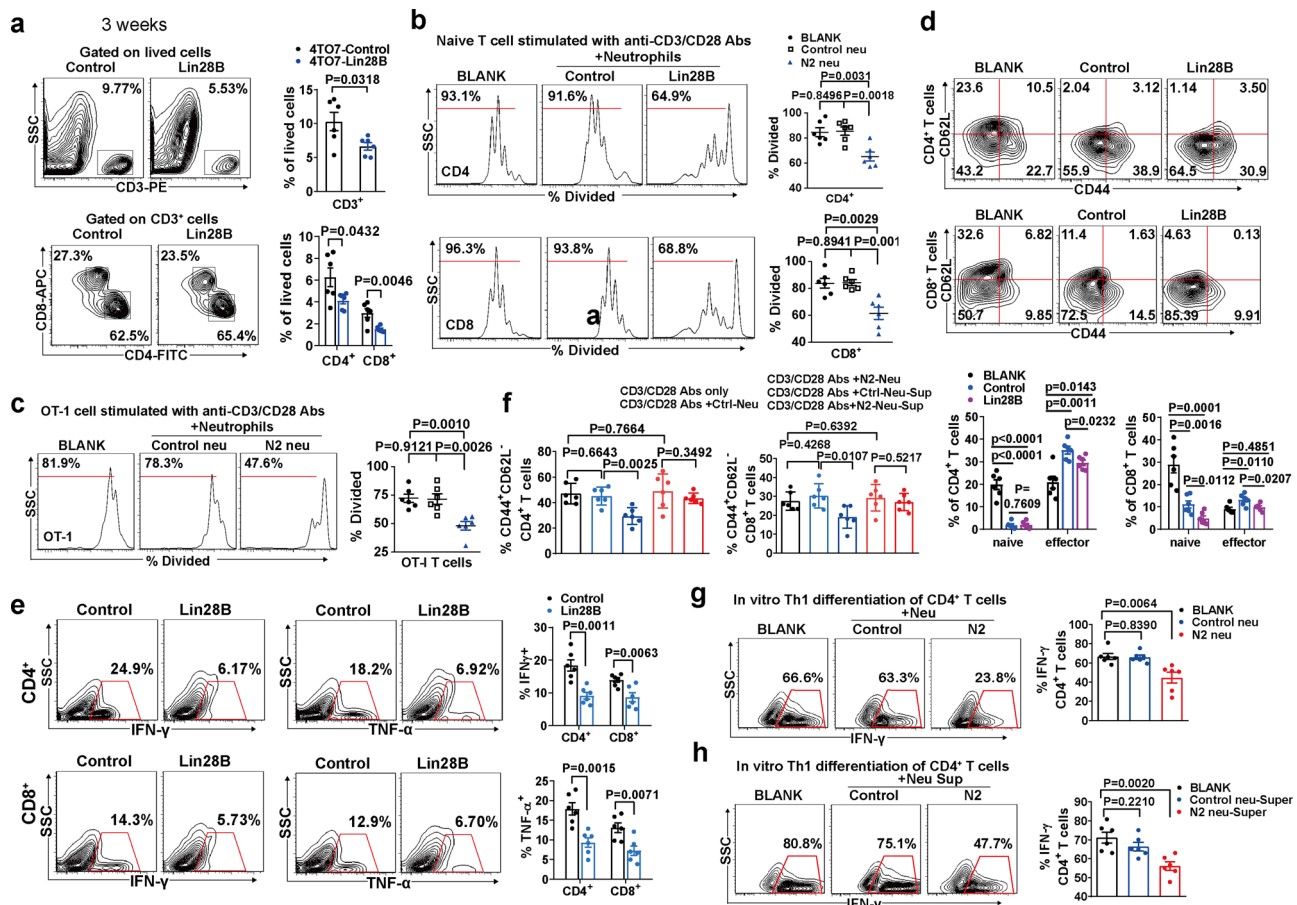

**Fig. 4 N2-converted neutrophils inhibit T cell proliferation, activation and Th1 differentiation. a** Representative flow cytometry plots and percentage of the indicated T cell subsets in the pre-metastatic lung ($n = 6$ mice in each group). **b** CFSE-labeled CD4$^+$ or CD8$^+$ splenocytes were stimulated with anti-CD3/CD28 antibodies and mixed without (BLANK) or with neutrophils purified from lungs of tumor-bearing mice. The percentage of proliferating T cells is shown ($n = 6$ culturing experiments). **c** CFSE-labeled OT-1 splenocytes (CD8$^+$) were mixed with OVA$_{257-264}$-pulsed bone marrow-derived DCs and neutrophils as in panel b. The percentage of proliferating T cells is shown ($n = 6$ culturing experiments). **d** CD4$^+$ and CD8$^+$ T cells in the pre-metastatic lung were subjected to CD44 and CD62L staining. Representative contour plots and summary data were shown ($n = 6$ mice in each group). **e** The expression of IFN-γ and TNF-α were analyzed in CD4$^+$ and CD8$^+$ T cells in the pre-metastatic lung ($n = 6$ mice in each group). **f** CD4$^+$ and CD8$^+$ T cells were purified from naïve mice's spleen stimulated with anti-CD3/CD28 antibodies and mixed with the indicated neutrophils. Twenty-four hours later, the cell fraction adopting a CD44$^+$ CD62L$^-$ phenotype was determined by flow cytometry ($n = 6$ culturing experiments). **g** Naïve CD4$^+$ T cells were stimulated under Th1-stimulating conditions in the presence of the indicated lung neutrophils. Four days later, the cells were re-stimulated with PMA and ionomycin for 6 h and Golgi-plug for 4 h. The expression of IFN-γ was analyzed by intracellular staining. Representative profile and percentage of IFN-γ$^+$ T cells were shown ($n = 6$ mice in each group). **h** The experiment was the same as for panel **g**, except that the neutrophils were replaced with the corresponding supernatant ($n = 6$ mice in each group). The lung cells (neutrophils, CD4$^+$ and CD8$^+$ T cells) were isolated from the pre-metastatic lung of tumor-bearing mice at 3 weeks after tumor inoculation. Data were presented as the mean ± SEM, and $p$ values were calculated using a two-tailed Student's $t$-test. *$p < 0.05$, **$p < 0.01$, ***$p < 0.001$, ns no significance. Source data are provided as a Source Data file.

contact was not required for N2 neutrophil-mediated inhibitory Th1 differentiation of CD4$^+$ T cells (Fig. 4h), suggesting that soluble factors are the dominant mechanism. IL-12a was the most important inducer of Th1 differentiation of CD4$^+$ T cells[29], and IL-6 was also reported to play a negative role in this process[30]. The N2-converted neutrophils produce BALFs with increased levels of IL-6 and IL-10, and a decreased content of IL-12a (Fig. 3b, c). We isolated spleen CD4$^+$ T cells from naïve mice and cultured them with neutrophil supernatant to survey the Th1 differentiation-related determinant. The supernatant from control neutrophils rose the generation of IFN-γ$^+$ CD4$^+$ T cells in an IL-12a-dependent manner (Fig. 5h). Moreover, the supernatant of N2 neutrophils was sufficient to inhibit the presence of IFN-γ$^+$ CD4$^+$ T cells, which was rescued by recombinant IL-12a, or the neutralization of IL-6 or IL-10 (Fig. 5h), implying that elevated IlL-6 or IL-10 overcomes the lower concentration of IL-12a to

attenuate Th1 differentiation of CD4$^+$ T cells. In contrast, IL-6 or IL-10 neutralization did not affect the Th1 differentiation of CD4$^+$ T cells induced by control neutrophil supernatants (Fig. 5h). Thus, accounting for the higher IL-12a levels unmasked the function of the lower content of IL-6 or IL-10 in driving Th1 differentiation of CD4$^+$ T cells. In addition, we observed that the PD-L2 blockade did not interfere with the Th1 differentiation of CD4$^+$ T cells within both supernatants (Fig. 5h). Therefore, N2-converted neutrophils exploit a unique avenue to tune up the ratio of immuno-stimulatory IL-12a to immunosuppressive IL-6 and IL-10 to control Th1 differentiation of CD4$^+$ T cells.

**Exosomes are crucial mediators of Lin28B-induced lung metastasis by creating immune-suppressive pre-metastatic niches.** To determine tumor-derived exosomes' role in the

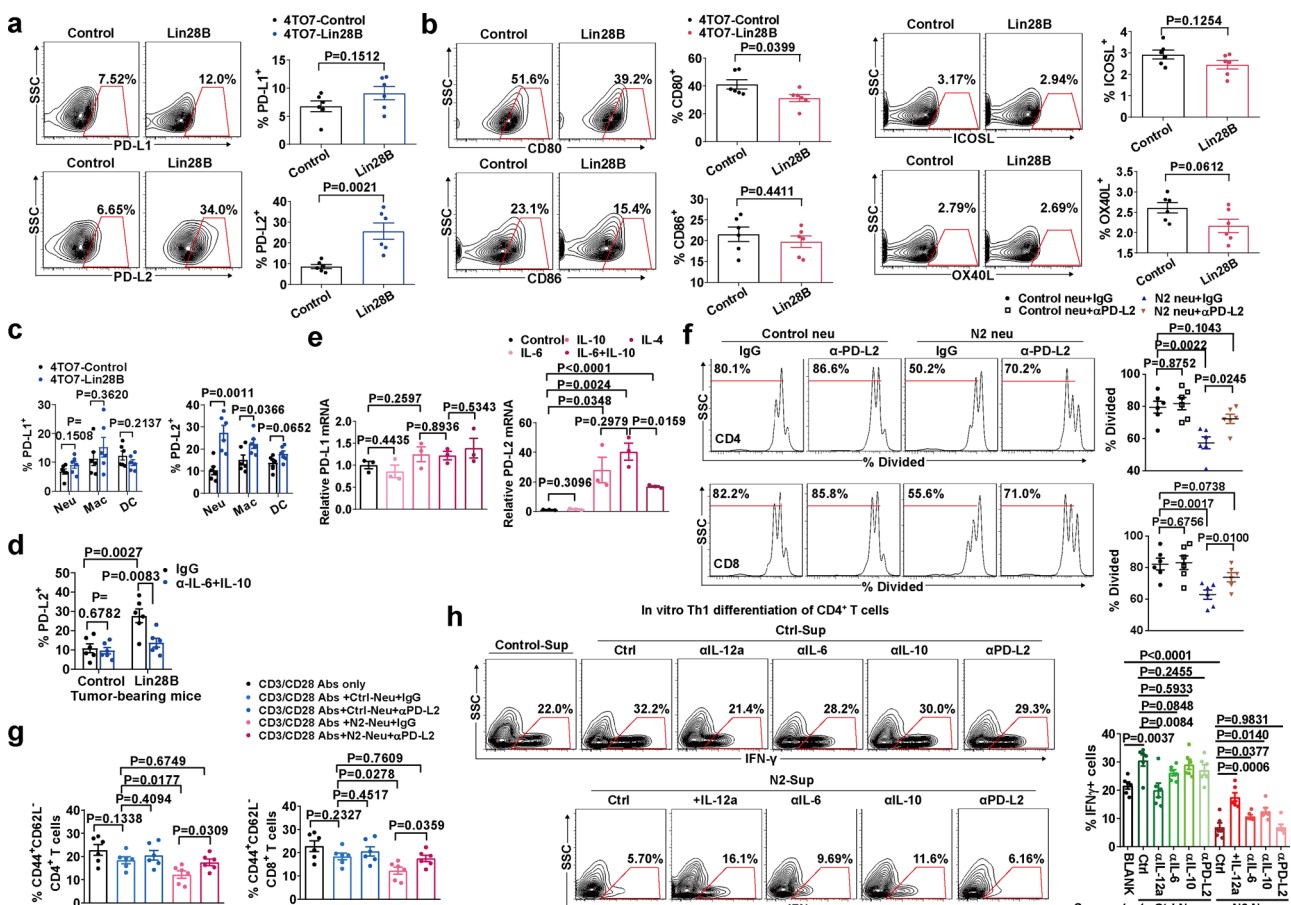

**Fig. 5 N2 neutrophils inhibit T cell activity primarily through PD-L2. a** PD-L1 and PD-L2 expression of neutrophils in the pre-metastatic lung ($n = 6$ mice in each group). **b** The experiment was the same as in panel a, except that CD80, CD86, OX40L, and ICOSL were measured. **c** Expression of PD-L1 and PD-L2 on the indicated cells in the pre-metastatic lung ($n = 6$ mice in each group). Neu neutrophil, Macro macrophage, DC dendritic cell. **d** IL-6 and IL-10 depletion decreased PD-L2 expression of lung neutrophils in 4TO7-Lin28B mice. The 4TO7-Lin28B or the control mice were intratracheally delivered with anti-IL-6 and anti-IL-10 neutralizing Abs or normal IgG as control once every 3 days starting 2 weeks after tumor inoculation. 1 week later, the lung neutrophils were evaluated for PD-L2 expression ($n = 6$ mice in each group). **e** Neutrophils were stimulated with the indicated cytokines, and the mRNA expression of PD-L1 and PD-L2 were determined. GAPDH served as an internal control ($n = 3$ mice in each group). **f, g** PD-L2 blocking attenuated N2 neutrophil-induced T cell proliferation and activation inhibition ($n = 6$ culturing experiments). **h** Naïve CD4$^+$ T cells were incubated with the neutrophil supernatants in the presence of indicated antibodies or cytokines. The expression of IFN-γ was analyzed by intracellular staining ($n = 6$ culturing experiments). The lung cells (neutrophils, macrophages, and DCs) were isolated from the pre-metastatic lung of tumor-bearing mice 3 weeks after tumor inoculation. Data were presented as the mean ± SEM, and $p$ values were calculated using a two-tailed Student's $t$-test. *$p < 0.05$, **$p < 0.01$, ***$p < 0.001$, ns no significance. Source data are provided as a Source Data file.

pre-metastatic stage, we purified serum exosomes and determined that tumor loading slightly increased the serum exosome content. Lin28B expression did not further increase serum exosome content (Supplementary Fig. 6a). Transmission electron microscopy showed exosomes displaying saucer-like morphology with an average diameter of ~100 nm (Fig. 6a, denoted by the blue arrow). The detection of exosome markers verified that the isolated particles were exosomes (Fig. 6b). Next, we checked whether exosomes were effectively captured by lung cells in our system. The results revealed approximately 3.82% of lung cells were positive for PKH-26 (Supplementary Fig. 6b). Exosomes were primarily incorporated by neutrophils, endothelial cells, DCs, macrophages, and fibroblasts (Supplementary Fig. 6b, c). In contrast to fibroblasts, the epithelial cells internalized less exosomes (Supplementary Fig. 6b, c).

To resolve the detailed components carried by the exosomes, we conducted high-throughput sequencing of small RNAs in 4TO7 tumor exosomes. The preliminary results showed that miRNAs accounted for more than 80% of the total small RNAs

carried by these exosomes (Fig. 6d), consistent with a previous study showing that miRNAs are the main content of exosomes[31]. In addition, snRNAs, snoRNAs, rRNAs, tRNAs, and mRNAs also were differentially included (Fig. 6d). A total of 622 miRNAs were detected in tumor exosomes, with 253 miRNA present with reads of more than 50 in number (Supplementary Data 1). Overall, miR-16, miR-125b, let-7 family members, miR-21a, miR-29a, and miR-92a were found to be the most abundant miRNAs (Supplementary Data 1). Together, these data indicated that miRNAs are the most abundant content of tumor exosomes

To determine if tumor-derived exosomes can induce pre-metastatic niches, we intravenously injected 4TO7 tumor-derived exosomes into naïve mice (Fig. 6c) and the results showed that in vivo administration of Lin28B-exosome induced higher expression of niche characteristic genes in the lung (Supplementary Fig. 6d). In addition, the levels of chemokines were markedly upregulated in BALFs, but not in serum (Supplementary Fig. 6e), and thus more neutrophils accumulated in the pre-metastatic lung (Fig. 6f). Moreover, 4TO7-Lin28B-exosome was powerful

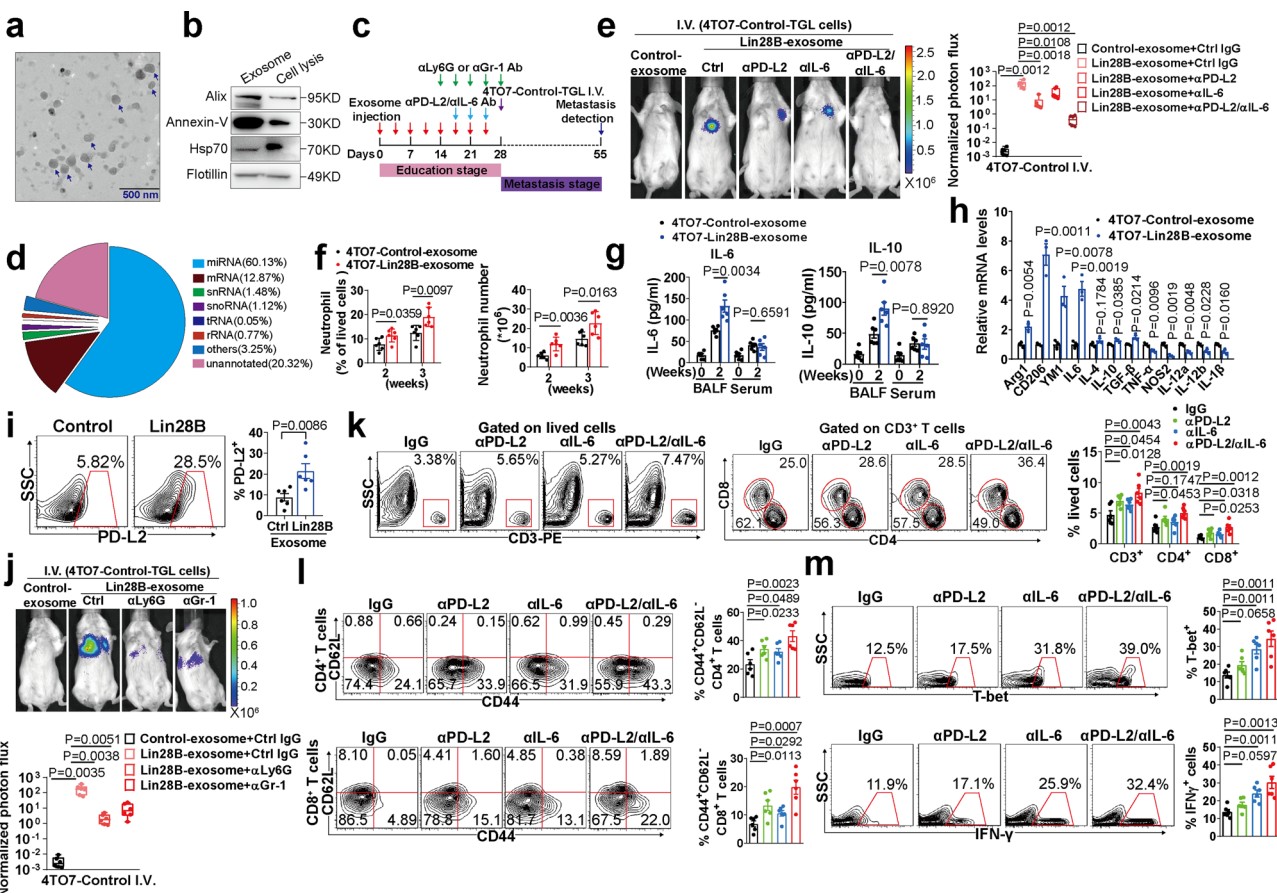

**Fig. 6 Exosomes are crucial mediators of Lin28B-induced lung metastasis by creating immune-suppressive pre-metastatic niches. a** Representative electron microscopy images of tumor exosomes. The images with saucer-like shapes were represented by a blue arrow. Scale bar: 500 nm. **b** IB analysis of exosomal proteins. The experiment was repeated three times independently with similar results; data from one representative experiment are shown. **c** Schematic diagram of exosome education and PD-L2, IL-6, Ly6G, or Gr-1 blocking. **d** Result summary of small RNA types and proportion from the high-throughput sequencing data of 4TO7 tumor exosomal RNAs. **e** The experiment was conducted as for panel **c**. Shown are images and bioluminescence imaging (BLI) quantification of lung metastasis ($n = 6$ mice in each group). **f** The experiment was conducted as for panel **c**. Shown are proportions and numbers of neutrophils in the lungs of mice treated with exosomes for 3 weeks ($n = 6$ mice in each group). **g** The experiment was the same as for panel **f**, except that the IL-6 and IL-10 were quantified in the BALFs of the treated mice ($n = 6$ mice in each group). **h** The experiment was the same as in panel **f**, except that the N1/N2 marker genes of lung neutrophils were quantified by qRT-PCR ($n = 3$ mice in each group). **i** Expression of PD-L2 on lung neutrophils of mice treated with exosomes as for panel **f** ($n = 6$ mice in each group). **j** The experiment was conducted as panel **c**. Shown are images and BLI quantification of lung metastasis ($n = 6$ mice in each group). **k**–**m** The experiment was conducted as panel **c**. T cells (**k**), the activated CD44$^+$CD62L$^-$ subset (**l**), and the Th1 subset of the CD4$^+$ T cells (**m**) were analyzed ($n = 6$ mice in each group). Data were presented as the mean ± SEM, and $p$ values were calculated using a two-tailed Student's $t$-test. For panels (**e**) and (**j**), boxes represent data within the 25th to 75th percentiles. Whiskers depict the range of all data points. Horizontal lines within boxes represent mean values. $*p < 0.05$, $**p < 0.01$, $***p < 0.001$, ns no significance. Source data are provided as a Source Data file.

enough to initiate IL-6 and IL-10 overproduction in BALFs (Fig. 6g) and thus shifted neutrophils into the N2 phenotype, resulting in increased PD-L2 expression (Fig. 6h, i), which, in turn, generated an immune-suppressive microenvironment (Supplementary Fig. 6f–h). In addition, when we recovered exosomes from *PyMT* mice, similar immune-suppressive effects also were found (Supplementary Fig. 6i). Notably, 4TO7-Lin28B-exosome markedly induced lung metastasis of less metastatic 4TO7-Control-TGL cells (Fig. 6e).

In addition to the lung, exosomes labeled by PKH-26 also could be captured by other organs, such as the liver, spleen, and bone marrow (Supplementary Fig. 6j). To corroborate the pro-metastatic effect of exosomes on tissues other than the lung, we performed intracardial (I.C.) injection of 4TO7-Control cells after systemic exosome education (Supplementary Fig. 6k). Metastasis was found mainly in the lung, but not in the bone or liver (Supplementary Fig. 6k). In addition, following exosome

injection, neutrophils of the N2 phenotype were found only in the lung (Fig. 6h), and not in the liver or bone marrow (Supplementary Fig. 6l). These results suggested that N2 conversion of neutrophils is the principle mechanism for exosome-induced lung metastasis.

To determine how neutrophils affect exosome-induced lung metastasis, we depleted neutrophils in mice by dosing with anti-Ly6G Ab (Fig. 6c). The results showed that neutrophil depletion obviously reduced the induction of lung metastasis by Lin28B-exosomes (Fig. 6j). In addition, neutrophil depletion restored the infiltration and activation of CD4$^+$/CD8$^+$ T cells, and the Th1 differentiation of CD4$^+$ T cells (Supplementary Fig. 6m–o), thus liberating the lung tissue from immune suppression. Given the lack of a genetic model for neutrophil depletion, we employed dosing with anti-Gr-1 Ab as an alternative method for the depletion of neutrophils. Since anti-Gr-1 Ab recognizes both Ly6C and Ly6G antigens, we characterized the CD11b$^+$Gr-1$^+$

population in the pre-metastatic lung with Abs specific to Ly6C and Ly6G. The majority of the CD11b$^+$Gr-1$^+$ cells were Ly6G$^+$Ly6C$^+$ cells (Supplementary Fig. 6p), indicating the anti-Gr-1 Ab primarily targets the lung neutrophils. The results showed that just as with the anti-Ly6G Ab, anti-Gr-1 Ab relieved the Lin28B-exosome induced immune suppression (Supplementary Fig. 6m–o) and diminished the lung metastasis (Fig. 6j), indicating neutrophils are essential for the exosome-induced lung metastasis.

To identify the immune-suppressive pre-metastatic niche's role in the Lin28B-promoted metastatic process, we serially blocked PD-L2 or IL-6 with neutralizing antibodies in Lin28B-exosome treated mice (Fig. 6c). Either blockage of PD-L2 or IL-6 exhibited a partial effect in attenuating immune suppression and reducing lung metastasis incidence of 4TO7-Control-TGL cells (Fig. 6e, k–m). The combined blocking was more beneficial in relieving T cell function and reducing the tumor cells' lung metastasis burden (Fig. 6e, k–m). We concluded that tumor-derived exosomes recapitulate the Lin28B-induced immune-suppressive pre-metastatic niche, which can be counteracted by PD-L2 and IL-6 elimination.

**Lung fibroblasts, neutrophils, and macrophages are the primary recipient cells responsible for increased CXCLs, IL-6, and IL-10 levels in the pre-metastatic lung under Lin28B expression conditions.** In the pre-metastatic lung of the 4TO7-Control mice, CXCLs were detected mostly in stromal cells (Supplementary Fig. 7a). However, the stromal cells showed a distinct response to Lin28B expression. Lin28B increased CXCL production in fibroblasts compared to the control (Supplementary Fig. 7b, c). In contrast, epithelial cells only slightly upregulated chemokines by Lin28B expression (Supplementary Fig. 7b, c). In addition, Lin28B-exosome was competent to render fibroblasts capable of producing increased CXCLs relative to epithelial cells (Supplementary Fig. 7d). IL-6 was abundantly expressed in neutrophils and macrophages (Supplementary Fig. 7e, f). Lin28B expression also enhanced IL-6 production in both cell types (Supplementary Fig. 7e, f). The neutrophil cell numbers were far higher than those of macrophages in the pre-metastatic lung (Supplementary Fig. 7h), suggesting a more important contribution of neutrophils in IL-6 production. In contrast, IL-10 was mostly expressed in macrophages (Supplementary Fig. 7e, g). Lin28B raised IL-10 mean fluorescence intensity (MFI) but did not significantly affect the percentage of IL-10$^+$ macrophages (Supplementary Fig. 7e, g). Hence, we concluded that lung fibroblasts, neutrophils, and macrophages are the major contributors to the increased production of chemokine, IL-6, and IL-10 in the context of Lin28B expression.

**Decreased exosomal let-7s are responsible for immunosuppressive niche establishment.** MiRNAs are the main content of exosomes[31]. Let-7s are a regulatory target of Lin28B and display an extensive inhibitory role in tumor progression[3,32], we, therefore, set out to focus on let-7s' role in pre-metastatic niche modification. Let-7s were dropped in 4TO7-Lin28B cells (Supplementary Fig. 8a). Moreover, exosomes harvested from 4TO7-Lin28B tumor ablated let-7s (Supplementary Fig. 8b). Importantly, there was a lower content of let-7s embedded in the serum exosomes (Fig. 7a). In contrast, the miRNAs, which were not targeted by Lin28B, remained unchanged (Fig. 7a and Supplementary Fig. 8a, b). A previous study reported that *miR-125a/b* is a regulatory target of Lin28B[33]. Consistently, we detected *miR-125a/b* repression by Lin28B, although the extent was smaller than that of let-7s (Fig. 7a and Supplementary Fig. 8a, b). Using the *PyMT* and *MMTV-Neu* models, we confirmed that Lin28B

expression inhibited let-7s and *miR-125a/b* contents embedded in the serum exosomes (Fig. 7b and Supplementary Fig. 8c). Therefore, we proposed that Lin28B impacts the pre-metastatic niche via decreased let-7s or *miR-125a/b* exosomal content.

Here, we selected let-7a as the representative let-7 member in most experiments. We injected naïve mice intravenously with Lin28B-exosome in the presence of agomir-let-7a or agomir-miR-125a. Agomir-let-7a supplementation repressed the characteristic lung niche gene expression (Supplementary Fig. 8d) and restrained CXCLs, IL-6, and IL-10 induction mediated by Lin28B-exosome, while agomir-*miR-125a* did not (Fig. 7c, d and Supplementary Fig. 8d, e). In turn, agomir-let-7a showed neutrophil inhibition (Fig. 7e and Supplementary Fig. 8f) and a lower increase in PD-L2 expression (Supplementary Fig. 8h), whereas agomir- *miR-125a* showed negligible effects (Supplementary Fig. 8f–h). Accordingly, agomir-let-7a relieved Lin28B-exosome-induced immune suppression (Fig. 7f, g and Supplementary Fig. 8i) and decreased lung metastasis of 4TO7-Control-TGL cells prompted by Lin28B-exosome (Fig. 7h). In summary, our data suggested that the deficient content of let-7s in Lin28B-exosomes is responsible for immunosuppression.

It was interesting to find that the ALDH$^+$ 4TO7-Lin28B cells express increased Lin28B (Supplementary Fig. 8j). Thus, exosomes harvested from the ALDH$^+$ subpopulation of 4TO7-Lin28B tumors encompassed much lower let-7s levels (Fig. 7i). Importantly, exosomes from the ALDH$^+$ subsets significantly promoted tumor cell metastasis compare to total tumor cell exosomes, and this function was ablated by let-7a replenishment (Fig. 7j). In contrast, exosomes from the ALDH$^-$ subsets only slightly increased metastatic incidence, which was smaller than that exerted by total tumor cell exosomes (Fig. 7j).

To evaluate whether decreased let-7s in exosomes were directly responsible for increased CXCLs, IL-6, and IL-10 production in the recipient cells, exosomes were added to the cell culture to mimic in vivo education. Lin28B-exosome heightened CXCLs production in 3T3 fibroblasts (Supplementary Fig. 9a) and let-7a mimics inhibited CXCLs upregulation (Supplementary Fig. 9a). By online RNA hybridization at BiBiServ (Bielefeld Bioinformatics Server)[34], we did not find let-7a binding sites in the 3′UTR of *CXCLs* mRNA. Lin28B-exosome upregulated STAT3 phosphorylation and this effect was relieved by let-7a mimics (Supplementary Fig. 9b). Importantly, the STAT3 pathway inhibitor S3I-201[35] was found to blunt the increased CXCLs induced by Lin28B-exosome (Supplementary Fig. 9c), indicating that deficient let-7s in Lin28B-exosome enhances STAT3 pathway activity to increase CXCLs production. Literatures indicated that IL-6 is a known let-7s target gene[36]. Indeed, by using bone marrow-derived neutrophils, we discovered that Lin28B-exosome augmented IL-6 generation and let-7a mimics restrained IL-6 upregulation (Supplementary Fig. 9d). Moreover, using bone marrow-derived macrophages (BMDMs), we determined that Lin28B-exosome increased IL-10 content, a trend that was diminished by let-7a (Supplementary Fig. 9e). Through online RNA hybridization at BiBiServ, we found a species-conserved let-7a binding site in the 3′UTRs of *IL-10* genes (Supplementary Fig. 9f). FL reporter experiments showed that IL-10 was a direct target of let-7a (Supplementary Fig. 9g, h). We noticed that Lin28B-exosome did not increase IL-6 and IL-10 expression compared with the control in 3T3 cells (Supplementary Fig. 9i). In addition, S3I-201 did not inhibit IL-6 and IL-10 production (Supplementary Fig. 9i), indicating that low let-7s in exosomes and the STAT3 pathway have no influence upon IL-6 and IL-10 upregulation in 3T3 cells. Moreover, we found that IL-6 or IL-10 stimulation all decreased *IL-6* mRNA expression in neutrophils (Supplementary Fig. 9j). In contrast, the control CCL2 did not affect (Supplementary Fig. 9j). This result indicated that *IL-6*

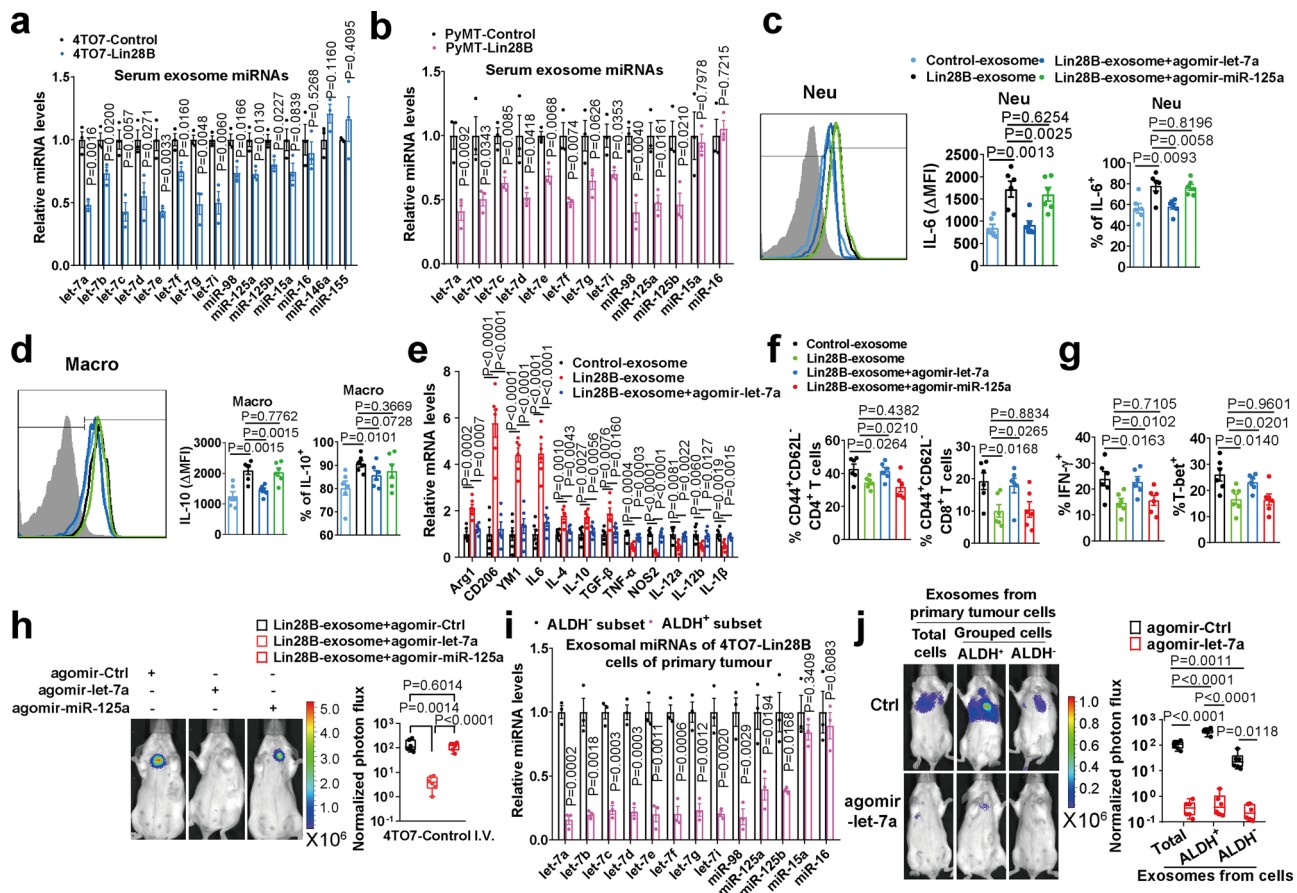

**Fig. 7 Decreased exosomal let-7s are responsible for immunosuppressive niche establishment. a** Let-7s and miR-125s family members were specially reduced in the circulation exosomes in mice bearing 4TO7-Lin28B tumor ($n = 3$ mice in each group). **b** The experiment was the same as in panel **a**, except that let-7s and miR-125s family members were measured in the serum exosomes of *PyMT* mice (11-week old) ($n = 3$ mice in each group). **c** Let-7a inhibited increased production of IL-6 in neutrophils mediated by Lin28B-exosome. Naïve mice were i.v injected with Control-exosome or Lin28B-exosome in the presence of the indicated agomir. IL-6 expression was determined in lung neutrophils by FACS ($n = 6$ mice in each group). **d** The experiment was the same as for panel **c**, except that the IL-10 was detected in macrophages ($n = 6$ mice in each group). **e** Let-7a inhibited neutrophil N2 conversion induced by Lin28B-exosomes. The exosome treatment was the same as for panel c. GAPDH served as an internal control ($n = 6$ mice in each group). **f** Let-7a rescued T cell inactivation induced by Lin28B-exosome ($n = 6$ mice in each group). **g** Let-7a restored Lin28B-exosome-mediated inhibitory Th1 differentiation of CD4[+] T cells ($n = 6$ mice in each group). **h** Let-7a attenuated Lin28B-exosome-mediated lung metastasis ($n = 6$ mice in each group). **i** ALDH[+] and ALDH[−] subsets of 4TO7-Lin28B cells isolated from primary tumors were analyzed for exosomal miRNA expression. U6 served as an internal control ($n = 3$ mice in each group). **j** The ALDH[+] and ALDH[−] subset exosomes were collected as for panel **i**, and the exosome education and the I.V. injection of tumor cells were the same as in Fig. 6c. Shown are images and BLI quantification of lung metastasis of 4TO7-Control cells ($n = 6$ mice in each group). Data were presented as the mean ± SEM, and p values were calculated using a two-tailed Student's *t*-test. For panels (**h**) and (**j**), boxes represent data within the 25th to 75th percentiles. Whiskers depict the range of all data points. Horizontal lines within boxes represent mean values. *$p < 0.05$, **$p < 0.01$, ***$p < 0.001$, ns no significance. Source data are provided as a Source Data file.

expression was negatively regulated by the IL-6 and IL-10 pathways in neutrophils, ensuring that the intracellular IL-6 levels do not become too high. This result also suggests that low let-7s in Lin28B-exosomes, but not the IL-6 or IL-10, may be the principle mechanism for increased IL-6 production in neutrophils (Supplementary Fig. 9d and j). Consistently, let-7a inhibition rescued decreased IL-6 and IL-10 production caused by Lin28B knockdown in MDA-MB-231 cells (Supplementary Fig. 9k), supporting that let-7s play an important role in regulating IL-6 and IL-10 expression in the cells. Together, these results suggested that low let-7s in Lin28B-exosome contributed to elevated CXCLs, IL-6, and IL-10 production in the target cells.

**Serum exosomal let-7s correlate with Lin28B expression of primary tumors and are potential indicators for predicting breast cancer patient prognosis and lung metastasis.** To extend our findings to human breast tumor patients, we analyzed clinical

serum exosomes collected from breast cancer patients and healthy donors (30 healthy donors and 30 breast cancer patients, Supplementary Data 1), respectively. Tumor loading did not change serum exosome secretion (Fig. 8a). However, let-7s were lower in breast cancer patients compared to healthy controls (Fig. 8b). In contrast, *miR-15* and U6 were unaffected (Fig. 8c). To determine the clinical implication of Lin28B expression with serum exosomal let-7s levels, we expanded our analysis to 91 clinical samples (Supplementary Data 1). The results showed the expression of Lin28B in primary tumors was inversely correlated with serum exosomal let-7a (Fig. 8d). Accordingly, serum exosomal let-7a was lowest in TNBC, relative to the luminal or HER2[+] subtypes (Fig. 8e). Due to a lack of clinical sample availability, we could not directly evaluate the relationship between serum exosomal let-7a and breast cancer patient prognostic. However, it was found that the let-7a detected in primary tumors was well proportional to their amount in serum exosomes (Fig. 8f), hinting their similar

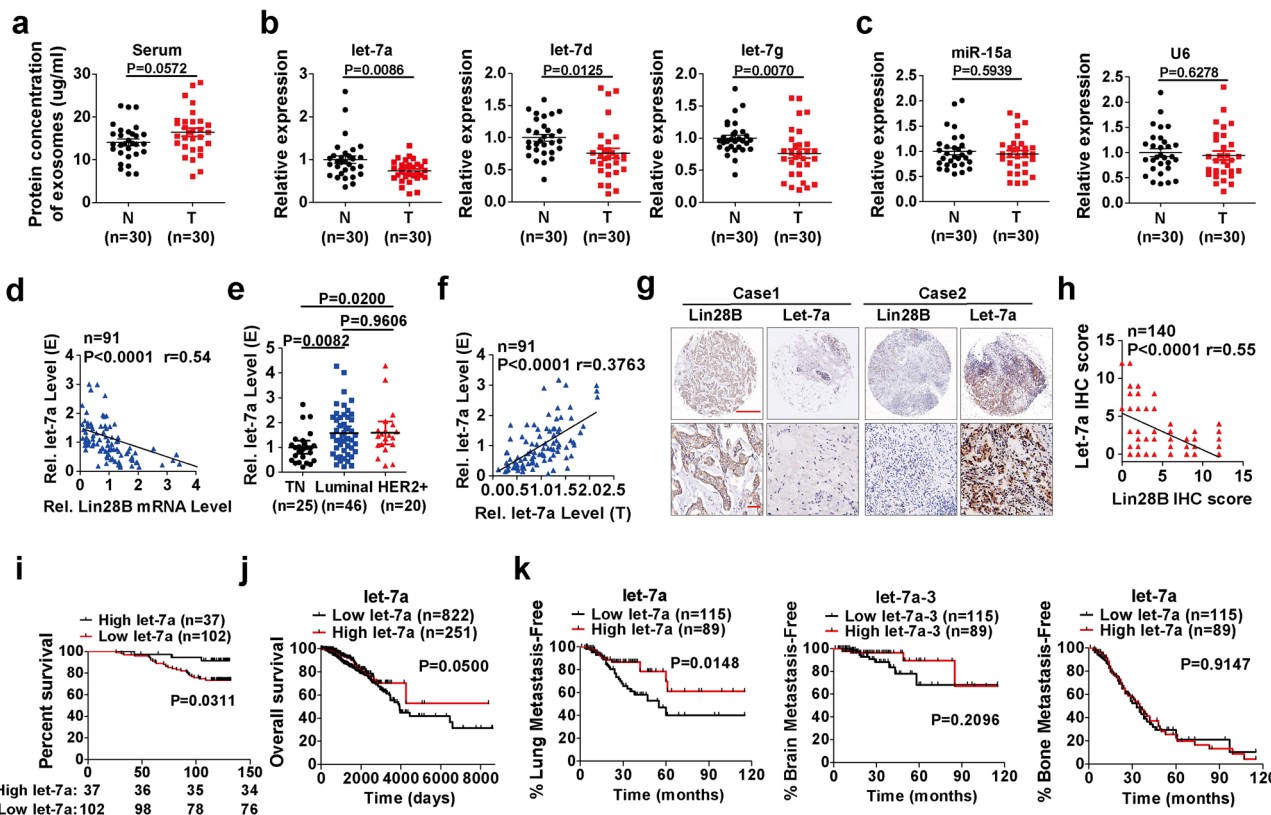

**Fig. 8 Serum exosomal let-7s correlate with Lin28B expression of primary tumors and are potential indicators for predicting breast cancer patient prognosis. a** Exosome protein concentrations purified from sera of healthy donors and primary breast cancer patients ($n = 30$ patients). **b** Let-7a, let-7d, and let-7g were quantified by qRT-PCR in exosomes prepared as in panel **a**. MiR-16 served as an internal control ($n = 30$ patients). **c** Mir-15a and U6 expression in exosomes prepared as in panel **a**. MiR-16 served as an internal control ($n = 30$ patients). **d** Pearson correlation between the relative *Lin28B* mRNA level of primary tumors and the relative let-7a amount of serum exosomes in the same breast cancer patient ($n = 91$ patients). **e** Relative expression of let-7a in serum exosomes in different breast cancer subtypes ($n = 91$ patients). **f** Pearson correlation between the relative let-7a level of primary tumors and the relative let-7a amount in serum exosomes in the same breast cancer patient ($n = 91$ patients). **g** Representative images of IHC staining of Lin28B along with in situ hybridization of let-7a on TMA Cohort II (140 cases). Bars: 500 μm (upper panels) and 50 μm (lower panels). **h** Correlation between the expression of Lin28B and let-7a as in panel **g** was assessed via linear regression ($n = 140$ patients). **i** Overall survival probabilities were calculated using the Kaplan–Meier method and analyzed using a log-rank test in a retrospective cohort of patients (139 cases) with breast cancer (Cohort II) with low ($n = 102$ patients) or high ($n = 37$ patients) expression of let-7a. **j** Overall survival probabilities were calculated using the Kaplan–Meier method and analyzed using a log-rank test in a cohort of GDC TCGA breast cancer patients (1073 cases), by let-7a expression. **k** Metastasis-free survival probabilities were calculated by let-7a expression using the Kaplan–Meier method and analyzed using a log-rank test in a clinical cohort of 204 breast cancer patients from GEO dataset GSE12276 with let-7a expression. In **a–c** and **e**, data were presented as the mean ± SEM, and p values were calculated using a two-tailed Student's t-test. *p < 0.05, **p < 0.01, ns no significance. Source data are provided as a Source Data file.

correlation with patient prognostic. We performed in situ hybridization of let-7a on TMA Cohort II with tumors from a retrospective cohort of breast cancer patients (Fig. 1e). As expected, Lin28B expression was inversely associated with let-7a levels (Fig. 8g, h). Accordingly, an overall reduced survival was found in breast cancer patients with lower let-7a in tumor tissue (Fig. 8i).

To further determine the clinical significance of let-7a expression, we analyzed the transcriptomic datasets of two clinical cohorts. First, the RNA-sequencing data of the Genomic Data Commons (GDC) The Cancer Genome Atlas (TCGA) breast cancer clinical cohort (https://xenabrowser.net, 1073 breast cancer patients) indicated that the lower expression of let-7a was associated with reduced overall survival of breast cancer patients (Fig. 8j). A similar correlation also was observed for let-7b, let-7c, and let-7d (Supplementary Fig. 9l). In addition, using a clinical cohort of 204 breast cancer patients from GEO dataset GSE12276, we determined that lower let-7a expression in breast tumors correlated with lung metastasis, but not with brain or bone

metastasis (Supplementary Table 7). Moreover, the lower let-7a expression also predicted reduced lung metastasis-free survival (Fig. 8k), implying the existence of a similar mechanism whereby Lin28B promotes human breast cancer progression via lower exosomal let-7s. At present, we do not know whether N2 neutrophils also appear in the pre-metastatic lung in patients with breast tumor. Due to the direct action of lower exosomal let-7s upon increased CXCLs, IL-6, and IL-10 secretion in BALFs (Fig. 7c, d and Supplementary Fig. 8e), we believe that the conversion of N2 neutrophils also occurs in the pre-metastatic lung in patients with breast tumor with lower exosomal let-7s. Together, our data strongly indicated a closer association of lower exosomal let-7s with poor prognosis and lung metastasis of patients with breast cancer.

## Discussion
While Lin28B is recognized as a breast cancer metastasis promoter, little is known about its detailed mechanism. Here, we

determine that Lin28B can establish immune suppression pre-metastatic niches to promote breast cancer lung metastasis. We show that Lin28B expression enables pre-metastatic lung inducing cytokine milieus that involve in neutrophil recruitment and N2 conversion, in which neutrophil N2 conversion is a key step for T cell function inhibition. Thus, our results indicate Lin28B expression in the primary tumor can alter the "immune state in the metastatic niche" (Supplementary Fig. 10). Moreover, we verify that exosomes with low let-7s are the prerequisite for Lin28B-induced immune suppression. Thus, our results reveal a direct functional link between Lin28B and immunosuppression, suggesting a metastasis-related mechanism in the early stage of tumor growth.

Phenotype conversion is an important mechanism by which neutrophils work as a primary tumor-promoter or suppressor[20]. Here, we discovered that neutrophil phenotypic conversion was also recapitulated in the pre-metastatic niche. Importantly, N2 neutrophils were proven to promote metastasis. In addition, we identified IL-6 and IL-10 as the inducers implicated in phenotypic modification of neutrophils. As the circulating neutrophils in the tumor-bearing mice did not show the N2 phenotype, we concluded that neutrophils' phenotypic conversion is context-dependent and that it can be influenced by diverse factors deriving from the surrounding cytokine milieu. Therefore, even without tumor cells arriving, Lin28B enables the pre-metastatic niche to produce IL-6 and IL-10, which is enough to induce N2 conversion. Accordingly, interfering with neutrophils' N2 conversion may be an accessible way to inhibit Lin28B-mediated breast cancer lung metastasis.

Neutrophils can affect tumor progression by immune regulatory mechanisms. In contrast to the antitumor immune response[37], increasing reports indicate that they have opposite immune-suppressive functions[38,39]. Here, we identified that N2-converted neutrophils in the pre-metastatic niche can repress T cell function. PD-L2 was identified to be responsible for N2 neutrophils-induced T cell inhibition. Molecular immune checkpoints are a potential mechanism for neutrophil-mediated immune regulation. Tumor neutrophils were activated by GM-CSF[40] or IL-6[41] to induce PD-L1 and in turn, acquire the capability to repress T cell activity. In addition to IL-4, we here identified IL-6 and IL-10 as additional factors to initiate neutrophils boosting PD-L2. We did not observe a similar upregulation of PD-L1 in N2 neutrophils. IFN-γ was reported to be a powerful inducer of PD-L1 expression[42]. We proposed that the decreased IFN-γ+ Th1 subset of CD4+ T cells may counteract PD-L1 production exerted by increased IL-6 in 4TO7-Lin28B mice. PD-L2 was also upregulated in macrophages. Although neutrophils are far more abundant than macrophages, we could not exclude any specific roles for macrophages in T cell immune in the pre-metastatic niche, and further investigation is needed to address this subject.

Helper T lymphocytes are in a core position concerning tumor immunity because the Th1 subset is particularly important in the activation of cytotoxic T lymphocytes (CTLs)[43]. Therefore, a deficiency in the Th1 population will lead to insufficient CD8+ CTL activation and immune suppression. IL-12a is a central factor for T-bet induction and Th1 differentiation[29], and IL-6 was also reported to influence this process negatively[30]. Here, we determined that N2 conversion results in reduced IL-12a levels. Thus, a deficiency of IL-12a in N2 neutrophils does not support the differentiation of CD4+ T cells toward Th1. Also, due to the lower IL-12a, the increased IL-6 and IL-10 may acquire a chance to compete with IL-12a and exhibit an inhibitory role in the Th1 differentiation of CD4+ T cells. Thus, lower IL-12a levels may be a driving factor for the attenuated Th1 differentiation and elevated IL-6 and

IL-10, further solidifying this process under Lin28B expression conditions. In contrast, PD-L2 blocking did not restore the Th1 differentiation of CD4+ T cells that were restrained by N2 neutrophils. Therefore, an altered cytokine milieu is a mechanism by which N2 neutrophils affect the differentiation of CD4+ T cells into the Th1 subset.

In contrast to tumor-free mice, CD4+/CD8+ T cells in the pre-metastatic lung are deficient in the CD44−CD62L+ naïve T cell subset, indicating that they are activated. Although tumor cells have not yet arrived at the lung, the dropped cell debris may carry specific antigen messages. For example, exosomes carry tumor antigens that stir up distal immune responses[44]. It is possible that lung DCs can engulf circulating exosomes, prompting DCs to capture antigens via exosomes. Thus, we suggest that the lung tissue is equipped with intrinsic machinery such as DCs to uptake tumor-shed antigen materials, migrate to the corresponding lymph node, and prime T cell activation. Primary tumors should be the largest antigen reservoir, where antigen-presenting cells (APCs) process antigens and induce immune activation in the local lymph node. The primed activated T cells then return to the tumor site, recognize the antigen-specific tumor cells, and induce immune responses. Most tumor antigens are neo-epitopes, which are solely created by tumor-specific DNA alterations that result in the formation of novel protein sequences[45]. A study showed that tumor-infiltrating lymphocytes (TILs) are phenotypically diverse, and not all tumor-infiltrating T cells are specific for tumor antigens. Recently, bystander CD8+ TILs were identified in human colorectal and lung tumors due to their phenotype overlap with tumor-specific CD8+ T cells but lack of CD39, suggesting that human CD8+ TILs also contain a population that recognizes cancer-unrelated epitopes[46]. As T cell priming was largely conducted in the peripheral lymphoid organs, we assumed that tumor antigen-specific T cells get to the lung through the circulatory system after priming, together with bystander T cells. When antigen-specific T cells meet tumor cells, the immune microenvironment centered on N2 neutrophils protects tumor cells from immune attack, or the tumor cells are recognized and killed. Therefore, N2 neutrophils might act to prepare the immune deficiency circumstances precluding tumor cell arrival. Simultaneously, the tumor antigen-specific T cells may also be influenced by the bystander T cells. So far, little is known about the immune organization in the pre-metastatic lung, which is an interesting and promising field.

Ovarian cancer cells with high Lin28A expression are capable of inducing the epithelial-mesenchymal transition (EMT), invasion, and migration of HEK 293 cells via exosomes[47]. In the present work, we unmasked an additional function whereby Lin28B affects the pre-metastatic niche via exosomes. Our clinical data show that higher Lin28B expression in breast tumors is correlated with lower exosomal let-7s. Importantly, the lowered exosomal let-7s were further associated with breast cancer patients' poor prognostic, suggesting that Lin28B mediates tumor progression via lower exosomal let-7s. The literature shows that exosomes are secreted by nearly all cell types[48]. Reports have shown that tumor tissue secretes exosomes actively[31]. However, our results find that tumor growth does not significantly affect the serum exosome load, as seen in the mouse breast cancer model and clinical human serum samples. Instead, tumor growth can change specific exosomal contents. It could be that tumor-derived exosomes are more stable than those from the normal cells and gradually become the dominant subset. Alternatively, tumor-derived exosomes may have a feedback effect to repress exosome release from other cells. It is also possible that tumor-derived exosomes have a better affinity for recipient cells at the target sites, thus acquiring more opportunities to enter recipient cells. Hence, tumor-derived exosomes may be the dominant group

impacting distal target cells; however, the underlying mechanism remains unclear.

We note that decreases in the levels of circulating let-7s also have been found in other diseases. For example, in chronic hepatitis C (CHC) patients, reduced levels of let-7s in plasma or extracellular vesicles (EVs) have been shown to correlate with advanced histological hepatic fibrosis stage, and let-7s levels in plasma declined markedly over time in parallel with fibrosis progression[49], indicating the potential role of reduced circulating let-7s in promoting the progression of a distinct disease. Therefore, we expect that there will be great interest in dissecting why exosomal let-7s levels are decreased in certain diseases and how this change promotes disease progression. Altogether, our study elucidated a mechanism that Lin28B promotes metastasis by influencing tumor exosomal let-7s content.

A previous study showed that lung epithelial cells were the main cells responsible for tumor exosome absorption, CXCL secretion, and neutrophil recruitment[12]. In contrast, our data revealed that fibroblasts were the main cells responsible for exosome receiving, thus permitting increased CXCL induction in the context of Lin28B expression. The main difference between our data and that of Liu et al. was the exosome-receiving cells. The differences presumably reflect the use of distinct tumor models. Liu et al. performed subcutaneous injection with Lewis lung carcinoma (LLC) cells, an epithelium-like mouse cell line, while we performed orthotopic mammary implantation with mesenchymal-like 4TO7 cells. We propose that fibroblasts may be more sensitive to 4TO7 cell-derived exosomes. For example, mesenchyme-like 4TO7 cell-derived exosomes may carry specific components that can be recognized more effectively by lung fibroblasts than by epithelial cells. Moreover, it also is possible that lung fibroblasts may be activated specifically by 4TO7 tumor-derived soluble factors before exosome arrival, thus potentiating their ability to receive exosomes. In turn, the different models may lead to fibroblast and epithelial cells with different efficiencies to receive exosomes, thus leading to distinct responses. These differences constitute an interesting phenomenon; further validation by other tumor models will be needed.

Lin28B does not significantly promote primary tumor growth but it can increase the ALDH$^+$ BCSCs fraction. High stem cell populations are not always coupled with high cell proliferation. Perhaps, Lin28B can also promote breast tumor cell EMT transition like Lin28A[50], a process accompanied by low proliferation. A higher BCSC subset may improve tumor cell survival upon arrival at the metastatic site, but it is unknown how this population affects the early metastasis stage. Here, we demonstrated that BCSCs are the prominent origin of exosomes containing much lower let-7s, highlighting its particularly important role in constructing the pre-metastatic niche. We observed a clear correlation between the stem cell population and the downregulated exosomal let-7s. Higher BCSCs characteristics may predict the potential for lower serum exosomal let-7s, immunosuppressive pre-metastatic niche, and a higher metastatic incidence. Therefore, with Lin28B expression, a breast cancer cell subset with BCSCs properties is vital to drive immune suppression and predicts poor prognosis. At the same time, we cannot exclude the possibility that the increased stem characteristics induced by Lin28B are an important mechanism for enhanced metastatic activity when the tumor cells arrive in lung tissues.

In summary, we explored a breakthrough function of Lin28B in the management of the pre-metastatic niche by decreasing immune surveillance, which is an extremely important pre-requisite for the survival and rapid growth of incoming cancer cells. This is of great significance by providing potential targets for the prevention and treatment of breast cancer metastasis.

## Methods

**Human samples.** This study was approved by the Institutional Research Ethics Committee of the Shanghai Institute of Nutrition and Health Sciences, Chinese Academy of Sciences. Five validated cohorts of patients with informed written consent were used in the study. The first cohort (Cohort I) consisted of 30 cases, from whom the tumors and matched adjacent tissues were obtained as tissue microarray (TMA) chips from the Biobank Center of National Engineering Center for Biochip at Shanghai (also known as Shanghai Outdo Biotech Company, Ltd) and used for IHC analysis. The clinicopathologic information of the patients were prospectively collected and provided in Supplementary Data 1. The second cohort (Cohort II) consisted of 140 cases from whom specimens were also obtained as TMA chips from the Biobank Center of National Engineering Center for Biochip and used for correlation analysis and Kaplan–Meier survival analysis. This group of 140 patients with a first diagnosis of breast cancer received surgical resection from January 2005 until September 2012. Follow-up started at the date of diagnosis and ended at death or on Aug 11, 2016. The clinicopathologic and follow-up data of patients were prospectively collected and provided in Supplementary Data 1. The third cohort (Cohort III) was generally consistent with the second cohort, except that the third cohort consisted of 145 cases. For the fourth cohort, the fresh tumor specimens and serum samples from a total of 91 breast cancer patients who were diagnosed with breast cancer and had undergone surgery from October 2019 to May 2020 were obtained from Tongji Hospital of Huazhong University of Science and Technology with written informed consent from all subjects. Similarly, a total of 30 serum samples from cancer-free female subjects were procured from Tongji Hospital of Huazhong University of Science and Technology after obtaining written informed consent from all subjects. The tissue specimens and serum exosome samples were stored at −80 °C until RNA extraction. Information for all 91 patients information with confirmed ER, PR, and HER2 expressions were summarized in Supplementary Data 1. For the fifth cohort, the fresh tumor and para-tumor specimens from a total of 24 TNBC patients who were diagnosed with TNBC and undergone surgery from March 2021 to May 2021 were obtained from Tongji Hospital of Huazhong University of Science and Technology with written informed consent from all subjects. The samples were used for western blotting. The patients involved in all the Cohorts were not part of any clinical trials and all the samples were collected as a standard of care.

**TMA immunohistochemistry.** Immunohistochemistry of the TMA chips was performed with a primary antibody against Lin28B (Cat #LS-B3424-200; LifeSpan BioSciences, Inc.). Briefly, TMA sections were deparaffinized in xylene, rehydrated with graded ethanol, and washed in dH2O. Antigen retrieval was performed in sodium citrate buffer (pH 6.0). After quenching of endogenous peroxidase activity and blocking of nonspecific binding, sections were incubated with a primary antibody at 4 °C overnight. After washing with PBS, sections were incubated with the corresponding secondary antibody for 1 h at room temperature. After further washing with PBS, sections were incubated with StrepABComplex/horseradish peroxidase (Dako, Inc., K0673) for 1 h at room temperature. Chromogenic immunolocalization was conducted using 0.05% 3,3′-diaminobenzidine (Dako, Inc., K4065). All sections were counterstained with hematoxylin. Normal serum was used in the place of the primary antibody as a negative control. The staining results were measured by the German semiquantitative scoring system by a pathologist who was blinded to the clinical outcome through immunoreactive scores (IRS), based on the staining intensity and area. In general, each specimen was assigned a score according to the intensity of staining (0 = no staining, 1 = weak staining, 2 = moderate staining, and 3 = strong staining) and the percentage of stained cells (0 = 0%, 1 = 1–24%, 2 = 25–49%, 3 = 50–74%, and 4 = 75–100%). The final IRS was determined by multiplying the intensity score with the score for the percentage of stained cells. As a result, nine grades were scored as 0, 1, 2, 3, 4, 6, 8, 9, and 12. The staining pattern of Lin28B was defined as low expression (IRS: 0–6), and high expression (IRS: 8–12).

**Cell line.** Cell lines were from our lab reserved. 4TO7, MDA-MB-231, and 293 T cells were grown in DMEM media (Thermo Fisher Scientific, 10564011) supplemented with 10% FBS (Thermo Fisher Scientific, 10100147 C) and 1% penicillin/streptomycin (P/S, Thermo Fisher Scientific, 15070063) at 37 °C under 5% CO2. 3T3 cells were cultured in RPMI 1640 media (Thermo Fisher Scientific, 11875085) at the same growth conditions. 293-GPG cells were grown in DMEM media supplemented with 10% FBS, 2 mM/L glutamine (Thermo Fisher Scientific, 25030164), 1 μg/mL tetracycline (Sigma, T-7660), 2 μg/mL puromycin (Sigma, P-7255), and 0.3 mg/mL G418 (Sigma, A1720). The cultures of the primary neutrophils, BMDCs, CD4$^+$, and CD8$^+$ T cells used in this study were produced as described in the subsequent sections. All cells were regularly tested negative for mycoplasma and authenticated through Short Tandem Repeat (STR) analysis.

**Animal models.** All mice were maintained in a specific-pathogen-free (SPF) facility, and all animal studies were approved by the Institutional Animal Care and Use Committee of Shanghai Institute of Nutrition and Health. BALB/c and C57BL/6 mice (female, 4–6 weeks old) were purchased from Shanghai SLAC Laboratory Animal Corporation (Shanghai, China) and used in animal experiments. Lin28B knock-in mice (*Lin28B*$^{KI}$) in the background of C57BL/6 were generated by

Cyagen Biosciences Inc. through CRISPR/Cas9-mediated genome editing (Supplementary Fig. 1d). *MMTV-Cre* mice in the background of C57BL/6 were purchased from Shanghai Biomodel Organism Co., Ltd. *MMTV-PyMT* mice in background of FVB/n were purchased from Nanjing GemPharmatech Co., Ltd. *MMTV-PyMT* mice were backcrossed into the C57BL/6 strain to N8 and then intercrossed with *MMTV-Cre* and *Lin28B*$^{KI}$ mice, to obtain *PyMT*; *MMTV-Cre* (PyMT-Control) and *PyMT*; *MMTV-Cre*; *Lin28B*$^{KI}$ (PyMT-Lin28B) mice. *MMTV-Neu* transgenic mice, which express an activated rat *c-Neu* oncogene (*Erbb2*) in an FVB/n background, were provided by G.-H. Hu (Shanghai Institute of Nutrition and Health, University of Chinese Academy of Sciences, Chinese Academy of Sciences, Shanghai, China). *MMTV-Neu* mice were backcrossed into the C57BL/6 strain to N8, and then intercrossed with *MMTV-Cre* and *Lin28B*$^{KI}$ mice, to obtain *MMTV-Neu*; *MMTV-Cre* (MMTV-Neu), and *MMTV-Neu*; *MMTV-Cre*; *Lin28B*$^{KI}$ mice (MMTV-Neu-Lin28B). The OT-1 mice were purchased from The Jackson Laboratory (Cat #003831).

To establish the orthotopic breast cancer model, $5 \times 10^5$ 4TO7-Control or 4TO7-Lin28B cells were suspended in 50 µl of growth factor–reduced Matrigel (Corning) diluted 1:1 in PBS and injected into the fat pad of mammary gland No. 4 of 4- to 6-week-old female syngeneic BALB/c mice. Tumor masses were determined weekly by caliper measurements in two perpendicular diameters of the implant and tumor volume was calculated using the equation $V$ (mm$^3$) $= (a \times b^2)/2$, where a is the largest diameter and b is the smallest diameter. The maximal tumor size/burden allowed (16 mm) was never exceeded. The primary tumor was surgically removed at 35 days (5 weeks) after tumor inoculation, to facilitate the observation of distant metastasis. In selected studies, 4TO7-Lin28B tumor-bearing mice were treated by intratracheally injecting with 5 µg of anti-IL-6 (Bio X cells, Clone MP5-20F3, BE0046) or anti-IL-10 (Bio X cells, Clone JES5-2A5, BE0049) neutralizing antibody or with a combination of the two antibodies twice a week beginning on day 10 post-tumor injection. The antibodies were injected consecutive four times for neutrophil phenotype assay or continuing thereafter until the primary tumors were removed for lung metastasis assay of mice. In some studies, the 4TO7-Control tumor-bearing mice were treated intratracheally with 200 ng of recombinant (r) IL-6 (PeproTech, AF-216-16) or IL-10 (PeproTech, AF-210-10) or with a combination of the two cytokines twice a week beginning on day 10 post-tumor injection for consecutive four times for neutrophil phenotype assay or continuing until the primary tumors were removed for lung metastasis assay of mice. In neutrophil depletion assay, 4TO7-Control or 4TO7-Lin28B tumor-bearing mice were treated intraperitoneally with 100 µg of anti-Ly6G neutralizing antibody (Bio X cells, Clone 1A8, BE0075-1) or rat IgG control (Bio X cells, Clone 2A3, BP0089) twice a week beginning on day 14 post-tumor injection until primary tumors were resected for consecutive six times (early phase) or beginning on day 35 just after primary tumors were removed for consecutive six times (late phase). Mouse lungs are harvested when the mice developed difficulty breathing and metastatic lesions were confirmed by histological analysis.

To analyze the role of Lin28B-exosome-educated pre-metastatic niche in tumor metastasis, 100 µg of exosomes in 100 µl PBS was intravenously injected into 4–6 weeks female BALB/c mice via tail vein twice a week for consecutive 4 weeks to induce the pre-metastatic niche formation. In selected studies, cholesterol-conjugated let-7a mimics agomir-let-7a (Guangzhou Ribobio Co., Ltd, miR30000774-4-5) was also included in the injective mixture to rescue the let-7s deficiency of Lin28B-exosome at a dose of 2 nmol every 3 days. Four weeks later, 4TO7-Control cells transduced with the SFG-NES-TGL vector (4TO7-Control-TGL) were injected via tail vein into exosome-educated mice ($5 \times 10^5$ cells per mice). For PD-L2/IL-6 blocking assay, the exosome-educated mice were injected intraperitoneally with 100 µg of anti-PD-L2 (Bio X cells, Clone TY25, BE0112) or anti-IL-6 neutralizing antibody or with a combination of the two antibodies beginning on day 17 post-exosome education for consecutive three times. For Ly6G or Gr-1 blocking assay, the exosome-educated mice were intraperitoneally injected with 100 ug of anti-Ly6G (Bio X cells, Clone 1A8, BE0075-1) or anti-Gr-1 (Bio X cells, Clone RB6-8C5, BP0075) neutralizing antibody beginning on day 14 post-exosome education for consecutive five times. The growth of metastatic lesions was assessed by bioluminescent imaging at the indicated times. To examine the exosome-mediated colonization of tumor cells in organs other than the lung, 4TO7-Control cells were injected in the left ventricle of mice at $3 \times 10^4$ in 100 µl of PBS. Metastatic lesions were assessed 3–4 weeks later by histological analysis.

**Constructs and reagents.** Plasmid pCMV-HA-hLin28B is from our lab preserved and retroviral vector pBABE-puro was purchased from Addgene (Cat #1764). To construct retroviral plasmid pBABE-puro-hLin28B, full-length Lin28B was amplified from HA-hLin28B and inserted between the BamHI and SalI sites of the pBABE-puro vector.

To construct luciferase reporter plasmids, the firefly luciferase-coding region was amplified by PCR from pGL3 basic vector (Promega) and inserted between the NotI and XhoI sites of plasmid pcDNA3 to form pcDNA3-FL (FL). The human IL-10 3′ UTR was amplified by PCR from 293 T cell genomic DNA and cloned downstream of the firefly luciferase-coding region between the XhoI and XbaI sites of plasmid pcDNA3-FL to form FL-hIL-10-wt. The mutated version of IL-10 3′ UTR (FL-hIL-10-mt) were generated by site-directed mutagenesis with plasmid FL-hIL-10-wt as the template, and the QuickChange mutagenesis key kit

(Stratagene, 200518). Oligonucleotide sequences for mutagenesis were provided in Supplementary Table 8.

For FACS assay, the following antibodies were used: APC-Cyanine7 CD45 monoclonal antibody (I3/2.3) (Thermo Fisher Scientific, A15395), FITC CD11b monoclonal antibody (M1/70) (Thermo Fisher Scientific, 11-0112-82), eFluor 450 Ly6G monoclonal antibody (1A8-Ly6g) (Thermo Fisher Scientific, 48-9668-82), eFluor 450 F4/80 monoclonal antibody (BM8) (Thermo Fisher Scientific, 48-4801-82), PE CD3e monoclonal antibody (145-2C11) (Thermo Fisher Scientific, 12-0031-82), FITC CD4 monoclonal antibody (GK1.5) (Thermo Fisher Scientific, 11-0041-82), APC CD8a monoclonal antibody (53-6.7) (Thermo Fisher Scientific, 17-0081-82), eFluor 450 CD44 monoclonal antibody (IM7) (Thermo Fisher Scientific, 48-0441-82), PerCP-Cyanine5.5 CD62L (L-Selectin) monoclonal antibody (MEL-14) (Thermo Fisher Scientific, 45-0621-82), PerCP-eFluor 710 CD273 (B7-DC) monoclonal antibody (122) (Thermo Fisher Scientific, 46-9972-82), PerCP-eFluor 710, CD274 (PD-L1, B7-H1) monoclonal antibody (MIH1) (Thermo Fisher Scientific, 46-5983-42), FITC CD31 (PECAM-1) monoclonal antibody (390) (Thermo Fisher Scientific, 11-0311-81), APC CD140a (PDGFRA) monoclonal antibody (APA5) (Thermo Fisher Scientific, 17-1401-81), APC CD326 (EpCAM) monoclonal antibody (G8.8) (Thermo Fisher Scientific, 17-5791-82), PE IL-6 monoclonal antibody (MP5-20F3) (Thermo Fisher Scientific, 12-7061-81), PE IL-10 monoclonal antibody (JES5-16E3) (Thermo Fisher Scientific, 12-7101-81), PE IFN gamma monoclonal antibody (XMG1.2) (Thermo Fisher Scientific, 12-7311-81), PE T-bet monoclonal antibody (4B10) (Thermo Fisher Scientific, 12-5825-80), PE GATA-3 monoclonal antibody (TWAJ) (Thermo Fisher Scientific, 12-9966-41), PE FoxP3 monoclonal antibody (FJK16s) (Thermo Fisher Scientific, 12-5773-80), PE ROR gamma (t) monoclonal antibody (AFKJS-9) (Thermo Fisher Scientific, 12-6988-80), PE Granzyme B monoclonal antibody (NGZB) (Thermo Fisher Scientific, 12-8898-80), PE CD80 (B7-1) monoclonal antibody (16-10A1) (Thermo Fisher Scientific, 12-0801-81), PE CD86 (B7-2) monoclonal antibody (GL1) (Thermo Fisher Scientific, 12-0862-81), PE CD252 (OX40 Ligand) monoclonal antibody (RM134L) (Thermo Fisher Scientific, 12-5905-81), PE CD275 (B7-H2) monoclonal antibody (HK5.3) (Thermo Fisher Scientific, 12-5985-81), CD16/CD32 monoclonal antibody (93) (Thermo Fisher Scientific Cat# 14-0161-82), APC Ly6C monoclonal antibody (HK1.4) (Thermo Fisher Scientific, 17-5932-82). All the FACS Abs was used at 1 µl/10$^6$ cells/100 µl staining volume. For WB, immunohistochemistry, and IF analyses, the following antibodies were used: Phospho-Stat3 (Tyr705) (D3A7) XP® Rabbit mAb (Cell Signaling technology, 9145 S, WB: 1/2000), Stat3 (79D7) Rabbit mAb (Cell Signaling technology, 4904 S, WB: 1/2000), β-actin (13E5) Rabbit mAb (Cell Signaling technology, 4970 S, WB: 1/3000), Alix (3A9) Mouse mAb (Cell Signaling technology, 2171 S, WB: 1/3000), Annexin-V Antibody (Cell Signaling technology, 8555 S, WB: 1/2000), HSP70 (D69) Antibody (Cell Signaling technology, 4876 S, WB: 1/2000), Flotillin-1 (D2V7J) XP® Rabbit mAb (Cell Signaling technology, 18634 S, WB: 1/2000), anti-mouse IgG, HRP-linked antibody (Cell Signaling technology, 7076 S, WB: 1/3000), anti-rabbit IgG, HRP-linked antibody (Cell Signaling technology, 7074 S, WB: 1/3000), Rabbit Polyclonal To Human LIN28B (LifeSpan BioScience, LS-B3423, IF: 1/200, WB: 1/2000), BD Pharmingen™ Purified rat anti-mouse Ly6G (1A8) (BD Bioscience, 551459, IF:1/200), rabbit anti-mouse CD45 mAb (D3F8Q) (Cell Signaling technology, 70257 S, IF: 1/50), rabbit anti-mouse CD11b mAb (E6E1M) (Cell Signaling technology, 17800 S, IF: 1/100), rat anti-mouse Ly6C monoclonal antibody (ER-MP20) (Abcam, ab54223, IF: 1/100), rabbit anti-mouse F4/80 mAb (CI:A3-1) (Bio-Rad, MCA497, IF: 1/200), Alexa Fluor 488 donkey anti-rabbit IgG (Invitrogen, A-21206, 1/500), Alexa Fluor 594 donkey anti-rat IgG (Invitrogen, A-21209, 1/500), Alexa Fluor 488 donkey anti-rat IgG (Invitrogen, A-21208, 1/500).

**Stable cell line construction and luciferase reporter assay.** For retrovirus production, 293-GPG packaging cells were seeded in 10 cm dishes and were transiently transfected with 10 µg of pBABE-Lin28B or pBABE-vector plasmid with Lipofectamine 2000 (Thermo Fisher Scientific, 11668-019) according to the manufacturer's protocol. Two days after transfection, viral supernatant was collected, filtered, and added to the 4TO7 cells in the presence of polybrene (10 µg/ml; Sigma, H9268). 4TO7 cells were further selected with 2 µg/ml puromycin for 2 weeks to produce stable cell lines 4TO7-Control or 4TO7-Lin28B.

For bioluminescent tracking, 4TO7 stable cell lines (4TO7-Control and 4TO7-Lin28B) were infected with the SFG-NES-TGL triple reporter retroviral vector encoding herpes simplex virus type 1 thymidine-kinase 1 (HSV1-TK), green fluorescent protein (GFP), and firefly luciferase[51]. GFP-positive cells were isolated by FACS.

For luciferase reporter assay, 293 T cells were plated in 24-well plates and transfection was performed 24 h later with Lipofectamine 2000. Transfection mixtures contained 50 ng of FL reporter plasmid and 20 pmol miRNA mimics. About 2 ng of plasmid RL (Renilla Luciferase, RL) was also included in the transfection mixes. In all experiments, FL and RL activities were measured 48 h after transfection with the dual-luciferase reporter assay system (Promega, E1910). FL activity was normalized to that of RL.

**Preparation of BALFs.** Mice were anesthetized at the indicated time points after tumor or exosome inoculation. Their chests were opened, their tracheae were cannulated, and 1 ml of PBS was infused intratracheally and withdrawn. This procedure was performed three times. BALFs were centrifuged at 450×g for 10 min

at 4 °C, and the cell-free supernatant was harvested for total protein analysis using the BCA protein assay kit (Beyotime, Shanghai, China, P0011) and for enzyme-linked immunosorbent assay (ELISA) assay.

**Isolation and application of exosomes.** Exosomes of tumor tissues were isolated as described by Liu et al.[12]. In brief, the orthotopic transplanted 4TO7 breast cancer tissue was ablated and cut into small pieces at 2 weeks after tumor inoculation and cultured in RPMI 1640 medium without FBS for 12 h, after which the medium was collected and centrifuged at 800 × g for 10 min, followed by a centrifugation step of 20,000 × g for 20 min to remove cellular debris. The supernatant was recovered and filtered using a 0.2-μm filter. The collected media was then subjected to ultra-centrifugation at 100,000 × g for 90 min at 4 °C and the supernatant was discarded. The pellet was washed once in a large volume of PBS, ultracentrifuged again, and resuspended in 50–200 μl PBS. Serum exosomes were purified from the plasma of tumor-bearing mice at 2 weeks after tumor inoculation according to after dilution (1:5) in PBS. The exosomal protein concentration was determined by the BCA protein assay kit. The morphology of purified exosomes was identified by transmission electron microscopy (Tecnai 12; Philips) and the expression of exosomal markers Alix, Annexin-V, Hsp70, and Flotillin by western blotting. The full scan blot was represented in the Source Data file.

Isolation of exosomes from 4TO7 cells was carried out as previously described in ref. [52] Briefly, cell lines were grown in DMEM medium supplemented with 5% EV-depleted FBS, which was obtained by ultracentrifugation of standard FBS at 134,000 × g overnight at 4 °C followed by filtration through a 0.1-μm vacuum filtration bottle. After 48 h, the supernatant fractions were collected (500 × g; 10 min) and filtered through a 0.2-μm filter (Millipore) to remove dead cells and large debris. Exosomes were pelleted and washed in PBS by ultracentrifugation according to the methods described above.

For in vivo exosome tracking, purified exosomes were fluorescently labeled with PKH-26 membrane dye (Sigma, PKH26GL-1KT). Labeled exosomes were washed in 20 mL PBS at 100,000 × g for 90 min at 4 °C and resuspended in RPMI 1640 media. The Labeled exosomes were then intravenously injected into syngeneic 4–6 weeks female BALB/c mice (100 μg of exosomes/mouse). At 24 h after injection, individual organs were collected and cells that had taken up PKH-26 labeled exosomes were assessed using flow cytometry.

**Bioluminescent imaging.** For bioluminescent imaging, mice were anesthetized and injected retro-orbitally with 1.5 mg of D-luciferin (Thermo Fisher Scientific, L2916) at the indicated times. Animals were imaged in an IVIS 100 chamber within 5 min after D-luciferin injection, and data were recorded using Living Image software (Xenogen). To measure lung metastasis, photon flux was calculated for each mouse by using a circular region of interest encompassing the thorax. After subtracting a background value obtained from a control mouse injected only with D-luciferin, photon flux was normalized to the value obtained 30 min after injection of the tumor cells. This latter value was arbitrarily set at 100.

**Lung tissue dissociation and cell isolation.** The pre-metastatic lung tissues at 2 or 3 weeks after tumor inoculation were collected, were cut into small pieces and incubated with a dissociation solution containing 2 mg/ml collagenase type IV (Sigma, C4-28), 1 mg/ml dispase (Roche, 4942078001), and 2 U/ml DNase I (STEMCELL Technologies, 07900) at 37 °C for 30 min. The solution was pipetted every 5 min during the incubation to make the digestion homogenous. Then, the suspension was passed through a 70-μm filter and treated with Ammonium chloride red cell-lysis buffer. For macrophages, cells were stained with CD11b and F4/80 antibodies and then sorted by flow cytometry. For fibroblasts, cells were stained with CD45, CD31, and CD140a antibodies, and then CD45−CD31−CD140a+ cells were sorted. For epithelial cells, cells were incubated with CD45, CD31, and CD326 antibodies, and then CD45−CD31−CD326+ cells were collected. The cell sorting was conducted on BD FACSAria using BD FACSDiva software and the purity of the isolated subpopulations regularly exceeded 90%.

**Neutrophil isolation, culture, and transfection.** The neutrophils in the single-cell suspension of pre-metastatic lung tissue were isolated using EasySep™ mouse neutrophil enrichment kit (STEMCELL Technologies, 19762) per manufacturer's instructions to the purity of greater than 90% neutrophils. In some experiments, The single-cell suspension was stained with antibodies and neutrophils (CD45+CD11b+Ly6G+) were sorted by flow cytometry with purities of >95%. The recovered neutrophils were subject to T cell proliferation, activation, and Th1 differentiation assay or to TRIzol for mRNA purification and quantification of N1/N2 marker genes of neutrophils by qRT-PCR. Alternatively, the neutrophils were seeded in six-well plates in RPMI 1640 supplemented with 10% FBS and 1% penicillin/streptomycin at a density of 4 × 10⁶ per well. Nine hours later, the conditional media were collected for cytokine detection by ELISA or T cell proliferation and Th1 differentiation assay.

The naïve neutrophils were isolated from murine bone marrow (BM) cells, as described previously[23]. Briefly, mice were euthanized and BM was harvested by flushing the femurs and tibias with HBSS media. The cells were washed once in HBSS and were layered over a three-layer discontinuous Percoll gradient (54/64/

72%), and centrifuged at 1060 × g for 30 min. The dense band at the 64/72% interface was collected as the neutrophil fraction. The purity of neutrophils was 98% after this procedure as assessed by flow cytometric analysis. The collected neutrophils were suspended in an appropriate amount of PBS. About 0.4 ml of cell suspension containing 1 × 10⁶ neutrophils and 100 pmol of miRNA mimics were incubated on ice for 10 min and transferred into a 4-mm electroporation cuvette (BTX). The single-cuvette electroporation was carried out using the Gene Pulser Xcell System (Bio-Rad) according to the manufacturer's instructions[53]. The neutrophils were reseeded in 12-well plates in pre-warmed RPMI 1640 supplemented with 10% FBS without antibiotic at a density of 1 × 10⁶ per well and treated with an appropriate amount of Lin28B-exosome or Control-exosome (5 μg exosomes/well). Twelve hours later, the medium was replaced with a fresh complete medium for an additional 12 h of incubation. The supernatants were collected for cytokine IL-6 detection by ELISA.

For the N2 conversation assay of neutrophils in vitro, the naïve neutrophils isolated from BM were seeded in six-well plates in RPMI 1640 supplemented with 10% FBS and 1% penicillin/streptomycin at a density of 1 × 10⁶ per well. IL-6, IL-10, or a combination of both cytokines was added into the cell media. Four hours later, the neutrophils were subject to TRIzol for mRNA purification and quantification of N1/N2 marker genes of neutrophils by qRT-PCR.

**Flow cytometric analysis and intracellular staining.** The single-cell suspension of lung tissue was prepared as described above. After removal of red blood cells, the cells were centrifuged and suspended in staining buffer containing 1% FBS and 0.1% NaN₃. For cell surface staining, cells (10⁶ cells in 100 μl total volume) were incubated with aqua live dead fixable stain (Thermo Fisher Scientific, L34957), FcR-blocking reagent anti-mouse CD16/32 and fluorescently labeled antibodies and incubated at 4 °C for 1 h in the dark. For intracellular staining, cells were first conducted with surface staining and then fixed/permeabilized with a Cytofix/Cytoperm kit (BD PharMingen, 554722), and stained with antibodies against indicated cytokines. The gating strategy of flow cytometric analyses was shown as Supplementary Fig. 12.

**T cell isolation and analysis from the pre-metastatic lung.** For CD4+ and CD8+ T cells were collected from the single-cell suspension of the pre-metastatic lung tissues with EasySep™ Mouse CD4+ (STEMCELL Technologies, 19852) and EasySep™ Mouse CD8+ T cell Isolation Kit (STEMCELL Technologies, 19853), respectively. For intracellular cytokine staining, the harvested CD4+ T cells were cultured in T cell medium (RPMI 1640, 10% heat-inactivated FBS, 1x antibiotics, 1x non-essential amino acid and 50 μM β-mercaptoethanol) and stimulated with phorbol 12-myristate 13-acetate (PMA) (50 ng/ml; Sigma, P8139), and ionomycin (500 ng/ml; Sigma, I3909) for 2 h, and then added to Brefeldin A (10 μg/ml; Sigma, B7651) for another 4 h at 37 °C. The expression of cytokines indicated distinct CD4+ T cell subtypes was determined by intracellular flow cytometric staining.

**Th1 differentiation of naïve CD4+ T cells.** Spleens were mechanically dissociated and strained through a 40-μm nylon mesh to produce a single-cell suspension. After RBC removal, the naïve CD4+ T cells were isolated with EasySep™ mouse naïve CD4+ T cell isolation kit (STEMCELL Technologies, 19765). Purified CD4+ T cells were directed toward to a Th1 differentiation activated with plate-bound anti-CD3 (5 μg/ml; Bio X Cell, Clone 145-2C11, BE-0001-1) and anti-CD28 (5 μg/ml; Bio X Cell, Clone 37.51, BE-0015-1) in the presence of anti-IL-4 antibody (5 μg/ml; Bio X Cell, Clone 11B11, BP0045) and IL-12a (2 ng/ml; PeproTech, 210-12) in T cell medium (RPMI 1640, 10% FBS, 1x antibiotics, 1x non-essential amino acid, and 50 μM β-mercaptoethanol), at a density of 1 × 10⁵ per well in 96-well round-bottom plates. The neutrophils isolated from the pre-metastatic lung of tumor-bearing mice (1:1 ratio of neutrophils to CD4+ T cells) or its supernatant (50% of v/v) were also included in the mixtures. In selected experiments, a neutralizing IL-6 mAb (5 μg/ml), IL-10 mAb (5 μg/ml), IL-12 mAb (5 μg/ml; Bio X Cell, Clone C17.8, BE0051), PD-L2 mAb (5 μg/ml), and IL-12a (2 ng/ml) were used. Four days after differentiation, the cells were re-stimulated with PMA (50 ng/ml) and iono-mycin (500 ng/ml) for 2 h, and then added to Brefeldin A (10 μg/ml) for another 4 h at 37 °C. The expression of IFN-γ was analyzed by intracellular staining.

**T cell proliferation.** The naïve CD4+ and CD8+ T cells were isolated with EasySep™ mouse naïve CD4+ T cell (STEMCELL Technologies, 19765) and CD8+ T cell isolation kit (STEMCELL Technologies, 19858), respectively, from fresh mouse spleen tissues. The proliferation of T cells was measured using carboxy-fluorescein succinimidyl ester (CFSE; Thermo Fisher Scientific, L34955) dye dilu-tion assay. The purified CD4+ or CD8+ T cells were labeled with CFSE and the labeled cells were seeded in 96-well round-bottom plates at a density of 1 × 10⁵ per well in T cell medium (RPMI 1640, 10% FBS, 1x antibiotics, 1x non-essential amino acid, and 50 μM β-mercaptoethanol) supplemented with plate-bound anti-CD3 and anti-CD28. The neutrophils isolated from the pre-metastatic lung of tumor-bearing mice (1:1 ratio of neutrophils to T cells) or its supernatant (50% of v/v) were also included in the mixtures. In selected experiments, the cells were treated with a neutralizing mAb directed against PD-L2. After 3 days, the cells were harvested and subjected to flow cytometry analysis and quantification. The cells

were gated on live CD4$^+$ and CD8 T$^+$ cells, and CFSE$^{low}$ cells as a percentage of CD4$^+$ or CD8$^+$ T cells were reported as a percentage of proliferating cells.

**T cell activation assay**. The naïve CD4$^+$ and CD8$^+$ T cells were isolated from fresh mouse spleen tissue as described above. The cells were seeded in 96-well round-bottom plates at a density of $1 \times 10^5$ per well and cultured in T cell medium (RPMI 1640, 10% FBS, 1x antibiotics, 1x non-essential amino acid, and 50 μM β-mercaptoethanol) supplemented with αCD3/αCD28 alone or in co-culture with the pre-metastatic lung-infiltrating neutrophils (1:1 ratio of neutrophils to T cells) or the neutrophil conditioned media (50% of v/v). The fraction of CD4$^+$ and CD8$^+$ T cells expressing CD44 or CD62L was determined at 24 h by flow cytometry.

**Antigen-restricted T cell proliferation assays**. BMDCs were produced as described by Clarke et al.[54]. In brief, BM was flushed from femurs and tibias of C57BL/6 mice. After RBCs removal, the cells were incubated for 30 min at 37 °C on Petri dishes to remove adherent macrophages. Non-adherent progenitor cells were seeded at $1.5 \times 10^6$ cells/ml in[54] 24-well tissue culture plates in RPMI 1640 with L-glutamine supplemented with 10% heat-inactivated FBS, β-mercaptoethanol (50 μM), penicillin/streptomycin (100 μg/ml), and 1% murine GM-CSF (Pepro-Tech, 315-03). DCs were used on days 6 and 7. The naïve CD8$^+$ T cells were isolated from spleens of 8–16- week old OT-I mice and labeled with CFSE. DCs were pulsed overnight with OVA$_{257-264}$ (Invivogen, vac-sin) at 1 μM and harvested, washed, and resuspended in RPMI 1640 with L-glutamine supplemented with 10% heat-inactivated FBS, β-mercaptoethanol (50 μM), and penicillin/streptomycin (100 μg/ml). CFSE-labeled OT-I cells were co-cultured with pulsed DCs at a 1:2 ratio of DCs to T cells ($1 \times 10^5$ BMDCs: $2 \times 10^5$ T cells) in 96-well around-bottom plates. Neutrophils purified from the pre-metastatic lung of tumor-bearing mice were also added in the mixture at a density of $1 \times 10^5$ per well for antigen-restricted T cell proliferation assay.

**Libraries preparation and analysis of small RNA sequencing**. Small RNA libraries were constructed using the QIAseq miRNA library kit (QIAGEN) per the manufacturer's protocol. Reverse transcription primer was hybridized after 3′ adapter ligation of 10 ng RNA per sample, following 5′ adapter ligation. Eleven cycles of PCR were performed by Illumina feasible barcode primers after first-strand cDNA synthesis. The prepared libraries were resolved on a native 6% Novex polyacrylamide gel (PAGE). DNA fragments corresponding to 147–157 bp (including 3′ and 5′ adapters) were recovered in 10 μL of DNase- and RNase-free water. Libraries were quantified by the Agilent 2100 bioanalyzer using High Sensitivity DNA Chip (Agilent Technologie) and sequenced using an Illumina NovaSeq 6000 (Illumina, USA). After raw data cleaning, the sequences from the miRNA-seq data sequences with a length between 18–40 nt were aligned against miRBase. The sequencing experiments and analyses were completed by Shanghai Biotechnology Corporation.

**Quantitative real-time PCR**. The sequences of primers used in this work were provided in Supplementary Table 8. Quantitative real-time PCR (qRT-PCR) of mRNA was performed as described previously[55]. In brief, total RNA was isolated by TRIzol LS reagent (Thermo Fisher Scientific, 10296-028), and DNA was removed from RNA samples with RNase-free DNase I (TaKaRa, D2215). Total RNA was reverse transcribed into cDNA with Moloney murine leukemia virus (MMLV) reverse transcriptase (Promega, M1701) and oligo(dT$_{18}$) according to the manufacturer's instructions. Fluorescence real-time PCR was performed with the double-stranded DNA dye SYBR green real-time PCR Master Mix (ToYoBo, QPK-201) with the ABI Prism 7900 system (Perkin-Elmer, Torrance, CA).

For miRNA qRT-PCR detection, Total RNAs extracted from tissues, cells, or tumor exosomes were used to generate cDNA using MMLV reverse transcriptase with specific stem-loop primers for miRNAs or specific primer for U6. qRT-PCR was performed using SYBR Green as described above. Target gene expression was normalized to the housekeeping genes U6. For serum exosomal RNA purification, the same amount of control miR-16 was added to the individual RNA mixture at the isopropanol precipitation step as the internal control. Target miRNA expression was normalized to miR-16 to obtain their relative expression in the serum exosomes.

**Immunofluorescent staining of lung tissue sections**. Lungs of mice were collected at the designated time points and rinsed with pre-chilled PBS. Tissues were fixed in 4% PFA, dehydrated by 30% sucrose PBS solution overnight, embedded in OCT on liquid nitrogen, and stored at −80 °C. Lungs were sectioned to 5 μm thickness, blocked, and permeated in 5% donkey serum in 0.1% Triton X-100 containing PBS, followed by primary antibody incubation overnight. DAPI staining was performed at room temperature for 5 min at the concentration of 1 μg/ml (Fisher Scientific, P36934) and fluorescence was monitored by confocal laser microscopy (Carl Zeiss).

**Statistics and reproducibility**. The data presentation and statistical analyses are described in the figure legends. Cox proportional hazards regression and multi-variate Cox proportional hazards analyses were performed with SPSS 19 statistical software. The remaining data analyses were performed by GraphPad Prism 8 (GraphPad Software, USA). For all statistical tests, the 0.05 level of confidence was accepted for statistical significance.

**Reporting summary**. Further information on research design is available in the Nature Research Reporting Summary linked to this article.

## Data availability
The small RNA-sequencing data of tumor exosomes generated in this study have been deposited in the GEO database under accession code GSE185588. The publicly available data used to analyze for the specific metastatic site of breast cancer patients induced by Lin28B and let-7a expression in this study are available in the GEO database under accession code GSE12276 [The data used to analyze for the clinical significance of let-7s upon the overall survival of breast cancer patients used in this study are available in the GDC TCGA breast cancer clinical cohort [https://xenabrowser.net/datapages/?dataset=TCGA-BRCA.mirna.tsv&host=https%3A%2F%2Fgdc.xenahubs.net&removeHub=https%3A%2F%2Fxena.treehouse.gi.ucsc.edu%3A443]. The remaining data are available within the Article, Supplementary Information or Source Data file. Source data are provided with this paper.

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

## Acknowledgements

We thank Lin Qiu, Yifan Bu, Feng Zhao, Zhonghui Weng, Aimin Huang, Cheng Wang, Kai Wang, Konglun Pan, and Hui Zhu at the Institute of Nutrition and Health core facilities for technical support. This study was supported by the National Key Research and Development Program of China (2020YFA0112300, 2017YFC1600100 to L.Z.), Program of EnShi TuJia & Miao Autonomous Prefecture Bureau of Scientific & Technological Affair (L.Z.), and National Natural Science Foundation of China (82173007 and 81872369 to L.Z., 81301855 to M.Q.).

## Author contributions

L.Z. supervised the work. L.Z. and M.Q. designed the experiments and prepared the manuscript. M.Q. performed most experiments. Y.X. contributed to clinical sample collection and pathology analyses. Z.Z. contributed to the computational statistical analysis. Y.W., X.W., L.L., K.X., Y.S., C.D., and W.J. performed a specific subset of the experiments and analyses. All authors discussed the results and commented on the manuscript.

## Competing interests

The authors declare no competing interests.
