## [Peer Review File · Nature Communications]

Reviewers' Comments:

Reviewer #1:

Remarks to the Author:

The manuscript presents the interesting observation that breast cancer derived exosome modules pre-metastatic niche to promote metastasis. The authors found high Lin28B expression in TNBC down-regulates let-7s in exosomes, which alters the lung microenvironment to enrich the N2 neutrophils and generate a favorable immune-landscape for cancer cell colonization. The authors provided strong evidence to show how N2 neutrophils affects the immune cells. However, more data is needed to show how exosomes lodge into pre-metastatic niche. The relevance of this study to clinical setting is also questionable with current data.

Major comments:

1. It is not clear whether the finding of this study is specific to TNBC. While the authors found that Lin28B is particularly high in TNBC, the survival analysis and the multivariate Cox regression analysis were performed in breast cancer patients with mixed subtypes, and the majority of those patients are ER+. From current data, it is not clear if Lin28 is specific to TNBC or it is a general metastasis inducer. Lin28B and let-7 need to be analyzed in each of subtype separately to fully understand their roles in breast cancer. Other subtype cell lines should also be used in animal experiments to address this question.
2. It is not clear whether the finding of this study is specific to lung metastasis. While the authors found that "conversion of neutrophils into N2 only occurs in pre-metastatic lung, but not in the spleen, peripheral blood, or bone marrow", there is not enough clinical evidence to support the specificity of Lin28B to lung metastasis. The patient cohort used in this study did not distinguish lung metastasis from other distant metastases. The authors need to compare the role of Lin28B and let-7s on metastases to different organs with the clinical data. There is limitation of the animal models used in this study as it is rare to see metastasis to organs other than lung. The authors should try i.c. injection after systemic exosome treatment to investigate the change of metastasis rate to different organs.
3. Why do the breast cancer-derived exosomes modulate only the lung environment. Does the exosomes have higher affinity to lung? The authors should check the bio-distribution of exosomes in mice.
4. While many let-7 miRNAs are down-regulated in the exosomes as shown in Figure 7a, it is not clear why authors selected only let-7a for the following study.
5. Lin28A also suppresses let-7s (PMID: 24825413, 26440890). As the authors mentioned in the introduction, "Lin28A is mainly expressed in HER2+ subtype, whereas Lin28B is mainly expressed in the TNBC subtype". Since both of them inhibits let-7s, what are the levels of let-7s in HER2+ breast cancer cells? Do the exosomes from HER2+ cells have the same effect?

Minor comments:

1. In Figure 3i, it is described as "intratracheal administration of the recombinant IL-6 and IL-10". However, in figures, it is labeled as "intraperitoneally".
2. Please note the time point of Figure 4a.
3. Fig 6a: a typical shape of exosome under transmission electron microscopy should be saucer shape instead of sphere shape.
4. In Figure 6c, 4T07 was injected through I.V. as labeled in figures. However, I.V. injection was not described in the "Animal Models" part in the "Materials and Methods" section.

Reviewer #2:

Remarks to the Author:

To the Authors:

In the manuscript by Mei-Yan Qi et al, the authors demonstrated that breast cancer stem cells expressing high Lin28B produce exosomes which contain low let-7s which in turn influence neutrophils to convert to a N2 phenotype and promote immune suppression of other immune cells, such as T cells, via PD-L2, which can then promote metastasis. The work has great potential and has clinical significance.

However, there are many questions that need to be addressed.

First, the work is a bit overwhelming and disconnecting. Data in one figure are often not carried over and further studied in other figures. For examples, the stemness of tumor cells, the chemokines listed in figure 2 (CXCL1, CXCL2, CXCL3, and CXCL5), then IL-6 and IL-10 chemokines are then introduced in Figure 3, the study of neutrophils, then the study of T cells and not so much neutrophils, the study of exosomes presented late in the manuscript, the role of PD-L2, etc. Thus, it is unclear if there is a focus.

Moreover, there are publications on exosomes involving Let-7 in exosomes as a biomarker in plasma (Kentaro Matsuura, et al, Hepatology 2017) and tumor cells expressing LIN28A (Vanessa A. Enriquez et al, BioMed Research International 2015) in invasion, etc. The authors need to highlight these works and also highlight how their work is novel as compared to these other works. There is little mechanistic studies here. For example conditional knockout tumors cells for LIN28A. And little understanding of Let-7 and other miRs in the exosomes and their biogenesis from tumor stem cells and what determines their uptake in cells, such as neutrophils, in pre-metastatic sites. Is there exosome uptake in other cells at pre-metastatic sites? Many details of the events taking place at pre-metastatic niches are incomplete. The work is highly dependent on neutrophils, yet there is no neutrophil images at pre-metastatic sites and surprisingly there is no work on NET formation at pre-metastatic niches as well. In addition to targeting with anti-ly6G abs, the authors should use genetic models of neutrophil depletion to determine if these cells are key for exosome uptake and for further education of immune suppression at these sites.

There are questions per figure:

Figure 1:

In a, provide an additional means of quantification, not just based on immunohistochemistry.

In h, there are no increases in primary tumor size yet in extended figure 1h, the authors show increases in proliferation which would likely lead to an increase tumor size. Explain.

Explain the connection of Lin28B and stem cells, are the Lin28B tumor cells all tumor stem cells as well.

Figure 2:

In b, there are positive Ly6C cells, is the increase cells, also myeloid cells and/or dendritic cells, in addition to neutrophils are pre-metastatic sites.

In d, show stains for Ly6C, CD11b, and CD45 as well.

In e, the late anti-Ly6G with no affect, are there neutrophils present, show early vs late neutrophils staining and quantification, are there NETs formed and hence explains the failure of this therapy later on.

Figure 3:

Explain in more detail N2 conversion and why was G-CSF ignored here.

Figure 4:

This figure and beyond has little work on neutrophils.

What is the role of NK cells here.

Why splenic T cells and not study T cells in the pre-metastatic sites

Figure 5:

PD-L2 can be expressed on DC and macrophages, further support that these cells are critical in this process

Figure 6:

Exosome characterization for miRs need to be shown

Why tail vein tumor models here when orthotopic models used in previous figures.

What cells in the Pre-metastatic sites take up exosomes?

Figure 8:

For the serum exosomes, what cells produce the exosomes in the serum.

In b, let-7a differences are not impressive and likely not significant as shown.

Where is the neutrophil link.

Reviewer #3:

Remarks to the Author:

Qi et al., provide data that expression of the embryonic stem cell factor Lin28B, is expressed predominantly in TN breast cancer and led to production of exosome which are inducing lung fibroblasts to release IL6 and IL10. These polarized neutrophils, the dominant myeloid cell type in

the lung pre-metastatic niche, to an immunosuppressive phenotype. Mechanistically, the authors show that exosome from Lin28B induced this effect as they do not contain Let-7a, which is able to suppress IL6 and IL10 production.

The data are mostly convincing; however, they are not well presented. Sometime is difficult to understand what the message is and some data, especially at the end, about pSTAT3 are not clearly interpreted (see point 14). There are even data which are shown without any comment: the entire Extended Data Fig. 5 is not mentioned anywhere. Extended Data Fig. 5 data are important, they represent a validation of the finding from the 4T07 model using the spontaneous PyMT tumours model where Lin28B is overexpressed. The data confirm the immunosuppression effect on T cells in a completely different setting (Point 1)

Also, the text has several typos, for instance Extended is consistently spelled wrong. The text needs to be improved from that point of view as well. Additionally, I thought the format of Nature Communication was Supplementary Figures? Indeed, on one occasion, this is also used (Line 210: Figure S4G).

Importantly, a previous study highly relevant for this work is not properly cited and not discussed at all. As explained below, the authors need to reconcile their funding with the one from Liu, Y., et al., Cancer Cell 2016. All the references need to be revised to confirm they are accurately cited. (Point 12).

In summary, despite the text at times confusing, the data are overall convincing, and in my opinion the manuscript, upon review, can meet the Nature Communication standards.

Comments:

1. The data about PyMT mice in Extended Data Fig. 1 and 5 need to be discussed more (in addition to be entirely mentioned). This validation means that the pro-tumorigenicity of the pre-metastatic niche in PyMT mice, which was previously characterized by a large bulk of studies, is increased upon Lin28B overexpression. Comparing the level of Lin28B with or without Lin28B overexpression is key to important the changes here. Do PyMT exosome from control mice contain Let-7s or at least some? If not, the discussion needs to envisage other mechanism whereby Lin28B overexpression increase metastatic potential, maybe by changing the cancer cell themselves.

2. In relation to this, Extended Data Fig. 8i and Fig. 7h,i show that ALDH+ 4T07-Lin28B cells express increased Lin28B and they release exosomes with much lower let-7s levels responsible for increase metastases. Does Lin28B expression increase the number of ALDH+ cells as well? Data from the PyMT model would argue this could be the case (Extended Data Fig. 1g). What are the ALDH+ cancer cells in the 4T07-control vs 4T07-Lin28B? Are the ALDH+ cancer the only source of Let-7s depleted exosome? The authors also need to discuss that changing in the cancer property them self can contribute to the enhanced metastatic activity.

3. Figure 1f, if Lin28B expression is higher in TN as TN have a worst prognosis, the inverse correlation between Lin28B expression and overall post-surgery survival could just be a consequence of this; does this hold true among TN only? While this does not impact on the relevance of Lin28B expression and its mechanism, it is an important difference to clarify.

4. Fig 2b,c and Extended Data Fig 2e,f the increase of Mac seems very similar to the one of neutrophils, but of course the absolute number of neutrophils is much higher. The text needs to describe the data accordingly.

5. It is unclear what the sentence in line 128-131 means. Myeloid-derived suppressor cells are a functional definition, I personally do not think a particular function of immunosuppression could be deduced by the presence of Ly6C. There is no rule preventing Ly6C+ neutrophils to act as immune-suppressive cells (as this very work shows). I appreciate that the authors stay away from this confusing terminology and, for consistency, I would suggest removing this sentence as well. However, it is interesting to see the decrease in Ly6C expression level in premetastatic lungs, but this occurs in both control and Lin28B expressing cells. A comment is required here, as 4T07 control tumour inducing pre-metastatic changes as well.

6. Fig 2g, the text mentioned that this is a time point of early stage it is not clear if this is at the time before tumour resection, day 35? It should be clear at what point of the schematic in 2e is referring to. Moreover, I would be surprised if the Ly6G antibody can go on depleting neutrophils much longer than 14 days, especially in BALB/c mice, which have a high number of neutrophils. In other words, I doubt that after tumour resection the treatment was still efficient. Unless the Early phase and Late phase are alternative settings, but this should be clear. Finally, if Early and Late are independent settings, the data should for both be shown, at least in the supplementary data. The Late setting should include the control as well as there could be onset of metastasis in absence of Lin28B triggered by surgery, in which case, it could still be neutrophils dependent.

7. Why Fig 3a say "lung" if the RNA is from isolated neutrophils?

8. The sentence in line 221-225 is a copy of the previous sentence (217-221) but it refers to the wrong figure (4e,f). Also, Fig 4f is not there.

9. Extended Data Fig. 4h-j, it is not clear what Control and N2 neutrophils refers too, are the pre-metastatic neutrophils from 4T07 (Control) or 4T07- Lin28B (N2)? If so it should be clear.

10. Line 242, in Fig. 5a for PD-L1 should state unchanged levels rather than slightly upregulated, more in line with the same as Fig 5b for CD86, OX40L, and ICOSL showing similar results.

11. Line 263, please refer to the Fig 3b-c where the levels of IL12a are shown.

12. Line 314-316, the interpretation of the data shown in Extended Data Fig. 7a-c is incorrect: the data in Extended Data Fig. 7a showed that both fibroblasts and epithelial cells are expressing CXCLs, but Extended Data Fig. 7b-c shows that only Fibroblasts respond with a higher CXCLs production when 4T07-Lin28B cells are used compared to 4T07-control. Epithelial cells do not show an increase in CXCLs in response to 4T07-Lin28B vs Control.

Having this interpretation right is essential as it can reconcile the data presented in this manuscript with a previous study showing that exosome target lung epithelial cells and activate a pre-metastatic niche via TLR3 dependent neutrophils recruitment (Liu, Y., et al., Cancer Cell (2016). <http://doi.org/10.1016/j.ccell.2016.06.021>).

Moreover, due to the similarity of this work to the finding of the above-mentioned study, the authors need to discuss it. Conversely, Liu, Y., et al., is mentioned once in the introduction and cited in a rather strange way, to support the statement of bone marrow-derived cells composition. It is also cited in the M&M to refer to exosome isolation. Liu, Y., et al., and the present manuscript both present data about an exosome-dependent pre-metastatic neutrophils modulation, therefore a detailed analysis of the potential differences in the data is needed.

13. Line 323, I guess it means IL10 only not IL6, IL10.

14. The data about STAT3 signalling are not clear to me. Extended Data Fig. 9c shows how that the STAT3 pathway inhibitor S3I-201 increases CXCLs in cells treated with both Control and Lin28B-exosome. There is no difference between Control and Lin28B-exosome, why the text links this effect to the Let-7s deficiency? It is unclear from the text what the interpretation is.

What would the level of IL16 and IL10 expression be in this STAT3 inhibitor experiment?

More generally, CXCL1 is a STAT3 target, why STAT3 inhibitor would increase CXCL1? Also Lin28B-exosome in the panel before increase pSTAT3, how Let-2a affect pSTAT3? What is in the authors interpretation about the cascade of events following exosome exposure, pSTAT3, CXCLs, ILs expression and when Let-2a inhibition act? Neutrophils modulation is due to ILs signalling or they act as paracrine? What cause what in the cascade of event?

The final model proposed is not entirely clear. Using a final drawing to explain the model would help the authors explain?

Reviewer #4:

Remarks to the Author:

In this study, Mei-Yan Qi and collaborators indicate that lin28b induces breast cancer lung

metastasis by producing immune-suppressive pre-metastatic niches and acting by suppressing the let-7 amount in exosomes of cancer cells.

In principle this is a very interesting study.

My main concern about this study is that most of the results are based on a single mouse breast cancer line used as control or that overexpresses Lin28b. Additional cell lines should be used, and including relevant model of human breast cancer with high level of Lin28b expression in immunocompromised mice.

In Extended Data Fig. 1 the over-expression of Lin28b in 4T07 is massive compared to the control cells which can indicate that results seen only derive from this artificial lin28b expression, but this never happens in real breast cancers.

There are not Lin28b loss-of-function experiments in the study to demonstrate that all results shown are not only the product of such high ectopic expression of Lin28b.

On the other end in the TCGA database, the levels of Lin28b mRNA expression in breast cancer is very low and only limited to few breast cancer specimens. This in principle excludes that Lin28b is a major driver of breast cancer metastasis as proposed by the authors.

How can the authors explain this contradiction?

Therefore, before to evaluate the study further I propose:

- 1) The authors should identify breast cancer cell lines that express high levels of endogenous Lin28b to use in this study and to obtain CRISPR/CAS9 Lin28b KO counterparts to assess the most relevant immunological/metastatic phenotypes analysed in the study.
- 2) Confirm the clinical relevance of Lin28b using gene expression data from independent Breast cancer cohorts from public databases (TCGA, GEO) and focus on Lin28b and let-7.
- 3) Measure Lin28b by Western blotting in their own samples (including cancer and normal) to make sure that the signal in the immunohistochemistry samples really corresponds to Lin28b protein.
- 4) Evaluate the relevance of let-7 in this pathway by repressing the miRNA in cells that are KO for Lin28b

Additionally figure legends are poorly written and mostly lack important details. For example no information on which housekeeping genes were used for normalisation in RT-qPCR.

Response to reviewers

General Note: The reviewers' comments were copied from the Editor's decision letter without changes and are in quotes and bold letters. Our responses below follow individual comments.

REVIEWER COMMENTS

Reviewer #1 (Remarks to the Author): with expertise in breast cancer metastasis

The manuscript presents the interesting observation that breast cancer derived exosome modules pre-metastatic niche to promote metastasis. The authors found high Lin28B expression in TNBC down-regulates let-7s in exosomes, which alters the lung microenvironment to enrich the N2 neutrophils and generate a favorable immune-landscape for cancer cell colonization. The authors provided strong evidence to show how N2 neutrophils affects the immune cells. However, more data is needed to show how exosomes lodge into pre-metastatic niche. The relevance of this study to clinical setting is also questionable with current data.

Major comments:

"1. It is not clear whether the finding of this study is specific to TNBC. While the authors found that Lin28B is particularly high in TNBC, the survival analysis and the multivariate Cox regression analysis were performed in breast cancer patients with mixed subtypes, and the majority of those patients are ER+. From current data, it is not clear if Lin28 is specific to TNBC or it is a general metastasis inducer. Lin28B and let-7 need to be analyzed in each of subtype separately to fully understand their roles in breast cancer. Other subtype cell lines should also be used in animal experiments to address this question."

Response: We thank this reviewer for their suggestion to clarify the precise role of Lin28B and let-7 in each subtype of breast cancer. To address this suggestion, we assessed the expression of Lin28B in an additional tumor tissue microarray (TMA) (Cohort III) (Supplementary Data 1). Combined with the previous results from TMA Cohort II, our data showed that high Lin28B expression is associated with poor prognosis and is an independent prognosticator of BC survival for all three BC subtypes (Fig. 1h and Table S3-5). These results indicate that the correlation of Lin28B expression with poor prognosis among BC patients is a general phenomenon, and not just specific to the TNBC subtype.

For let-7s, we analyzed the RNA-sequencing data of the GDC TCGA breast cancer clinical cohort (<https://xenabrowser.net>, 1073 BC patients). In these tumors, the low expression of let-7a was associated with reduced overall survival of BC patients (Fig. 8j). Similar correlation also was observed for let-7b, let-7c, and let-7d (Supplementary Fig. 9I). Due to the lack of information on pathological subtypes of the BC patients in this cohort, we could not assess whether the correlation also was recapitulated in all three BC subtypes. We speculate that the

correlation may be shared among all subtypes. Interestingly, high Lin28B expression and low let-7a expression in BC tumors dictated lung metastasis, but not brain or bone metastasis, in a cohort of 204 patients from the GSE12276 dataset (Table 6 and 7), indicating similar function and cooperative mechanism of these two effectors in promoting BC progression and lung metastasis. These data suggested that the low let-7s may correlate with reduced overall survival of BC patients in all three BC subtypes.

With the *MMTV-Neu* mouse model (HER2⁺ subtype), we determined that Lin28B did not accelerate primary tumor growth (Supplementary Fig. 1i), but increased the incidence of lung metastasis in tumor-bearing mice (Supplementary Fig. 1j). Thus, each of three distinct mouse models (4TO7, PyMT and MMTV-Neu) indicated that Lin28B promotes lung metastasis of breast tumors.

We noticed (as presented in Fig. 1d) that the percentage of over-expression of Lin28B was highest in tumors of the TNBC subtype. We therefore believe that studying the metastasis-promoting mechanism of Lin28B will be of particular significance for the prevention and treatment of TNBC progression.

“2. It is not clear whether the finding of this study is specific to lung metastasis. While the authors found that “conversion of neutrophils into N2 only occurs in pre-metastatic lung, but not in the spleen, peripheral blood, or bone marrow”, there is not enough clinical evidence to support the specificity of Lin28B to lung metastasis. The patient cohort used in this study did not distinguish lung metastasis from other distant metastases. The authors need to compare the role of Lin28B and let-7s on metastases to different organs with the clinical data. There is limitation of the animal models used in this study as it is rare to see metastasis to organs other than lung. The authors should try i.c. injection after systemic exosome treatment to investigate the change of metastasis rate to different organs.”

Response: To address this reviewer’s suggestions, we searched the GEO dataset for the relevant clinical data to explore the potential metastatic tropism mediated by Lin28B. Within GEO, a clinical cohort of 204 patients met our requirements and was subjected to further analysis (GSE12276, Supplementary Data 1). The results showed that higher Lin28B expression in BC tumors correlated significantly with lung metastasis ($P=0.045$, see Table S6). In contrast, no significant association was detected between Lin28B expression in primary tumors and in brain or bone metastasis ($P=0.300$ and 0.369 , respectively; see Table S6). This result is consistent with our data showing that Lin28B expression in BC tumors primarily promotes lung metastasis in several mouse models (Fig. 1k and Supplementary Fig. 1g, j). In contrast, we were unable to obtain relevant clinical data surveying the relationship between the gene expression profile of BC tumors and the liver metastasis tropism, and so lack clinical data to support a role for Lin28B in promoting liver metastasis. Notably, when using the mouse orthotopic BC models (4TO7, PyMT and MMTV-neu), we occasionally observed Lin28B-induced bone metastasis (Supplementary Fig. 1c), but never observed liver metastasis (data not shown), suggesting that Lin28B shows weaker correlation with liver metastasis.

Using the same dataset (GSE12276, Supplementary Data 1), we determined that low let-7a expression in BC tumors correlated with increased lung metastasis, but not with brain or bone metastasis (Table 7). Notably, low let-7a expression predicted reduced lung-metastasis free survival (Fig. 8k), supporting the close correlation of both Lin28B and let-7a with lung metastasis in human BC patients.

In other work, we conducted i.c. injection of 4TO7 cells after systemic Lin28B-exosome treatment to explore the pro-metastatic effect of Lin28B-exosome in different organs (Supplementary Fig. 6k). Education by Lin28B-exosome increased the metastasis of 4TO7 cells to lung, but not to bone or liver (Supplementary Fig. 6k). By exosome tracing assay, we found that exosomes were extensively absorbed by multiple organs (Supplementary Fig. 6j). However, neutrophil N2 conversion was found primarily in lung (Fig. 6h), and not in other organs such as liver or bone marrow (Supplementary Fig. 6l), hinting that neutrophil N2 conversion may be the primary mechanism employed by Lin28B-exosome to promote lung metastasis.

Collectively, the clinical data indicated that high Lin28B and low let-7 expression determine breast tumor metastasis to the lung. I.c. injection experiments further confirmed this possibility.

“3. Why do the breast cancer-derived exosomes modulate only the lung environment. Does the exosomes have higher affinity to lung? The authors should check the bio-distribution of exosomes in mice.”

Response: We thank the reviewer for this suggestion. We repeated the exosome tracing assay and showed that exosomes were absorbed extensively by multiple organs, including lung, liver, spleen, and bone marrow (Supplementary Fig. 6j). However, Lin28B-exosome-mediated metastasis was observed only in lung in our model, and not in liver or bone (Supplementary Fig. 6k). Moreover, N2 conversion of neutrophils appeared mainly in lung (Fig. 6h), but not in liver or bone marrow (Supplementary Fig. 6l). These results indicated that the absorption of exosome is not sufficient for the process of metastasis. Rather, local cell response by stromal cells and myeloid cells is another mechanism for the ultimate metastasis. It is possible that particular cells in lung tissue respond to exosomes to facilitate the sequential events needed for pre-metastatic niche formation, which is not well established in other organs. We think that lung differs from liver or bone marrow in the abundance of fibroblasts and epithelial cells, which is capable of not just receiving exosome signaling, but also of releasing chemokines and inducing subsequent neutrophil recruitment and (ultimately) N2 conversion. The detailed mechanistic differences among various organs deserve further study.

“4. While many let-7 miRNAs are down-regulated in the exosomes as shown in Figure 7a, it is not clear why authors selected only let-7a for the following study.”

Response: We have demonstrated that let-7a was one of the most abundant miRNA among all

of the let-7 family members, as assessed by qRT-PCR in exosomes isolated from 4TO7 cells and primary tumors (Fig. R1a, please see page 13). At the request of Reviewer 2, we also conducted high-throughput sequencing of small RNAs in 4TO7 tumor exosomes. Exosomes was pooled from a total five 4TO7 tumors for sequencing, to avoid potential sample-to-sample variation. The results showed that let-7a was indeed highly abundant among all the miRNAs in the exosomes (Supplementary Data 1). Although, let-7a only ranked fifth among all the let-7 family members (Supplementary Data 1), a result that is not totally consistent with our qRT-PCR data (Fig. R1a, please see page 13). We had been told (by the staff of this sequencing company) that qRT-PCR currently should be the most accurate method for quantifying miRNAs. Consequently, we transfected let-7s, together with a FL- FL-hIL-10 3'UTR reporter, into 293T cells. In this experiment, each of the let-7s demonstrated inhibition of the expression of the FL-hIL-10-wt reporter, but not of the mutant FL-hIL-10-mt reporter (Fig. R1b, please see page 13), indicating that each of the let-7s has comparable activity for the repression of target gene expression. This result was consistent with the observation that the seed sequence of all the let-7s family members (2-8 nt) was conserved (Fig. R1c, please see page 13). This result suggests that we can use let-7a as a representative let-7s member family. To address this point, we modified the text in the Results section in the revised MS. The inserted sentence is as follows: "Our results showed that each of the let-7s has comparable activity for the repression of target gene expression (data not shown); therefore, we selected let-7a as the representative let-7 member in most experiments" (page 21, paragraph 2, line 1 in the revised manuscript).

Fig. R1 Let-7s family members show similar activity to inhibit their target gene expression.

(a) Let-7s family members were quantified in exosomes isolated from the 4TO7 cells and the primary tumors by qRT-PCR. U6 served as internal control. The relative proportion of the individual members was shown. (b) Let-7s family members showed inhibitory activity to reporter FL-hIL-10 3'UTR. 293T cells were transfected with FL reporter FL-hIL-10-wt or FL-hIL-10-mt, together with let-7s or control mimics. The internal control plasmid Renilla luciferase (RL) was also include in the transfection mixture. Luciferase activity was analyzed 48 h later and the ratio of FL in relative to RL (FL/RL) were shown (n=3). Data are presented as the mean \pm SEM of three independent experiments, and p-values were calculated using two-tailed Student's *t*-test. **p* < 0.05, ns, no significance. (c) The alignment of let-7s family members by CLUSTAL 2.1. The seed sequence (2-8 nt) signaled in blue was conserved among all the let-7s family members.

"5. Lin28A also suppresses let-7s (PMID: 24825413, 26440890). As the authors mentioned in the introduction, "Lin28A is mainly expressed in HER2+ subtype,

whereas Lin28B is mainly expressed in the TNBC subtype". Since both of them inhibits let-7s, what are the levels of let-7s in HER2+ breast cancer cells? Do the exosomes from HER2+ cells have the same effect?"

Response: Lin28A was reported to be highly expressed in HER2⁺ breast tumors. However, many HER2⁺ cell lines, including the human SKBR3, MDA-MB-435, and AU565 cell lines, and the mouse NF639 cell line, there was no detectable Lin28A expression, although Lin28A was detected in T47D cells (luminal subtype) (Fig. R2a). That observation appears to conflict with our other results, suggesting further clarification is needed.

Nonetheless, we still wished to know whether Lin28A/let-7s play a similar role in tumors with elevated Lin28A expression (as observed for tumors with elevated Lin28B expression). When we introduced Lin28A into 4TO7 cells, the level of let-7a was clearly decreased in both the cells and in the corresponding exosomes (Fig. R2b). In addition, our preliminary results showed that Lin28A-exosome (derived from 4TO7-Lin28A cells) also induces an immune-suppressive pre-metastatic niche in the lung via a mechanism that overlaps with, but is distinct from, that exerted by Lin28B-exosome (our unpublished data). Given that Lin28A and Lin28B share a similar mechanism to inhibit let-7s expression, this result suggests that decreased let-7s content in exosomes is crucial for the immune-suppressive pre-metastatic lung. We have added a description of this result to the Discussion section in the revised MS. The added sentence is as follows: "Moreover, our preliminary results showed that Lin28A expression in 4TO7 cells reduced the level of let-7a in both the cells and in the corresponding exosomes. Lin28A-exosome (derived from 4TO7-Lin28A cells) also induces an immune-suppressive pre-metastatic niche in the lung, employing a mechanism that overlaps with, but is distinct from, that exerted by Lin28B-exosome (our unpublished data). Given that Lin28A and Lin28B share similar roles in inhibiting let-7s expression, this result supports the hypothesis that reduced let-7 levels in exosomes are crucial for the immune-suppressive state of the pre-metastatic lung" (page 30, paragraph 2, line 6 in the revised manuscript).

Fig. R2 Lin28A inhibits let-7a production in 4TO7 cells and the exosomes.

(a) Lin28A was detected by western blotting in the HER2⁺ cell lines as indicated. T47D cells served as a positive control. β-actin served as a loading control. (b) Let-7a was reduced by Lin28A in 4TO7 cells and the corresponding exosomes. Data are presented as the mean ± SEM of three independent experiments, and p-values were calculated using two-tailed Student's *t*-test. ***p* < 0.01.

Minor comments:

"1. In Figure 3i, it is described as "intratracheal administration of the recombinant IL-6 and IL-10". However, in figures, it is labeled as "intraperitoneally"."

Response: We are sorry for our mistake. We have corrected the label to "intratracheally" in Fig. 3i in the revised Figures.

"2. Please note the time point of Figure 4a."

Response: We thank reviewer for this suggestion. We have added the time point "3 weeks" in Fig. 4a in the revised Figures.

"3. Fig 6a: a typical shape of exosome under transmission electron microscopy should be saucer shape instead of sphere shape."

Response: We have examined our pictures carefully and found that while some exosomes were indeed saucer-shaped (Fig. R3, denoted by blue arrow), others were sphere-shaped; the distinction may reflect the angle of view. If so, the lack of saucer-like shapes in the old Fig. 6a may be reflect the magnification used for the picture. In the revised manuscript, we added a blue arrow in the picture to indicate the saucer-like shape in the revised Fig. 6a. In addition, the sentence "Transmission electron microscopy showed exosomes displaying sphere-like morphology with an average diameter of ~ 100 nm (Fig. 6a)" was changed as "Transmission electron microscopy showed exosomes displaying saucer-like morphology with an average diameter of ~ 100 nm (Fig. 6a, denoted by blue arrow)" in page 17, paragraph 1, line 4 in the revised manuscript.

Fig. R3 The electron microscopy images of exosomes.

The representative images of exosomes with saucer-like shape were showed with blue arrow. Scale bar: 500 nm.

“4. In Figure 6c, 4T07 was injected through I.V. as labeled in figures. However, I.V. injection was not described in the “Animal Models” part in the “Materials and Methods” section.”

Response: The detail methods for I.V. injection as labeled in Fig. 6c were described in the “Isolation and application of exosomes” part in the old version of the manuscript. In the revised version of the manuscript, we have moved the IV injection description to the “Animal Models” subsection in the “Materials and Methods”.

Reviewer #2 (Remarks to the Author): with expertise in exosomes and metastasis

To the Authors:

In the manuscript by Mei-Yan Qi et al, the authors demonstrated that breast cancer stem cells expressing high Lin28B produce exosomes which contain low let-7s which in turn

Influence neutrophils to convert to a N2 phenotype and promote immune suppression of other immune cells, such as T cells, via PD-L2, which can then promote metastasis.

The work has great potential and has clinical significance. However, there are many questions that need to be addressed.

“ First, the work is a bit overwhelming and disconnecting. Data in one figure are often not carried over and further studied in other figures. For examples, the stemness of tumor cells, the chemokines listed in figure 2 (CXCL1, CXCL2, CXCL3, and CXCL5), then IL-6 and IL-10 chemokines are then introduced in Figure 3, the study of neutrophils, then the study of T cells and not so much neutrophils, the study of exosomes presented late in the manuscript, the role of PD-L2, etc. Thus, it is unclear if there is a focus.”

Response: We truly appreciate the insightful comments from this reviewer. As shown in Fig. 2, our data suggested that Lin28B expression promotes neutrophil recruitment in the pre-metastatic niche. Given that CXCL release leads to the recruitment of neutrophils, we present the results for CXCLs in Fig. 2. Similarly, Fig. 3 was used to show that Lin28B expression promotes neutrophil N2 conversion by IL-6 and IL-10, so we present the results of IL-6 and IL-10s levels in Fig. 3. IL-6 has been reported as an N2 inducer of neutrophils. By stimulating neutrophils with the cytokines in culture, we demonstrated that the combination of IL-6 and IL-10 results in a much stronger shift toward the N2 phenotype, compared to the effect of IL-6 alone (Extended Fig. 3e,f, Fig. 3d). Furthermore, by administering IL-6 and IL-10 intratracheally into 4T07-Control mice (Supplementary Fig. 3g), we found that, while either IL-6 or IL-10 drove neutrophil N2 conversion, the combination of IL-6 and IL-10 produced a

stronger effect close to that seen with Lin28B (Supplementary Fig. 3h); this effect was reversed by neutrophil depletion (Fig. 3j). Thus, our results indicated that IL-6- and IL-10-mediated neutrophil N2 conversion is a key mechanism promoting lung metastasis of BC in the context of Lin28B expression.

Per the reviewer's suggestion, we have reorganized Fig. 4 and Extended Fig. 4, moving the results that related to neutrophils from Extended Fig. 4 to Fig. 4. Thus, in the revised Fig. 4 we show the inhibitory properties of CD4⁺ and CD8⁺ T cells in the pre-metastatic lung (Fig. 4a, d and e), and that the modulation of those T cell repression is mediated by N2 neutrophils (Fig. 4b, c, f, g and h). The revised Fig. 4 emphasizes the central role of N2 neutrophils in T cell inhibition.

Additionally, we wish to point out that the data presented in Fig. 6 still focuses on the increase of PD-L2 expression in the neutrophils. As demonstrated in Fig. 6, our data showed that exosomes are essential for neutrophil N2 conversion (Fig. 6h) and for increased PD-L2 expression by neutrophils (Fig. 6i). We believe that PD-L2 expression by neutrophils is of great importance to the subsequent T cell inhibition. Treatment with anti-IL-6 and anti-PD-L2 suggested that the N2 conversion and subsequent PD-L2 expression are involved in T cell inhibition (Fig. 6k-m). Considered together, the results in Fig. 6 suggest that exosome education is the key mechanism for the establishment of the immune-suppressive pre-metastatic niche.

As summarized in the Introduction section, Lin28B is the master regulator of stem cell self-renewal and promotes BC progression by increasing tumor stemness. However, little is known about whether the increased stem characteristics also affect metastasis initiation, especially in the early pre-metastatic niche. In Fig. 1, we have provided the first evidence that Lin28B expression elevates tumor stem characteristics, as demonstrated in several BC models (Supplementary Fig. 1k, m-p); these results confirm the increased stemness property induced by the expression of Lin28B in breast tumors. This result also prompted us to hypothesize that the increased stemness property also may have an influence on metastasis, even in the early stages of tumor growth. Concordantly, we then demonstrated (in Fig. 7) that lower let-7 levels in exosomes generated by ALDH⁺ cells are the principal mechanism for establishment of the immune-suppressive pre-metastatic niche.

“ Moreover, there are publications on exosomes involving Let-7 in exosomes as a biomarker in plasma (Kentaro Matsuura, et al, Hepatology 2017) and tumor cells expressing LIN28A (Vanessa A. Enriquez et al, BioMed Research International 2015) in invasion, etc. The authors need to highlight these works and also highlight how their work is novel as compared to these other works.”

Response: We thank this reviewer for alerting us to these publications. We have highlighted these works in the revised Discussion section. Specifically, these papers now are cited on page 29, paragraph 2, line 1, and page 30, paragraph 2, line 2, respectively, in the revised manuscript.

“ There is little mechanistic studies here. For example conditional knockout tumors cells for LIN28A. And little understanding of Let-7 and other miRs in the exosomes and their biogenesis from tumor stem cells and what determines their uptake in cells, such as neutrophils, in pre-metastatic sites. Is there exosome uptake in other cells at pre-metastatic sites? Many details of the events taking place at pre-metastatic niches are incomplete.”

Response: We thank the reviewer for these comments. Construction of a Lin28B knockdown model will be key to defining the Lin28B-mediated mechanism affecting relevant immunological/metastatic phenotypes. We carefully checked the endogenous Lin28B levels in mouse BC cell lines 67NR, 168, 4T07, 4T1, E0771, EMT6 and NF639. Unfortunately, there was no apparent Lin28B expression in these cell lines (Fig. R4). Lin28B was abundantly expressed in human breast cancer cell line MDA-MB-231 (Fig. R4 and Supplementary Fig. 1I). However, MDA-MB-231 cells could be grown only in immune-deficient NOD/SCID mice, which lack T/B immune cells, so we are unable to survey the immune-regulative effect of Lin28B KO by MDA-MB-231 cells in immune-deficient mice. Here, we used *MMTV-Neu-Lin28B* transgenic mice (HER2⁺ subtype) to corroborate Lin28B's function upon neutrophil N2 conversion and immune suppression. With the mouse model, we obtained further evidence that Lin28B expression increased the lung metastasis occurrence of *MMTV-Neu* mice (Supplementary Fig. 1j). In addition, Lin28B promoted neutrophil recruitment (Supplementary Fig. 2i) and N2 conversion (Supplementary Fig. 3b) in the pre-metastatic lung in the mouse model. In turn, immune suppression appeared in the pre-metastatic niche in the situation of Lin28B expression (Supplementary Fig. 5h-j). Moreover, Lin28B expression resulted in the production of serum exosomes with lower let-7s content (Supplementary Fig. 8c), consistent with a pro-metastatic mechanism whereby Lin28B induces immune-suppressive pre-metastatic niche through neutrophil N2 conversion.

The novel mechanism that we revealed here is that Lin28B induces neutrophil N2 conversion and immune suppression in the pre-metastatic niche through exosomes with decreased let-7s content; this mechanism has not (to our knowledge) been proposed previously. However, in the present work we focused primarily on N2 neutrophils, immune inhibition, and exosomes, with little resolution on exosomes' biogenesis, secretion, and uptake by recipient cells. To date, little is known about the precise mechanism of exosome biogenesis and metabolism. It is generally clear that exosomes are endocytic in origin and can be uptaken by recipient cells by direct membrane fusion (PMID: 24959705). We proposed that similar mechanism also is employed in the situation of tumor cells. Whatever the mechanism, we cannot exclude the possibility that tumor cell- (especially tumor stem cell-) derived exosomes have unique stability and/or better affinity for recipient cells. In the present work, we focused primarily on neutrophil function, with less investigation of exosome biogenesis, stability, and uptake. We expect there to be great interest in dissecting the specific mechanism and characteristics of tumor exosomes, which will be helpful for exploring a novel means to influencing exosomal constituents and thus improving BC patients' prognosis.

In Supplementary Fig. 6b, c, the PKH-26 labeled exosome tracing assay showed that exosomes were uptaken mainly by neutrophils, macrophages, DCs, fibroblasts, and endothelial cells in the pre-metastatic lung.

Fig. R4 The amplification plot of *Lin28B* gene in different BC cell lines.

Lin28B was quantified by qRT-PCR in BC cell lines as indicated. Showing that *Lin28B* is highly expressed in human BC cell line MDA-MB-231, but is limiting in other mouse BC cell lines we have detected.

“ The work is highly dependent on neutrophils, yet there is no neutrophil images at pre-metastatic sites and surprisingly there is no work on NET formation at pre-metastatic niches as well.”

Response: We appreciate this reviewer’s comment. We performed IF staining to trace the neutrophil localization in the pre-metastatic sites using anti-Ly6G Ab, and the result is presented in Fig. 2d in the revised MS. As shown in Fig. 2e, we also have performed experiments to stain the pre-metastatic lung with anti-CD45, -CD11b, and -CD11c Abs and the results showed that Ly6G is well co-localized with CD45, CD11b, and CD11c (Fig. 2e), which is consistent with the FACS results indicating that most Ly6G cells were CD45⁺CD11b⁺Ly6C⁺ (Fig. 2b).

In response to this reviewer’s suggestion, we performed IF experiments to survey the neutrophil extracellular traps (NETs) formed by neutrophils in the pre-metastatic lung. The results showed that NETs were easily detected in the lung in tumor-bearing mice at the late stage (5-6 weeks), but less so at the early stage (2-3 weeks) (Fig. R5), accounting for the gradual accumulation of NETs with tumor growth. This result also indicated that NETs probably play a role in affecting lung metastasis in the late stage of tumor inoculation, but not in the early stage. Thus, NETs may not be involved in the immune-suppressive pre-metastatic niche formation in the early stage in the context of *Lin28B* expression.

Among the remaining neutrophils, some were isolated from the pre-metastatic lung by FACS. These cells appeared to be naïve neutrophils: when incubated with appropriate amounts of IL-6 and IL-10, these cells underwent N2 conversion, as may happen in the pre-metastatic lung site. We believe that all of the experiments that we designed show that Lin28B expression enables pre-metastatic lung inducing cytokine milieus that regulate neutrophil recruitment and N2 conversion, in which neutrophil N2 conversion is a key step for the “immune suppression of metastatic niche”.

Fig. R5 NETs staining in the lung neutrophils of 4T07 tumor-bearing mice. IF staining of NETs was performed in lungs of 4T07-Control and 4T07-Lin28B tumor-bearing mice at the early and late stage of tumor inoculation, respectively. Representative IF staining was shown. White arrows indicate the typical NETs structure, which were co-stained with H3cit, MPO and DAPI, (Ci-H3: NET, MPO: Neutrophils, DAPI: Nucleus). Scale bars, 100 μ m.

“In addition to targeting with anti-ly6G abs, the authors should use genetic models of neutrophil depletion to determine if these cells are key for exosome uptake and for further education of immune suppression at these sites”

Response: We agree with reviewer that a genetic model is the most desirable method to reveal the in vivo function of neutrophil depletion, rather than using the anti-Ly6G neutralizing Ab. We have asked the relevant labs for the genetic model of neutrophil depletion; however, no such neutrophil depletion mice strain is currently available. Instead, we used the anti-Gr-1 Ab, which also was used in selected experiments, to deplete the neutrophils in addition to the anti-Ly6G Ab (Fig. 6c). As anti-Gr-1 Ab recognizes both Ly6C and Ly6G antigens, we characterized the CD11b⁺Gr-1⁺ population in the pre-metastatic lung with Abs specific to Ly6C and Ly6G. The majority of the CD11b⁺Gr-1⁺ cells in the pre-metastatic lung were Ly6G⁺Ly6C⁺ neutrophils (Supplementary Fig. 6p), indicating the anti-Gr-1 Ab primarily targets the neutrophils.

To determine the role of neutrophils upon exosome-induced immune suppression, we depleted neutrophils by either anti-Ly6G or anti-Gr-1 Abs in exosome-educated mice (Fig. 6c). The

results indicated that neutrophil depletion diminished Lin28B-exosome induced lung metastasis (Fig. 6j). Moreover, neutrophil depletion counteracted the decreased infiltration of CD4⁺ and CD8⁺ T cells, as well as their reduced activation induced by Lin28B-exosome (Supplementary Fig. 6m-n). Meanwhile, neutrophil depletion also attenuated the inhibited Th1 differentiation of CD4⁺ T cells induced by Lin28B-exosome (Supplementary Fig. 6o), indicating lung neutrophils in Lin28B-exosome-educated mice were involved in immune-suppression and subsequent metastasis.

There are questions per figure:

“ Figure 1:

In a, provide an additional means of quantification, not just based on immunohistochemistry.”

Response: We thank the reviewer for this suggestion. We performed an additional experiment to detect Lin28B expression by western blotting in 24 pairs of TNBC tumor and para-tumor tissue (Supplementary Data 1). The results showed that Lin28B expression was increased in 11 cases and remained undetectable in 13 cases, confirming Lin28B was up-regulated in BC tumor tissues. This representative western blotting result is presented in Fig. 1c in the revised manuscript.

“ In h, there are no increases in primary tumor size yet in extended figure 1h, the authors show increases in proliferation which would likely lead to an increase tumor size. Explain.”

Response: Thank you for pointing this out. Indeed, although the over-expression of Lin28B did increase the proliferation of breast cancer cells in vitro (Supplementary Fig. 1n, corresponding to extended fig. 1h in the old version of the figures), over-expression of Lin28B did not increase the overall primary tumor size in vivo (Fig. 1i, j and Supplementary Fig. 1f, i). Actually, tumor cells in vivo are surrounded with a complex microenvironment, which is composed of myeloid cells, immune cells, stromal cells, endothelial cells, etc. The complex microenvironment also may have effects on the proliferation of tumor cells. For example, the release of EMT driver TGF- β into the tumor microenvironment may modulate EMT of tumor cells. EMT is known to be associated with increased cell migration, invasion, and decreased proliferation. Concordantly, our preliminary results showed that Lin28B expression in primary tumors induced EMT in 4T07 cells in vivo (Fig. R6), raising the possibility that Lin28B also regulates tumor cell proliferation through the tumor microenvironment. We postulate that Lin28B tumor may eventually show unchanged tumor size, as a result of the influence of Lin28B on the cellular microenvironment and the cell itself. Thus, we think it is not surprising that there was no increase in the tumor size with Lin28B expression.

Fig. R6 The expression of Lin28B promotes EMT transition in BC cells.

The 4T07 primary tumors were dissociated into single cells with enzyme digestion method, and then the 4T07-Lin28B or the control tumor cells were obtained by puromycin selection. The expression of EMT markers was examined by western blotting.

“ Explain the connection of Lin28B and stem cells, are the Lin28B tumor cells all tumor stem cells as well.”

Response: In 4T07 cells without Lin28B expression, there is a subset of cells retaining an ALDH⁺ property (Supplementary Fig. 1k), accounting for proteins other than Lin28B that serve to maintain stem identity. When we over-expressed Lin28B in 4T07 cells, the fraction of ALDH⁺ cells was increased from 19.7% to 50.5% (Supplementary Fig. 1k), suggesting that Lin28B expression may increase the stem cell fraction, without rendering all the cells ALDH⁺ stem cells. When we divided 4T07-Lin28B cells into ALDH⁺ and ALDH⁻ subsets, the ALDH⁺ cells expressed higher Lin28B, whereas the ALDH⁻ subset expressed lower Lin28B (Supplementary Fig. 8j), indicating the higher the expression of Lin28B, the higher the proportion of ALDH⁺ stem cells.

“ Figure 2:

In b, there are positive Ly6C cells, is the increase cells, also myeloid cells and/or dendritic cells, in addition to neutrophils are pre-metastatic sites.”

Response: Actually, the Ly6C expression (right panel) was analyzed on the gated CD11b⁺Ly6G⁺ cells (left panel) in Fig. 2b. Although dendritic cells and monocytes are also Ly6C⁺, those cells are negative for Ly6G. Therefore, DCs and monocytes were excluded from the gated CD11b⁺Ly6G⁺ population (Fig. 2b, left panel) and were not further analyzed for Ly6C expression. Thus, the CD11b⁺Ly6G⁺Ly6C⁺ cells represent neutrophils, but not monocytes or dendritic cells (Fig. 2b, right panel). In the revised MS, we added an arrow in Fig. 2b, to make clear that the Ly6C expression was analyzed on the gated CD11b⁺Ly6G⁺ cells.

“ In d, show stains for Ly6C, CD11b, and CD45 as well.”

Response: We performed additional experiments to stain the pre-metastatic lung with anti-CD45, -CD11b and -Ly6C Abs. The results showed that Ly6G was well co-localized with

CD45 and CD11b (Fig. 2e). In addition, most Ly6G⁺ cells also were co-localized with Ly6C, indicating that most of the Ly6G⁺ cells were CD45⁺ CD11b⁺ Ly6C⁺. Besides, only a portion of Ly6C⁺ cells were Ly6G⁺ (Fig. 2e), indicating Ly6C⁺ cells other than neutrophils also appeared in the pre-metastatic lung. In summary, the IF result was consistent with the FACS result that the Ly6G⁺ neutrophils were CD45⁺CD11b⁺Ly6C⁺. We have added a description of this result to the Results section in the revised MS. The added sentence is as follows: "Moreover, Ly6G was well co-localized with CD45, CD11b and Ly6C (Fig. 2e), consistent with the FACS results indicating that Ly6G⁺ neutrophils were CD45⁺CD11b⁺Ly6C⁺ (Fig. 2b)" (page 9, paragraph 1, line 1 in the revised manuscript).

" In e, the late anti-Ly6G with no affect, are there neutrophils present, show early vs late neutrophils staining and quantification, are there NETs formed and hence explains the failure of this therapy later on."

Response: We thank the reviewer for this great suggestion. We performed additional experiments to detect the staining and quantification of neutrophils in the later phase. By FACS, we determined that in the later stage, neutrophil proportion was decreased compared with that in the early stage (Supplementary Fig. 2I and Fig. 2C). Notably, Lin28B expression did not further increase neutrophil infiltration (Supplementary Fig. 2I). An IF experiment further confirmed this tendency (Supplementary Fig. 2m).

Moreover, we showed that NETs indeed were detected in lung neutrophils in tumor-bearing mice in the later stage, and (to a lesser degree) in the early stage (Fig. R5, please see page 20), suggesting that the failure of Ly6G Ab therapy may reflect the emergence of NETs. We have added a description of this result to the Results section in the revised MS. The added sentence is as follows: "Additionally, we showed that neutrophil extracellular traps (NETs), which are released by neutrophils to trap cancer cells to form distal metastasis²⁰, were detected in lung neutrophils in the late stage, but accumulated to lower levels in the early stage in the tumor-bearing mice (data not shown). This observation suggested that the failure of Ly6G Ab therapy also may reflect NET emergence" (page 9, paragraph 2, line 15 in the revised manuscript).

**" Figure 3:
Explain in more detail N2 conversion and why was G-CSF ignored here."**

Response: We thank the reviewer for this suggestion. We have explained N2 conversion in more detail and added the sentence, "Phenotypic conversion is considered to be the pivotal mechanism by which neutrophils modulate primary tumor growth²¹. Under certain circumstances, neutrophils can be polarized to the anti-tumor N1 phenotype or the pro-tumor N2 phenotype^{22, 23}. It is unknown whether a similar mechanism is recapitulated under our experimental conditions. We speculated that Lin28B not only triggers neutrophil recruitment, but also enables their phenotypic conversion, thus endowing neutrophils with a metastasis-promoting function" in page 10, paragraph 3, line 1 in the revised manuscript.

Cytokines IL-4, IL-6, and IL-10 can be readily measured in BALFs in our 4T07 orthotopic model (Fig. 3c). In contrast, the G-CSF content in BALFs was undetectable, indicating that high G-CSF content may not be an essential prerequisite here. Also, we believe that neutrophil phenotypic conversion also may be modulated by factors other than IL-10 and IL-6. We look forward to exploring, in future work, new cytokines that may be involved in N2 conversion.

“ **Figure 4:**

This figure and beyond has little work on neutrophils.”

Response: Here, we can observe that N2 neutrophils are associated with inhibitory proliferation, activation and Th1 differentiation of CD4⁺/CD8⁺ T cells. We believe that Fig. 4 shows N2 neutrophils have great effect on the immune-suppressive function of T cells in the pre-metastatic niche, which is of great importance for lung metastasis. Thus, we suggest that the data presented in Fig. 4 also are related to neutrophils. Some of the data shown in Supplementary Fig. 4 are related to neutrophils, and we think those data could be incorporated into Fig 4. To make it easier for the reader to understand the relation of these data to neutrophils, we reorganized Fig. 4 and Supplementary Fig. 4 in the revised MS.

“ **What is the role of NK cells here.”**

Response: Our data showed that the NK cell proportion in the pre-metastatic niche was not changed with Lin28B (Fig. R7a). In addition, the proportion of activated NK cells that are positive for CD69 also was not changed by Lin28B expression (Fig. R7b). Therefore, our preliminary data implied NK cell function is not modulated by Lin28B expression. In response to this reviewer’s suggestion, we have modified the text to reflect this point. The sentence, “In contrast, Lin28B expression did not alter either the proportion of NK cells or their activation in the pre-metastatic lung (data not shown), thus excluding the potential role of NK cells in the immune inhibition” was added on page 14 paragraph 1 line 5 in the revised manuscript.

Fig. R7 Lin28B expression does not increase the proportion and activity of NK cells in the pre-metastatic niche.

(a,b) The total lung cells from the pre-metastatic lung of tumor-bearing mice at 3 wks after tumor inoculation were subjected to the indicated antibody staining. Representative flow cytometry plots

(a) and proportion (b) of the total NK cells ($CD45^+NK1.1^+$) and its activated form ($CD45^+NK1.1^+CD69^+$) were shown ($n=3$). Data are presented as the mean \pm SEM of three independent experiments, and p-values were calculated using two-tailed Student's *t*-test. ns, no significance.

“ Why splenic T cells and not study T cells in the pre-metastatic sites.”

Response: Splenic naïve T cells were used in our vitro assays to determine the direct immune-suppressive effect of N2 neutrophils. We also analyzed T cells in the pre-metastatic niche to determine whether their numbers and phenotype would be changed by Lin28B expression (Fig. 4a, e and f). In general, we first analyzed the proportion and phenotype of T cells in the pre-metastatic niche and then conducted the in vitro assay using splenic T cells to detect whether the observed effect was directly elicited by N2 neutrophils. Therefore, we studied both splenic T cells and T cells in the pre-metastatic niche, with a major focus on the T cells in the pre-metastatic niche.

“ Figure 5:

PD-L2 can be expressed on DC and macrophages, further support that these cells are critical in this process.”

Response: We have determined that exhaustion-ligand PD-L2 was slightly raised by Lin28B on lung macrophages, whereas DCs were hardly affected (Fig. 5c). In turn, we purified macrophages from the pre-metastatic lung and determined that the macrophages isolated from lungs of 4T07-Lin28B mice indeed have function in inhibiting Th1 differentiation of $CD4^+$ T cells compared with the control (Fig. R8a, b), implying that macrophages also may associated with immune inhibition through PD-L2, and neutrophil N2 conversion is not the only mechanism involved in immune suppression in the pre-metastatic niche in the context of Lin28B expression. PD-L2 expression by DCs was not significantly changed, implying that PD-L2 is not an important mechanism for Lin28B-induced immune suppression. Whatever the mechanism, it will be an interesting project to study the roles of macrophages as well as DCs in the pre-metastatic niche involved in distal metastasis of breast cancer mediated by Lin28B.

Fig. R8 Macrophages from the pre-metastatic lung of 4T07-Lin28B tumor-bearing mice inhibit Th1 differentiation of $CD4^+$ T cells.

(a, b) The naïve $CD4^+$ T cells were stimulated under the Th1 condition in the presence of lung macrophages from the tumor-bearing mice. 4 days later, the cells were re-stimulated with PMA and

ionomycin for 6hr and Golgi-plugin for 4 hr. Expression of IFN- γ was analyzed by intracellular staining. Representative profile (a) and percentage (b) of IFN- γ ⁺ T cells were shown (n=3). Data are presented as the mean \pm SEM of three independent experiments, and p-values were calculated using two-tailed Student's *t*-test. *p < 0.05.

**“ Figure 6:
Exosome characterization for miRs need to be shown”**

Response: We thank the reviewer for this comment. In response to this reviewer’s suggestion, we conducted high-throughput sequencing of small RNAs in 4TO7 tumor exosomes to resolve the detailed components carried by these exosomes. The preliminary results showed that miRNAs accounted for more than 80% of the total small RNAs carried by the exosomes (Fig. 6d), consistent with a previous study showing that miRNAs are the main content of exosomes (PMID: 29335551). In addition, snRNAs, snoRNAs, rRNAs, tRNAs, and mRNAs also were differentially included (Fig. 6d). A total of 622 miRNAs were detected in tumor exosomes, with 253 miRNA present in reads of more than 50 in number (Supplementary Data 1). Furthermore, miR-16, miR-125b, let-7 family members, miR-21a, miR-29a, and miR-92a were found to be the most abundant miRNAs (Supplementary Data 1). Together, these data indicated that miRNAs are the most abundant content in the tumor exosomes. We have modified the text as necessary, including the addition of clarifying sentences in page 17 paragraph 2 line 1 in the revised manuscript.

“ Why tail vein tumor models here when orthotopic models used in previous figures.”

Response: In order to survey the pure exosome’s function, we first educated the lung tissue with exosomes, and tumor cells then were injected via tail vein to determine whether metastasis would arise. If we use an orthotopic model instead of the tail vein, tumor-released factors would have an additional influence on lung metastasis, and the results would not solely reflect exosome effects on metastasis initiation. A similar method also was reported in a previous study (PMID: 27505671). Thus, we consider this model to be the most appropriate available method to verify the exosomes’ function.

“ What cells in the Pre-metastatic sites take up exosomes?”

Response: As shown as part of the previous version of the MS, we performed this experiment: the results were presented in Supplementary Fig. 6b, c. In brief, PKH-26-labeled tumor-derived exosomes were injected intravenously into naive mice. Twenty-four hours later, the cell subsets that were positive for PKH-26 were determined by FACS (Supplementary Fig. 6b). The results revealed that approximately 3.82% of lung cells were positive for PKH-26 (Supplementary Fig. 6b). Exosomes were incorporated primarily by neutrophils, endothelial cells, DCs, macrophages, and fibroblasts (Supplementary Fig. 6b, c). In contrast to fibroblasts, the epithelial cells internalized fewer exosomes (Supplementary Fig. 6b, c). The CD45⁺CD11b⁻ cells, including T cells, B cells, and NK cells, showed little absorption of exosomes (data not shown).

“ Figure 8:

For the serum exosomes, what cells produce the exosomes in the serum.”

Response: The literature shows that exosomes are secreted by nearly all cell types under physiological and pathological conditions (PMID: 31288712). In addition, exosomes are found in different body fluids such as blood, plasma, urine, saliva, breast milk, bronchial lavage fluid, cerebral spinal fluid, amniotic fluid, and malignant ascites (PMID: 31466253). Thus, exosomes can be produced by all the cell types and presented in all the body fluids.

Here, we have modified the text in the Discussion section to emphasize the secretory characteristics of exosomes. The sentence was added as following: “The literature shows that exosomes are secreted by nearly all cell types⁴⁹” (page 29, paragraph 2, line 7 in the revised manuscript).

“ In b, let-7a differences are not impressive and likely not significant as shown.”

Response: The p values for let-7a, let-7d and let-7g are 0.0083, 0.0123 and 0.0072, respectively in Fig. 8b, so these effects are statistically significant.

“ Where is the neutrophil link.”

Response: Here, our main finding is the N2 neutrophils are the crucial mechanism for Lin28B induced lung metastasis. Specifically, exosomes were determined to be essential for N2 neutrophil formation, as the lower let-7s in exosomes contribute to increased CXCLs, IL-6, and IL-10 secretion in BALFs, leading to increased neutrophil recruitment and N2 conversion (Fig. 7c, d, e and Supplementary Fig. 8e, f). Hence, the exosomes with lower let-7s are the key point for Lin28B-induced N2 neutrophils and poor prognoses in mouse models. It is interesting to note that high Lin28B expression of human BC tumors also led to lower exosomal let-7s (Fig. 8d). Importantly, the lower let-7s also were correlated with poor prognosis and lung metastasis of BC patients (Fig. 8i-k, Supplementary Fig. 9l and Table 7), implying the use of a similar mechanism whereby Lin28B promotes human BC progression via lower exosomal let-7s. Given the difficulty of obtaining pre-metastatic human lung tissues, we do not know whether N2 neutrophils also appear in the pre-metastatic lung in human samples. Due to the direct action of lower exosomal let-7s upon increased CXCLs, IL-6, and IL-10 in BALFs (Fig. 7c, d and Supplementary Fig. 8e), we believe that N2 neutrophils also are recapitulated in the pre-metastatic lung in BC patients with lower exosomal let-7s.

In response to this reviewer’s comment, we have added a sentence to explicate the possible involvement of N2 neutrophils in the pre-metastatic lung in human BC patients. The sentences, “At present, we do not know whether N2 neutrophils also appear in the pre-metastatic lung in BC patients. Due to the direct action of lower exosomal let-7s upon increased CXCLs, IL-6, and IL-10 secretion in BALFs (Fig. 7c, d and Supplementary Fig. 8e), we believe that the conversion of N2 neutrophils also occurs in the pre-metastatic lung in BC patients with lower exosomal

let-7s. Together, our data strongly indicated a closer association of lower exosomal let-7s with poor prognosis and lung metastasis of BC patients" was added in page 25, paragraph 1, line 10 in the revised manuscript.

Reviewer #3 (Remarks to the Author): with expertise in metastasis, neutrophils, stem cells

Qi et al., provide data that expression of the embryonic stem cell factor Lin28B, is expressed predominantly in TN breast cancer and led to production of exosome which are inducing lung fibroblasts to release IL6 and IL10. These polarized neutrophils, the dominant myeloid cell type in the lung pre-metastatic niche, to an immunosuppressive phenotype. Mechanistically, the authors show that exosome from Lin28B induced this effect as they do not contain Let-7a, which is able to suppress IL6 and IL10 production.

The data are mostly convincing; however, they are not well presented. Sometime is difficult to understand what the message is and some data, especially at the end, about pSTAT3 are not clearly interpreted (see point 14). There are even data which are shown without any comment: the entire Supplementary Fig. 5 is not mentioned anywhere. Supplementary Fig. 5 data are important, they represent a validation of the finding from the 4T07 model using the spontaneous PyMT tumours model where Lin28B is overexpressed. The data confirm the immunosuppression effect on T cells in a completely different setting (Point 1)

Also, the text has several typos, for instance Extended is consistently spelled wrong. The text needs to be improved from that point of view as well. Additionally, I thought the format of Nature Communication was Supplementary Figures? Indeed, on one occasion, this is also used (Line 210: Figure S4G).

Importantly, a previous study highly relevant for this work is not properly cited and not discussed at all. As explained below, the authors need to reconcile their funding with the one from Liu, Y., et al., Cancer Cell 2016. All the references need to be revised to confirm they are accurate cited. (Point 12).

In summary, despite the text at time confusing, the data are overall convincing, and in my opinion the manuscript, upon review, can meet the Nature Communication standards.

Comments:

"1. The data about PyMT mice in Supplementary Fig. 1 and 5 need to be discussed more (in addition to be entirely mentioned). This validation means that the pro-tumourigenicity of the pre-metastatic niche in PyMT mice, which was previously characterized by a large bulk of studies, is increased upon Lin28B overexpression. Comparing the level of Lin28B with or without Lin28B overexpression is key to

important the changes here. Do PyMT exosome from control mice contain Let-7s or at least some? If not, the discussion needs to envisage other mechanism whereby Lin28B overexpression increase metastatic potential, maybe by changing the cancer cell themselves."

Response: We thank this reviewer for their excellent suggestion. To address this question, we performed additional experiments to test whether Lin28B was up-regulated in the *PyMT* mice. First, we performed a western blotting assay and determined that Lin28B was readily detected in *PyMT-Lin28B* mice, but not in the control mice (Supplementary Fig. 1e). Second, increased neutrophil accumulation and N2 conversion also were found in the pre-metastatic lung in *PyMT-Lin28B* mice (Supplementary Fig. 2i, 3a), indicating that Lin28B induces the neutrophil N2 phenotype. Third, immune suppression was recapitulated in the lung of mice inoculated with exosomes derived from *PyMT-Lin28B* tumors (Supplementary Fig. 6i). Fourth, low let-7 content was observed in serum exosomes recovered from *PyMT-Lin28B* mice (Supplementary Fig. 7b), in contrast to those obtained from the control mice. Collectively, our data from the *PyMT* mice supported the notion that Lin28B enhances the pro-tumorigenicity of the pre-metastatic niche through the production of exosomes with decreased let-7a content. Thus, our data indicated that Lin28B expression indeed promotes lung metastasis in *PyMT* mice.

"2. In relation to this, Supplementary Fig. 8i and Fig. 7h,i show that ALDH+ 4TO7-Lin28B cells express increased Lin28B and they release exosomes with much lower let-7s levels responsible for increase metastases. Does Lin28B expression increase the number of ALDH+ cells as well? Data from the PyMT model would argue this could be the case (Supplementary Fig. 1g). What are the ALDH+ cancer cells in the 4TO7-control vs 4TO7-Lin28B? Are the ALDH+ cancer the only source of Let-7s depleted exosome? The authors also need to discuss that changing in the cancer property them self can contribute to the enhanced metastatic activity."

Response: Lin28B expression indeed increased the proportion of ALDH⁺ 4TO7 cells (Supplementary Fig. 1k). Per this reviewer's suggestion, we performed additional experiments in the *PyMT* model to verify the effect of Lin28B on BC cell stem characteristics. Given that *PyMT-Lin28B* tumor cells contain endogenous EGFP (Supplementary Fig. 1d), we could not use the ALDHFLUOR™ kit (STEMCELL inc) to evaluate the proportion of ALDH⁺ cell. Instead, we measured stem gene expression in *PyMT* tumors by qRT-PCR. As shown in Supplementary Fig. 1o, Lin28B expression significantly up-regulated expression of the stem-related genes *Sox2*, *Klf4*, *c-Myc*, and *Nanog*, indicating increased stem characteristics are mediated by Lin28B in *PyMT* model. Moreover, we knocked down Lin28B in human breast cancer MDA-MB-231 cells (Supplementary Fig. 1l) and the results showed that Lin28B knockdown decreased the proportion of ALDH⁺ cells (Supplementary Fig. 1m), confirming that Lin28B expression acts to increase the proportion of ALDH⁺ BC cells.

In breast carcinomas, aldehyde dehydrogenase (ALDH) has been used as a marker to identify the BC stem cells possessing higher tumor-initiating capacity (TIC) (PMID: 24981741). In 4TO7-Control and 4TO7-Lin28B cells, the common feature of ALDH⁺ cells should be that they

have TIC. In 4TO7-Lin28B cells, the higher level of Lin28B is expected to convert a fraction of the original ALDH⁻ cells into ALDH⁺ cells. In addition, the original ALDH⁺ cells expressed more Lin28B in this case. As a result, the ALDH⁺ subset was expanded with Lin28B expression (Supplementary Fig. 1k). Importantly, the ALDH⁺ subset contained higher Lin28B levels (Supplementary Fig. 8j), resulting in turn with decreased exosomal let-7s than obtained without Lin28B expression. Therefore, although the ALDH⁺ subsets of both 4TO7-Control and 4TO7-Lin28B cells both possess TIC (PMID: 18371393), the ALDH⁺ subset in 4TO7-Lin28B cells (given Lin28B expression) should contain much less let-7s, resulting in turn in the acquisition of a stronger capability to induce the immune-suppressive pre-metastatic niche (Fig. 7e-g and Supplementary Fig. 7f, h, i) and lung metastasis (Fig. 7h, j).

Let-7s also are regulated by other signal pathway(s) or transcription factor(s) (PMID: 26962302, 31413295 and 21903590). For example, Myc protein inhibits the transcription (in HepG2 cells) of the microRNA cluster MC-let-7a-1~let-7d via a noncanonical E-box (PMID: 21903590). Therefore, let-7s also may be regulated by factors other than Lin28B. Thus, ALDH⁺ cancer cells are not the only source of let-7-depleted exosome.

In addition to the decreased exosomal let-7s (Fig. 7a, b and Supplementary Fig. 8a-c), Lin28B expression also can potentiate the stem characteristics of BC cells (Supplementary Fig. 1k, m-p), indicating that increased stem characteristic must be an important mechanism for enhanced metastatic activity when the tumor cells have arrived at lung tissues. In the present manuscript, we focused primarily on the role of Lin28B in the pre-metastatic stage when the most tumor cells have not yet reached the lung tissues, thus excluding the direct influence of cancer stem cell on the occurrence of metastasis. In response to this reviewer's suggestion, we added the sentence, "At the same time, we cannot exclude the possibility that the increased stem characteristics induced by Lin28B are an important mechanism for enhanced metastatic activity when the tumor cells arrive in lung tissues", to the Discussion section (page 32, paragraph 1, line 9 in the revised manuscript).

"3. Figure 1f, if Lin28B expression is higher in TN as TN have a worst prognosis, the inverse correlation between Lin28B expression and overall post-surgery survival could just be a consequence of this; does this hold true among TN only? While this does not impact on the relevance of Lin28B expression and its mechanism, it is an important difference to clarify."

Response: We thank this reviewer for their comment. To address this issue, we performed a new experiment to detect the expression of Lin28B in an additional tumor tissue microarray (TMA) (Cohort III, 145 cases) (Supplementary Data 1). Combining these data with the results from TMA Cohort II, we showed that high Lin28B expression was clearly associated with poor prognosis and was an independent prognosticator of the survival of patient with BC of the TNBC subtype (Fig. 1h and Table S5). Moreover, consistent correlation also was found in luminal and HER2⁺ subtypes (Fig. 1h and Table S3, S4). Taken together, these results indicated that Lin28B is associated with poor prognosis of patients with BC of all BC subtypes, not only TNBC.

Nonetheless, as presented in Fig. 1d, the expression of Lin28B was highest in BC of the TNBC subtype. Therefore, we believe that studying the metastasis-promoting mechanism of Lin28B will be of particular significance for the prevention and treatment of TNBC progression.

“4. Fig 2b,c and Extended Data Fig 2e,f the increase of Mac seems very similar to the one of neutrophils, but of course the absolute number of neutrophils is much higher. The text needs to describe the data accordingly.”

Response: We thank this reviewer for pointing out this observation. Indeed, the macrophages also were recruited in the pre-metastatic niche along with Lin28B expression, but the absolute number was much lower than neutrophils (Supplementary Fig. 2e, f). In response to this reviewer’s suggestion, the sentence “whereas F4/80⁺ macrophages only modestly accumulated” was modified to “F4/80⁺ macrophages accumulated similarly, but to absolute numbers that were much lower than those of neutrophils” in page 8, paragraph 2, line 4 in the revised manuscript

“5. It is unclear what the sentence in line 128-131 means. Myeloid-derived suppressor cells are a functional definition, I personally do not think a particular function of immunosuppression could be deduced by the presence of Ly6C. There is no rule preventing Ly6C⁺ neutrophils to act as immune-suppressive cells (as this very work shows). I appreciate that the authors stay away from this confusing terminology and, for consistency, I would suggest removing this sentence as well. However, it is interesting to see the decrease in Ly6C expression level in premetastatic lungs, but this occurs in both control and Lin28B expressing cells. A comment is required here, as 4T07 control tumour inducing pre-metastatic changes as well.”

Response: We agree with this reviewer. Therefore, we have removed the sentence about myeloid-derived suppressor cells (MDSCs) in page 9, paragraph 1, line 3 in the revised manuscript.

“6. Fig 2g, the text mentioned that this is a time point of early stage it is not clear if this is at the time before tumour resection, day 35? It should be clear at what point of the schematic in 2e is referring to. Moreover, I would be surprised if the Ly6G antibody can go on depleting neutrophils much longer than 14 days, especially in BALB/c mice, which have a high number of neutrophils. In other words, I doubt that after tumour resection the treatment was still efficient. Unless the Early phase and Late phase are alternative settings, but this should be clear. Finally, if Early and Late are independent settings, the data should for both be shown, at least in the supplementary data. The Late setting should include the control as well as there could be onset of metastasis in absence of Lin28B triggered by surgery, in which case, it could still be neutrophils dependent.”

Response: The Fig. 2i data (corresponding Fig. 2g in the old version of the figures) was from

early neutrophil depletion, which began on Day 14 post-tumor inoculation and continued until just before tumor resection (denoted by blue arrow in Fig. 2f). In contrast, the late neutrophil depletion started on Day 35, just after primary tumors were removed, and continued for the consecutive 6 time points (denoted by sky blue arrow in Fig. 2f). Thus, the early phase and late phase are independent settings. To clarify the distinction in time points between the early and late phases of neutrophil depletion, we added a methods description in the "Animal model" subsection of the "Materials and Methods". In addition, the time points now also are explained in the corresponding figure legend section. The early neutrophil depletion (denoted by a blue arrow in Fig. 2f) was sufficient to inhibit lung metastasis in 4TO7-Lin28B mice (Fig. 2h). However, a similar metastasis-inhibiting effect was not found with late neutrophil depletion (Supplementary Fig. 2k), indicating that the early neutrophils, but not the late neutrophils, are involved in metastasis-promoting. In response to this reviewer's suggestion, we added the late setting of lung metastasis of 4TO7-Lin28B mice in Supplementary Fig. 2k of the revised manuscript.

Tumor resection at the late stage, which often is accompanied by neutrophil decreases (PMID: 25822788), can lead to metastasis, suggesting the potential metastasis-inhibitory function of neutrophils in the late stage of tumor growth. However, we did not observe an onset of lung metastasis following neutrophil depletion in 4TO7-Control mice in the late phase (Supplementary Fig. 2n), implying that both the removal of the harmful neutrophils, and the metastatic ability of tumor cells, are essential for the ultimate metastasis.

"7. Why Fig 3a say "lung" if the RNA is from isolated neutrophils?"

Response: We thank this reviewer for noting this discrepancy. We apologize for this mistake and have revised "lung" to "neutrophil" in Fig. 3a.

"8. The sentence in line 221-225 is a copy of the previous sentence (217-221) but it refers to the wrong figure (4e,f). Also, Fig 4f is not there."

Response: We thank the reviewer for noting this error. Indeed, these two sentences were repeated, and we have deleted the second sentence in the revised MS. Here, we intended to refer to Fig. 4c, d, but not to Fig. 4e, f, in the old version of the MS. In response to the comment of another reviewer, the reference to Fig. 4c, d was changed to Fig. 4d in the revised MS.

"9. Supplementary Fig. 4h-j, it is not clear what Control and N2 neutrophils refers too, are the pre-metastatic neutrophils from 4TO7 (Control) or 4TO7- Lin28B (N2)? If so it should clear."

Response: We thank this reviewer for this insightful comment. Indeed, the Control neu are the lung neutrophils from the pre-metastatic lung of 4TO7-Control mice and the N2 neu are the lung neutrophils from the pre-metastatic lung of 4TO7-Lin28B mice. We have clarified details of the two kinds of neutrophils in the revised manuscript. The sentence "As expected, pre-metastatic neutrophils isolated from 4TO7-Lin28B mice (N2 neu) suppressed proliferation

of both CD4⁺ and CD8⁺ T cells, whereas a similar effect was not found with control neutrophil exposure isolated from 4T07-Control mice (control neu)” was added on page 12, paragraph 4, line 2 in the revised manuscript. In addition, at the request of another reviewer, the Supplementary Fig. 4h-j in the old version of the manuscript was moved to Fig. 4f-h in the revised manuscript.

“10. Line 242, in Fig. 5a for PD-L1 should state unchanged levels rather than slightly upregulated, more in line with the same as Fig 5b for CD86, OX40L, and ICOSL showing similar results.”

Response: We agree with this reviewer’s comment. We have corrected the sentence as “In contrast, PD-L1 was unchanged with Lin28B expression (Fig. 5a)” on page 15, paragraph 2, line 5 of the revised manuscript.

“11. Line 263, please refer to the Fig 3b-c where the levels of IL12a are shown.”

Response: We have referred Fig. 3b, c in this sentence, which is shown in page 16, paragraph 1, line 8 in the revised manuscript.

“12. Line 314-316, the interpretation of the data shown in Supplementary Fig. 7a-c is incorrect: the data in Supplementary Fig. 7a showed that both fibroblasts and epithelial cells are expressing CXCLs, but Supplementary Fig. 7b-c shows that only Fibroblasts respond with a higher CXCLs production when 4T07-Lin28B cells are used compared to 4T07-control. Epithelial cells do not show an increase in CXCLs in response to 4T07-Lin28B vs Control.”

Response: We thank this reviewer for pointing out this error. Indeed, the interpretation of the data shown in Supplementary Fig. 7a-c is incorrect. In fact, the result in Supplementary Fig. 7a was from 4T07-Control mice, and showed that both fibroblasts and epithelial cells are the main cells expressing CXCLs, in contrast to the myeloid cells. However, Lin28B expression led primarily to increased CXCL production in fibroblasts, but not in epithelial cells (Supplementary Fig. 7b, c), indicating specific up-regulation of CXCLs in fibroblasts under Lin28B expression conditions. To address this issue, we added text to clarify that the results in Supplementary Fig. 7a are from 4T07-Control mice. The sentences “In the pre-metastatic lung of the 4T07-Control mice, CXCLs were detected mostly in stromal cells (Supplementary Fig. 7a). However, the stromal cells showed a distinct response to Lin28B expression. Lin28B increased CXCL production in fibroblasts compared to the control (Supplementary Fig. 7b, c). In contrast, epithelial cells only slightly upregulated chemokines by Lin28B expression (Supplementary Fig. 7b, c)” were added on page 20, paragraph 2, line 1 in the revised manuscript.

Here, we also revised the Figure Legend of Supplementary Fig. 7a to indicate that the experiment was conducted in 4T07-Control mice; we expect that this revision will make this point better understood by the reader.

“ Having this interpretation right is essential as it can reconcile the data presented in this manuscript with a previous study showing that exosome target lung epithelial cells and activate a pre-metastatic niche via TLR3 dependent neutrophils recruitment (Liu, Y., et al., Cancer Cell (2016).

<http://doi.org/10.1016/j.ccell.2016.06.021>).

Moreover, due to the similarity of this work to the finding of the above-mentioned study, the authors need to discuss it. Conversely, Liu, Y., et al., is mentioned once in the introduction and cited in a rather strange way, to support the statement of bone marrow-derived cells composition. It is also cited in the M&M to refer to exosome isolation. Liu, Y., et al., and the present manuscript both present data about an exosome-dependent pre-metastatic neutrophils modulation, therefore a detailed analysis of the potential differences in the data is needed.”

Response: We agree this reviewer’s point: we need a detailed analysis to interpret the difference between our data and those of the study of Liu et al., given the similarity of our work to that of Liu et al. The main difference between the results presented by Liu et al. and ourselves is the exosome-receiving cells in the pre-metastatic lung. In response to this reviewer’s suggestion, we have performed analysis and interpretation, which was added to the Discussion section (page 31, paragraph 2, line 1). Specifically, the following text was added: “A previous study showed that lung epithelial cells were the main cells responsible for tumor exosome absorption, CXCL secretion, and neutrophil recruitment¹². In contrast, our data revealed that fibroblasts were the main cells responsible for exosome receiving, thus permitting increased CXCL induction in the context of Lin28B expression. The main difference between our data and that of Liu et al. was the exosome-receiving cells. The differences presumably reflect the use of distinct tumor models. Liu et al. performed subcutaneous injection with Lewis lung carcinoma (LLC) cells, an epithelium-like mouse cell line, while we performed orthotopic mammary implantation with mesenchymal-like 4TO7 cells. We propose that fibroblasts may be more sensitive to 4TO7 cell-derived exosomes. For example, mesenchyme-like 4TO7 cell-derived exosomes may carry specific components that can be recognized more effectively by lung fibroblasts than by epithelial cells. Moreover, it also is possible that lung fibroblasts may be activated specifically by 4TO7 tumor-derived soluble factors before exosome arrival, thus potentiating their ability to receive exosomes. In turn, the different model may lead to fibroblast and epithelial cells with different efficiencies to receive exosomes, thus leading to the distinct responses. These differences constitute an interesting phenomenon; further validation by other tumor models will be needed”.

In addition, we note that we had originally cited Liu et al. in the Introduction and M&M sections, citations that are retained in the revised MS. However, we realize that the most important citation should be in the Discussion, where the difference of the exosome-receiving cells presented in our data and those of Liu et al. is now noted.

“13. Line 323, I guess it means IL10 only not IL6, IL10.”

Response: Yes, the intent was to refer to IL-10 only, and not to both IL-6 and IL-10. To clarify

this point, the sentence "In contrast to IL-6, IL-10 was mostly expressed in macrophages (Supplementary Fig. 7e, g)" was corrected as "In contrast, IL-10 was expressed primarily in macrophages (Supplementary Fig. 7e, g)" on page 20, paragraph 2, line 11 in the revised manuscript.

"14. The data about STAT3 signalling are not clear to me. Supplementary Fig. 9c shows how that the STAT3 pathway inhibitor S3I-201 increases CXCLs in cells treated with both Control and Lin28B-exosome. There is no difference between Control and Lin28B-exosome, why the text links this effect to the Let-7s deficiency? It is unclear from the text what the interpretation is."

Response: We thank reviewer for this comment. In fact, this result was an error on our part, for which we sincerely. To ensure the accuracy of this point, we repeated this experiment three times and confirmed that Lin28B-exosome raised CXCL production in 3T3 fibroblasts (Supplementary Fig. 9c). Importantly, STAT3 pathway inhibitor S3I-201 attenuated the production of CXCLs induced by Lin28B-exosome, and thus blunted the difference between cells treated with Lin28B-exosome and the control (Supplementary Fig. 9c). In contrast, S3I-201 did not significantly inhibit CXCL production in cells treated with Control-exosomes (Supplementary Fig. 9c), accounting for the lower STAT3 pathway activity in cells treated with Control-exosomes (Supplementary Fig. 9b). We have corrected the image in Supplementary Fig. 9c in the revised manuscript.

"What would the level of IL16 and IL10 expression be in this STAT3 inhibitor experiment?"

Response: We thank reviewer for the good question. In response to this reviewer's suggestion, we verified the effect of STAT3 inhibitor S3I-201 upon IL-6 and IL-10 expression in 3T3 cells. 3T3 cells were first treated with STAT3 inhibitor S3I-201 for 1h and then were treated with Control-exosome or Lin28B-exosome. 24 hours later, the supernatant was subjected to ELISA assay for IL-6 and IL-10 detection. The results showed that Lin28B-exosome did not significantly increase IL-6 and IL-10 expression compared with the control (Supplementary Fig. 9i). In addition, S3I-201 did not significantly inhibit IL-6 and IL-10 production (Supplementary Fig. 9i). These results indicated that the STAT3 pathway has no influence upon IL-6 and IL-10 up-regulation in Lin28B-exosome-treated 3T3 cells. The basal levels of both IL-6 and IL-10 were lower in fibroblasts than in neutrophils and macrophages (Supplementary Fig. 7f, g); thus, the effect of attenuated target gene inhibition exerted by lower let-7s in Lin28B-exosome may be negligible in 3T3 cells. In addition, the lower basal levels of IL-6 and IL-10 in fibroblasts suggest that their expression is tightly controlled at the transcriptional level; this inference implies that the up-regulation of these cytokines would require a longer interval of stimulation and/or specific conditions. Moreover, it also is possible that the expression of IL-6 and IL-10 are controlled in a complex manner in 3T3 cells, such that STAT3 activation is not sufficient for their up-regulation. Together, our preliminary results showed that while Lin28B-exosome can raise CXCL production in 3T3 cells via STAT3 (Supplementary Fig. 9c), the corresponding effect on IL-6 and IL-10 expression was weaker (Supplementary Fig. 9i).

In response to this reviewer's suggestion, we have added sentences in the Result section in the revised manuscript. The added sentence is as following: "We noticed that Lin28B-exosome did not increase IL-6 and IL-10 expression compared with the control in 3T3 cells (Supplementary Fig. 9i). In addition, S3I-201 did not inhibit IL-6 and IL-10 production (Supplementary Fig. 9i), indicating that low let-7s in exosomes and the STAT3 pathway have no influence upon IL-6 and IL-10 up-regulation in 3T3 cells" (page 23, paragraph 1, line 10 in the revised manuscript).

" More generally, CXCL1 is a STAT3 target, why STAT3 inhibitor would increase CXCL1? Also Lin28B-exosome in the panel before increase pSTAT3, how Let-2a affect pSTAT3? What is in the authors interpretation about the cascade of events following exosome exposure, pSTAT3, CXCLs, ILs expression and when Let-2a inhibition act? Neutrophils modulation is due to ILs signalling or they act as paracrine? What cause what in the cascade of event?"

Response: As noted above, due to our negligence, we used the wrong picture in Supplementary Fig. 9c in the previous version of the manuscript. In the revised MS, we have replaced the wrong picture with the correct one. Indeed, as expected, the STAT3 inhibitor inhibited CXCLs production in Lin28B-exosome treated 3T3 cells (Supplementary Fig. 9c).

We did find that Lin28B-exosome up-regulated STAT3 activity and that this effect was counteracted by let-7a mimics (Supplementary Fig. 9b), indicating that the deficiency of let-7a in Lin28B-exosome is responsible for increased STAT3 activity. However, it is still unclear how low let-7s in Lin28B-exosome lead to elevated STAT3 activity. In the present work, we only found the causal relationship between low let-7a in Lin28B-exosome and the higher STAT3 activity, but the precise mechanism remains unknown.

Regarding the signal cascade, our analysis is as follows:

- (1) Lin28B-exosome heightened CXCLs production in 3T3 cells, an effect was that was abolished by let-7a mimics (Supplementary Fig. 9a), indicating that the action axis is Lin28B-exosome→let-7s→CXCLs.
- (2) Lin28B-exosome up-regulated STAT3 phosphorylation and this effect was relieved by let-7a mimics (Supplementary Fig. 9b), indicating that the action axis is Lin28B-exosome→let-7s→STAT3.
- (3) The STAT3 pathway inhibitor S3I-201 was found to blunt the increased CXCLs induced by Lin28B-exosome (Supplementary Fig. 9c), indicating that the action axis is Lin28B-exosome→STAT3→CXCLs.

Taken together, we concluded the overall action axis is Lin28B-exosome→let-7s→STAT3→CXCLs.

We proposed the following model: when Lin28B-exosome (which contain decreased let-7s) enter into the cells, the Lin28B-exosome show decreased ability to inhibit STAT3 or other factor(s) involved in STAT3 pathway regulation, thus resulting in enhanced STAT3 signal pathway activity. The higher STAT3 activity then acts to up-regulate CXCL expression.

To survey whether ILs or a downstream pathway are involved in IL production in neutrophils, we incubated naïve neutrophils with IL-6, IL-10 or CCL2 for 4 hours. The cells then were harvested for qRT-PCR assay. The results determined that IL-6 or IL-10 stimulation resulted in decreased *IL-6* mRNA expression (Supplementary Fig. 9j). In contrast, the control cytokine CCL2 did not significantly affect *IL-6* or *IL-10* expression (Supplementary Fig. 9j). The results suggested that the secreted IL-6 and IL-10 act to ensure that intracellular IL-6 levels do not become too high. The results also suggested that low let-7s in Lin28B-exosome, but not the secreted IL-6 or IL-10, may be the principle mechanism for increased IL-6 production in neutrophils stimulated with Lin28B-exosome (Supplementary Fig. 9d and 9j). In response to this reviewer's suggestion, we have added sentences in the Result section to clarify this result. The added sentence is as following: "Moreover, we found that IL-6 or IL-10 stimulation all decreased *IL-6* mRNA expression in neutrophils (Supplementary Fig. 9j). In contrast, the control CCL2 did not affect (Supplementary Fig. 9j). This result indicated that *IL-6* expression was negatively regulated by the IL-6 and IL-10 pathways in neutrophils, ensuring that the intracellular IL-6 levels do not become too high. This result also suggests that low let-7s in Lin28B-exosomes, but not the IL-6 or IL-10, may be the principle mechanism for increased IL-6 production in neutrophils (Supplementary Fig. 9d and 9j)" (page 23, paragraph 1, line 14 in the revised manuscript).

" The final model proposed is not entirely clear. Using a final drawing to explain the model would help the authors explain?"

Response: We now have incorporated a final schematic/mechanistic drawing to explain the proposed model, as presented in Supplementary Fig. 9m.

Reviewer #4 (Remarks to the Author): with expertise in breast cancer and stem cells

In this study, Mei-Yan Qi and collaborators indicate that lin28b induces breast cancer lung metastasis by producing immune-suppressive pre-metastatic niches and acting by suppressing the let-7 amount in exosomes of cancer cells.

In principle this is a very interesting study.

" My main concern about this study is that most of the results are based on a single mouse breast cancer line used as control or that overexpresses Lin28b. Additional cell lines should be used, and including relevant model of human breast cancer with high level of Lin28b expression in immunocompromised mice."

Response: We thank this reviewer for the valuable comments. We performed the experiments using only the 4T07 cell line because of the challenge we experienced in identifying a mouse cell line that can disseminate to the distal organs but cannot produce visible metastatic lesions, therefore providing a model for our hypothesis. We had detected Lin28B expression in multiple

mouse breast cancer cell lines that were available to us (67NR, 168, 4TO7, 4T1, EMT6, E0771 and NF639) and Lin28B is not expressed in any of those mouse cell lines (Fig. R4, please see page 10). Therefore, the effect of Lin28B knockdown cannot be verified in these mouse cell lines at present.

The human breast cancer cell line MDA-MB-231, which is invasive and highly metastatic, expresses high levels of Lin28B (Fig. R4 and Supplementary Fig. 1l). However, due to immune rejection, this human breast cancer cell line can be used to produce tumors and induce metastasis only in immunodeficient animal hosts (e.g., NOD/SCID mice); however, the immunodeficient nature of the NOD-SCID mouse precludes studying the immune-regulatory function of Lin28B. This observation explains why we did not use a human breast cancer cell model in this manuscript.

To further confirm Lin28B's function in BC metastasis, we employed *MMTV-Neu* (a HER2-positive mouse BC model) with *Lin28B^{Kl}* mice. *MMTV-Neu-Lin28B* mice were generated by crossing the *MMTV-Neu* mouse expressing an activated rat *c-Neu* oncogene with *MMTV-Cre* and *Lin28B^{Kl}* mice. The results showed that Lin28B over-expression increased the lung metastatic incidence of *MMTV-Neu* mice (Supplementary Fig. 1j). In addition, Lin28B over-expression increased the proportions of the lung neutrophils (Supplementary Fig. 2i), notably including those displaying N2 phenotype (Supplementary Fig. 3b), and induced an immune-suppressive pre-metastatic niche (Supplementary Fig. 5h-j). Moreover, Lin28B expression resulted in the production of the serum exosomes containing lower let-7s in the tumor-bearing mice (Supplementary Fig. 8c), supporting the pro-metastatic mechanism whereby Lin28B induces an immune-suppressive pre-metastatic niche through neutrophil N2 conversion.

Using the *PyMT* BC model, we performed additional experiments and obtained further results as follows: Firstly, Lin28B expression induced neutrophil accumulation in the pre-metastatic niche (Supplementary Fig. 2i). Secondly, Lin28B expression also induced neutrophil N2 conversion in the pre-metastatic niche (Supplementary Fig. 3a). Thirdly, let-7 levels were decreased in the serum exosomes in *PyMT-Lin28* mice (Fig. 7b). Importantly, the *PyMT-Lin28B-exosome* education permitted the establishment of an immune-suppressive lung microenvironment (Supplementary Fig. 6i). These results are consistent with the results from the 4TO7 and *MMTV-Neu* BC models.

Due to lack of a suitable mouse BC cell line that expresses a higher level of Lin28B, we cannot currently verify the effect of Lin28B knockdown on immune suppression using an in vivo model. However, using the mouse BC models (4TO7, PyMT and MMTV-Neu), we obtained consistent results in support of the hypothesis that Lin28B induces immune suppression through N2 neutrophils and exosomes with lower let-7 levels. In the near future, we hope to identify a mouse BC cell line with high endogenous Lin28B expression, which would permit the performance of the critical relevant experiments proposed by the reviewers.

“ In Supplementary Fig. 1 the over-expression of Lin28b in 4TO7 is massive

compared to the control cells which can indicate that results seen only derive from this artificial lin28b expression, but this never happens in real breast cancers.

There are not Lin28b loss-of-function experiments in the study to demonstrate that all results shown are not only the product of such high ectopic expression of Lin28b.”

Response: We thank this reviewer for their comments. We inferred that the massive band of Lin28B may reflect differences in sample loading and/or longer exposure time. Hence, we repeated this experiment with reduced total protein loading and were able to determine that Lin28B was properly detected in 4TO7 cells (Supplementary Fig. 1a). Importantly, the expression level of Lin28B in 4TO7 cells was lower than that found in MDA-MB-231 cells (Fig. R9). This observation implies that the level of Lin28B expression seen in 4TO7 cells could happen in real breast cancer cells.

As we note above, we currently lack a suitable mouse BC cell line that expresses high levels of Lin28B, given that the scientists have collected a relatively low number of mouse mammary breast cancer cell line in this century (compared to the number of corresponding human cell lines). As described above, we have measured Lin28B expression in *PyMT* and *MMTV-Neu* transgenic mice, which are commonly used mouse breast mouse models. The results demonstrated that Lin28B was well expressed in these tumors (Supplementary Fig. 1e, h). Hence, we used several distinct mouse models (4TO7, PyMT and MMTV-Neu) to corroborate the function and mechanism of Lin28B. We believe that these data are reliable evidence in support of the hypothesis that Lin28B induces immune suppression through N2 neutrophils and exosomes with decreased let-7 content.

Fig. R9 The expression level of Lin28B in 4TO7 cells is lower than that found in MDA-MB-231 cells.

The expression of Lin28B was analyzed by western blotting in 4TO7-Control (line 1), 4TO7-Lin28B (line 2) and MDA-MB-231 cells (line 3) cells. β-actin served as a loading control.

“ On the other end in the TCGA database, the level of Lin28b mRNA expression in breast cancer is very low and only limited to few breast cancer specimens. This in principle excludes that Lin28b is a major driver of breast cancer metastasis as proposed by the authors.

How can the authors explain this contradiction?”

Response: We thank this reviewer for these comments. We suggest that it is important to analyze the protein expression level in these TCGA cohorts because it is possible that the

Lin28B protein level was higher in more lung metastatic clinic sample in these cohorts. Lin28B frequently is up-regulated in different human tumor types including breast cancer (PMID: 19483683). In our IHC assays, Lin28B was readily detected in human breast cancer samples, such that expression of the protein was up-regulated in 11 out of 24 cases, while remaining undetectable in 13 cases (Fig. 1c). Moreover, by analyzing the transcriptomic datasets of a clinical cohort of 204 patients from GEO database (GSE12276), Lin28B was detected in all 204 of the primary tumors (Supplementary Data 1). Notably, Lin28B expression was strongly associated with lung metastasis, but not with brain or bone metastasis (Table S6). Moreover, Lin28B was over-expressed in multiple human tumor types (PMID: 19483683). Yuan et al. reported that Lin28B is highly expressed in 65% of patients with human colon cancer (98/150) and is associated with tumor progression and poor prognosis (PMID: 29669301). Viswanathan et al. verified that Lin28B expression was observed in 20% of human hepatic cell carcinomas (HCCs) (19/98) and associated with advanced disease in patients with HCC (PMID: 19483683). In addition, Lin28B was found to be highly expressed in a discrete subset of human tumors including Wilms' tumor, ovarian carcinoma, and germ cell tumors (PMID: 19483683). It is unknown why *Lin28B* mRNA levels are lower in the TCGA database. This inconsistency may reflect the source of the cohorts used by ourselves and the other groups noted here, which differs from that of the TCGA cohort. Additionally, in our study the *Lin28B* mRNA expression level was still significantly higher (Fig. 8d), a result that was consistent with the expression level of Lin28B protein (Fig. 1c).

“ Therefore, before to evaluate the study further I propose:

1) The authors should identify breast cancer cell lines that express high levels of endogenous Lin28b to use in this study and to obtain CRISPR/CAS9 Lin28b KO counterparts to assess the most relevant immunological/metastatic phenotypes analysed in the study.”

Response: We thank this reviewer for the valuable comments. We performed the experiments using only the 4TO7 cell line because of the challenge we experienced in identifying a mouse cell line that can disseminate to the distal organs but cannot produce visible metastatic lesions, therefore providing a model for our hypothesis. We had detected Lin28B expression in multiple mouse breast cancer cell lines that were available to us (67NR, 168, 4TO7, 4T1, EMT6, E0771 and NF639) and Lin28B is not expressed in any of those mouse cell lines (Fig. R4, please see page 10). This observation precluded our assessment of the effect of Lin28B knockdown in any of these mouse cell lines.

The human breast cancer cell line MDA-MB-231, which is invasive and highly metastatic, expresses high levels of Lin28B (Fig. R4 and Supplementary Fig. 1l). However, due to immune rejection, this human breast cancer cell line can be used to produce tumors and induce metastasis only in immunodeficient animal hosts (e.g., NOD/SCID mice); however, the immunodeficient nature of the NOD-SCID mouse precludes studying the immune-regulatory function of Lin28B. This observation explains why we did not use a human breast cancer cell model in this manuscript.

To further confirm Lin28B's function in BC metastasis, we employed *MMTV-Neu* (a HER2-positive mouse BC model) with *Lin28B^{Kl}* mice. *MMTV-Neu-Lin28B* mice were generated by crossing the *MMTV-Neu* mouse expressing an activated rat *c-Neu* oncogene with *MMTV-Cre* and *Lin28B^{Kl}* mice. The results showed that Lin28B over-expression increased the lung metastatic incidence of *MMTV-Neu* mice (Supplementary Fig. 1j). In addition, Lin28B over-expression increased the proportions of the lung neutrophils (Supplementary Fig. 2i), notably including those displaying N2 phenotype (Supplementary Fig. 3b), and induced an immune-suppressive pre-metastatic niche (Supplementary Fig. 5h-j). Moreover, Lin28B expression resulted in the production of the serum exosomes containing lower let-7s in the tumor-bearing mice (Supplementary Fig. 8c), supporting the pro-metastatic mechanism whereby Lin28B induces an immune-suppressive pre-metastatic niche through neutrophil N2 conversion.

Using the *PyMT* BC model, we performed additional experiments and obtained further results as follows: Firstly, Lin28B expression induced neutrophil accumulation in the pre-metastatic niche (Supplementary Fig. 2i). Secondly, Lin28B expression also induced neutrophil N2 conversion in the pre-metastatic niche (Supplementary Fig. 3a). Thirdly, let-7 levels were decreased in the serum exosomes in *PyMT-Lin28* mice (Fig. 7b). Importantly, the *PyMT-Lin28B-exosome* education permitted the establishment of an immune-suppressive lung microenvironment (Supplementary Fig. 6i). These results are consistent with the results from the 4TO7 and *MMTV-Neu* BC models.

Due to lack of a suitable mouse BC cell line that expresses a higher level of Lin28B, we cannot currently verify the effect of Lin28B knockdown on immune suppression using an in vivo model. However, using the mouse BC models (4TO7, PyMT and MMTV-Neu), we obtained consistent results in support of the hypothesis that Lin28B induces immune suppression through N2 neutrophils and exosomes with with let-7 levels. In the near future, we hope to identify a mouse BC cell line with high endogenous Lin28B expression, which would permit the performance of the critical relevant experiments proposed by the reviewers.

"2) Confirm the clinical relevance of Lin28b using gene expression data from independent Breast cancer cohorts from public databases (TCGA, GEO) and focus on Lin28b and let-7."

Response: We thank this reviewer for this suggestion. To confirm the clinical relevance of Lin28B expression, we firstly performed a IHC assay for Lin28B expression in an additional TMA cohort, Cohort III (Supplementary Data 1, 145 cases). Combined with the results from TMA II, our data showed that high Lin28B expression correlated with poor prognosis and was an independent prognosticator of overall BC survival for all three BC subtypes (Fig. 1h and Table S3-5). This result indicated that the correlation of Lin28B with poor prognosis of patients with BC is rather a general phenomenon, and not limited to TNBC. Secondly, we analyzed the Gene Expression Omnibus (GEO) breast cancer cohort for the transcriptomic datasets of the clinical samples. We analyzed the expression profiles of a cohort of 204 patients with detail metastatic information (GSE12276, Supplementary Data 1), to identify the potential correlation of Lin28B

with metastatic site. The clinical analysis indicated that Lin28B expression associated strongly with lung metastasis, but not with brain or bone metastasis ($P=0.045$ vs 0.300 and 0.369 , see Table S6).

For let-7, we searched the RNA-sequencing data of the GDC TCGA breast cancer clinical cohort (<https://xenabrowser.net>, 1073 BC patients) and the results indicated that lower expression of let-7a was associated with reduced overall survival of patients with BC (Fig. 8j). A similar correlation also was observed with let-7b, let-7c and let-7d (Supplementary Fig. 9i). In addition, with the clinical cohort of 204 patients with BC from GSE12276, we observed that lower let-7a expression in BC tumors correlated with increased lung metastasis, but not with brain or bone metastasis (Table 7). Importantly, the lower let-7a expression predicted reduced lung-metastasis-free survival (Fig. 8k). Taken together, the independent BC cohorts from two public databases (TCGA, GEO) supported our hypothesis that higher Lin28B or lower let-7a is a predicted signal for poor prognosis and increased lung metastasis.

“3) Measure Lin28b by Western blotting in their own samples (including cancer and normal) to make sure that the signal in the immunochemistry samples really corresponds to Lin28b protein.”

Response: We thank the reviewer for suggesting this experiment. Using western blotting, we determined that our anti-Lin28B Ab reacts primarily with the 35-Kd protein in MDA-MB-231 cells (Fig. R10), and the intensity of this band was attenuated in sh-Lin28B-transfected MDA-MB-231 cells (Supplementary Fig. 11), demonstrating that the 35-Kd protein indeed corresponded to authentic Lin28B protein. In response to this reviewer’s suggestion, we have modified the text as necessary. The sentence “The human breast cancer cell line MDA-MB-231, which is invasive and highly metastatic, expresses high levels of Lin28B. With this cell line, we confirmed that the anti-Lin28B Ab reacts primarily with the 35-Kd endogenous Lin28B protein (data not shown)” were added in page 7, paragraph 2, line 2 in the revised manuscript.

Fig. R10 Lin28B Ab reacts only with the endogenous Lin28B protein in MDA-MB-231 cells.

(a) Lin28B expression was analyzed by western blotting in MCF-7 (line 1) and MDA-MB-231 cells (line 2). Showing that the Lin28B Ab reacts only with the 35-Kd Lin28B protein. Asterisk (*):

Non-specific band. (b) β -actin was analyzed by western blotting with the same membrane as in panel a. Due to this experiment was performed after Lin28B blotting (panel a), the Lin28B signal

kept visible (labeled by blue arrow).

“4) Evaluate the relevance of let-7 in this pathway by repressing the miRNA in cells that are KO for Lin28b”

Response: We thank the reviewer for this comment. We agree that repressing miRNA expression in cells that are KO for Lin28B would be an important tool for evaluating the relevance of let-7 in this pathway. As noted above, the mouse BC cell lines to which we had access did not express Lin28B (Fig. R4, please see page 10) and the MDA-MB-231 cells could not be used to generate tumors in immunocompetent mice; therefore, we performed in vitro experiments using MDA-MB-231 cells. The results showed that Lin28B KD decreased IL-6 induction in MDA-MB-231 cells (Supplementary Fig. 9k). Then, let-7a inhibitor restored the IL-6 production inhibited by Lin28B KD (Supplementary Fig. 9k), which was comparable to the levels seen with MDA-MB-231 cells without a Lin28B KD (Supplementary Fig. 9k). In addition, a similar trend was also found in IL-10 expression (Supplementary Fig. 9k), suggesting that Lin28B affects IL-6 and IL-10 production through let-7s and that let-7s play an important role in Lin28B-mediated IL-6 and IL-10 up-regulation. We have added the description of this result to the Result section in the revised MS. The sentence was added as following : “Consistently, let-7a inhibition rescued decreased IL-6 and IL-10 production caused by Lin28B knockdown in MDA-MB-231 cells (Supplementary Fig. 9k), supporting that let-7s play an important role in regulating IL-6 and IL-10 expression in the cells” (page 23, paragraph 1, line 20 in the revised manuscript).

“Additionally figure legends are poorly written and mostly lack important details. For example no information on which housekeeping genes were used for normalisation in RT-qPCR.”

Response: We are sorry that the information was too general in the figure legends. Per this reviewer’s suggestion, we carefully checked and modified all the figure legends. We added details throughout, and expect that these revisions will facilitate understanding by the reader of how the experiments were performed. In addition, for the qRT-PCR experiments, we now indicate all the housekeeping genes used for normalization.

Reviewers' Comments:

Reviewer #1:

Remarks to the Author:

Authors addressed all my critiques.

Reviewer #2:

Remarks to the Author:

In the revised manuscript, the authors have addressed all the concerns of this reviewer.

Please add control lungs for the immunohistochemistry panels for the lung pre-metastatic niches in Figure 2d. Otherwise, the novelty within the revised manuscript is greatly enhanced.

Reviewer #3:

Remarks to the Author:

I am very pleased to find that the authors have addressed all my comments. I appreciate the great effort that was done to address my comments and I believe that the manuscript is much improved.

I am also pleased to see that the suggestion of adding a model was taken on board. Here I have one last comment, I am surprised to see that the model does not make clear that the target cells for the exosome and the fibroblast, which in turn release CXCLs and influence the level of neutrophils and macrophages IL6 and IL10 levels to induce an immune-suppressive environment. I think, given the fact that the exosome target cells if one of the key differences with the previous work from Liu, Y., et al., Cancer Cell 2016, placing the role of lung fibroblast clearly in the model seems important.

This is the only think I would suggest adjusting in the final version.

Reviewer #4:

Remarks to the Author:

I don't have any additional comments

Reviewer #2 (Remarks to the Author):

In the revised manuscript, the authors have addressed all the concerns of this reviewer.

Please add control lungs for the immunohistochemistry panels for the lung pre-metastatic niches in Figure 2d. Otherwise, the novelty within the revised manuscript is greatly enhanced.

Response: We thank reviewer for this comment. We performed a immunohistochemistry (IHC) experiment to monitor the pre-metastatic niche by using anti-Fibronectin (FN) Ab, as is known that FN was upregulated after tumor-cell inoculation (PMID: 16341007), and FN was considered as typical pre-metastatic niche factor (PMID: 28481877, 27505671). Our results suggested that tumor inoculation significantly increased FN expression in the pre-metastatic lung (Fig. R1), indicating the additional production of FN by tumor growth. Moreover, the data consistent with previous study (PMID: 16341007). We observed that FN more frequently appeared at the peritracheal and peribronchial (Fig. R1 and Supplementary Fig. 2i, red arrow), suggesting that the pre-metastatic niche was preferentially performed near the trachea and bronchus. As expected, Lin28B increased FN expression in the pre-metastatic niche compared with the control (Fig. R1 and Supplementary Fig. 2i, red arrow), suggesting the enhanced niche production induced by Lin28B. Similarly, neutrophils also showed increased aggregation at the peritracheal and peribronchial (Fig. R1 and Supplementary Fig. 2i, blue arrow), supporting the conclusion that increased neutrophil enrichment in the pre-metastatic niche with Lin28B high expression. We have added this result as Supplementary Fig. S2i. We also added a description of this result to the Result section in the revised MS. The added sentence is as follows: "With immunohistochemistry (IHC) assay, we further revealed that Lin28B significantly increased the expression of the niche characteristic gene Fibronectin (FN) in the pre-metastatic lung. It was frequently observed at the peritracheal and peribronchial (red arrow, Supplementary Fig. 2i). Moreover, neutrophils showed similar accumulation (blue arrow, Supplementary Fig. 2i), suggesting the increased neutrophil distribution in the pre-metastatic niche induced by Lin28B" (page 8, paragraph 2, line 8 in the revised manuscript).

Fig. R1 IHC staining of FN and Ly6G in the pre-metastatic lung.

The lung tissues from naive mice or 4TO7 tumor-bearing mice at 2 weeks after tumor inoculation were staining with the anti-FN or anti-Ly6G Abs. The experiment was repeated three times independently with similar results. Data from one representative experiment are shown. Scale bars: 200 μ m. It was found that the pre-metastatic niche was more evident at the peritracheal and peribronchial as observed by FN staining (red arrow).

Reviewer #3 (Remarks to the Author):

I am very pleased to find that the authors have addressed all my comments. I appreciate the great effort that was done to address my comments and I believe that the manuscript is much improved.

I am also pleased to see that the suggestion of adding a model was taken on board. Here I have one last comment, I am surprised to see that the model does not make clear that the target cells for the exosome and the fibroblast, which in turn release CXCLs and influence the level of neutrophils and macrophages IL6 and IL10 levels to induce an immune-suppressive environment. I think, given the fact that the exosome target cells if one of the key differences with the previous work from Liu, Y., et al., Cancer Cell 2016, placing the role of lung fibroblast clearly in the model seems important.

This is the only think I would suggest adjusting in the final version.

Response: We thank reviewer for this suggestion. We agree with reviewer that we should incorporate fibroblasts into the model. We have drawn this picture again and taken fibroblasts into the model. Due to the text limitations, the original Supplementary Fig. S9m was changed to Supplementary Fig. S10 in the revised manuscript.